# One-Prompt Strikes Back: Sparse Mixture of Experts for Prompt-based Continual Learning

**Minh Le**[*1], **Bao Ngoc Dao**[*4], **Huy Nguyen**[2], **Quyen Tran**[5], **Anh Nguyen**[3], **Nhat Ho**[2]

[1] Trivita AI     [2] The University of Texas at Austin     [3] Northeastern University

[4] Hanoi University of Science and Technology     [5] Rutgers University

## Abstract

Prompt-based methods have recently gained prominence in Continual Learning (CL) due to their strong performance and memory efficiency. A prevalent strategy in this paradigm assigns a dedicated subset of prompts to each task, which, while effective, incurs substantial computational overhead and causes memory requirements to scale linearly with the number of tasks. Conversely, approaches employing a single shared prompt across tasks offer greater efficiency but often suffer from degraded performance due to knowledge interference. To reconcile this trade-off, we propose **SMoPE**, a novel framework that integrates the benefits of both task-specific and shared prompt strategies. Inspired by recent findings on the relationship between Prefix Tuning and Mixture of Experts (MoE), SMoPE organizes a shared prompt into multiple "prompt experts" within a sparse MoE architecture. For each input, only a select subset of relevant experts is activated, effectively mitigating interference. To facilitate expert selection, we introduce a prompt-attention score aggregation mechanism that computes a unified proxy score for each expert, enabling dynamic and sparse activation. Additionally, we propose an adaptive noise mechanism to encourage balanced expert utilization while preserving knowledge from prior tasks. To further enhance expert specialization, we design a prototype-based loss function that leverages prefix keys as implicit memory representations. Extensive experiments across multiple CL benchmarks demonstrate that SMoPE consistently outperforms task-specific prompt methods and achieves performance competitive with state-of-the-art approaches, all while significantly reducing parameter counts and computational costs. Our code is publicly available at `https://github.com/Minhchuyentoancbn/SMoPE`.

## 1 Introduction

*Continual Learning* (CL) is a critical research area focused on enabling neural networks to learn from a sequence of tasks in dynamic environments while retaining knowledge from previous tasks (Aljundi et al., 2017; De Lange et al., 2021; Zhang et al., 2023; Tran et al., 2024). A primary challenge in CL is *catastrophic forgetting*, where performance on earlier tasks deteriorates as new ones are learned (McCloskey & Cohen, 1989; French, 1999; Mehta et al., 2023). Recently, prompt-based approaches have gained attention as a promising direction in CL, offering strong performance and high memory efficiency (Wang et al., 2022b;a; Smith et al., 2023; Le et al., 2024a). These methods adapt pre-trained models using a small set of learnable parameters, referred to as *prompts*, which function as task-specific instructions to guide adaptation and alleviate forgetting. Several studies have demonstrated the effectiveness of prompt-based methods, reporting state-of-the-art results across a range of CL benchmarks.

A common strategy in prompt-based CL is to allocate a distinct subset of prompt parameters to each task (Wang et al., 2022a; 2023; Le et al., 2024a; Tran et al., 2025). This task-specific partitioning helps mitigate interference by isolating knowledge within separate prompt modules. While effective, such approaches face several notable limitations. First, when task identity is unknown at inference time, the model must infer the appropriate prompt for each input. Existing methods often rely on

---
[*]Equal Contribution

forwarding the input through the full pre-trained model to compute a query, introducing *non-negligible computational overhead* (Huang et al., 2024; Kim et al., 2024). Second, assigning dedicated prompts to each task hinders scalability and limits knowledge sharing. *As the number of tasks increases, the number of learnable prompt parameters grows linearly*, making this approach inefficient for long-term continual learning. Moreover, strict prompt partitioning restricts transferability by preventing the reuse or adaptation of previously learned prompts, thereby limiting positive knowledge transfer across tasks.

In contrast to task-specific prompt methods, Huang et al. (2024) recently introduced OVOR, a highly parameter-efficient and computationally lightweight approach that employs a single shared prompt across all tasks. Despite its efficiency, OVOR *underperforms relative to state-of-the-art task-specific prompt methods*. We hypothesize that this performance gap arises from excessive knowledge interference: because the same prompt is continually updated across tasks, it struggles to retain task-specific information, leading to degraded performance. This raises a key question:

> *(Q) Can we strike a balance between these two paradigms, retaining the parameter efficiency of a single prompt while achieving performance competitive with task-specific approaches?*

Most prompt-based CL methods rely on *Prefix Tuning* (Li & Liang, 2021) to integrate prompts into pre-trained models. However, recent work by Le et al. (2024a) offers a new perspective: each attention head can be viewed as a composition of multiple *Mixture of Experts* (MoE) models (Jacobs et al., 1991; Shazeer et al., 2017), and prefix tuning effectively introduces new *prompt experts* into these models. Building on this insight, we propose **SMoPE** (Sparse Mixture of Prompt Experts), a novel method that retains a single shared prompt while structuring it as multiple prompt experts within a sparse MoE framework.

SMoPE introduces a *prompt-attention score aggregation* mechanism that computes a unified proxy score for each expert, enabling sparse and dynamic expert selection for each input. Like OVOR, SMoPE maintains a single shared prompt across tasks for high parameter efficiency. However, unlike OVOR, which updates all prompt components uniformly, SMoPE selectively activates and updates only a small, relevant subset of experts for each input. This targeted update mechanism reduces interference and promotes positive transfer by reusing previously learned experts across tasks. A common challenge in sparse MoE architectures is the risk of imbalanced expert utilization. To address this while preserving learned knowledge, we introduce an *adaptive noise mechanism* that encourages the use of underutilized experts without overwriting important ones. Furthermore, to enhance expert specialization in the CL setting, we propose a novel *prototype loss* that leverages prefix keys as an implicit memory of past tasks. Extensive experiments on standard CL benchmarks show that SMoPE significantly outperforms task-specific prompt methods despite using a single shared prompt. Moreover, SMoPE matches or exceeds the performance of current state-of-the-art methods while reducing computational cost by up to **50%** and requiring substantially fewer learnable parameters.

**Contributions.** The primary contributions of this work are: **1.** We propose **SMoPE**, a novel approach that integrates a sparse mixture of experts architecture into the prefix tuning framework, featuring a prompt-attention score aggregation mechanism for efficient and dynamic expert selection. **2.** We introduce an adaptive noise mechanism to encourage balanced expert utilization without overwriting prior knowledge, and a prototype-based loss that leverages prefix keys as implicit memory. **3.** We show that SMoPE achieves state-of-the-art results on multiple CL benchmarks, while using significantly fewer parameters and reducing computation compared to existing prompt-based methods.

## 2 BACKGROUND AND RELATED WORK

### 2.1 CONTINUAL LEARNING

*Continual Learning* (CL) involves training a model on a sequence of tasks $\{\mathcal{D}_1, \ldots, \mathcal{D}_T\}$. Each task $\mathcal{D}_t = \{(\boldsymbol{x}_i^t, y_i^t)\}_{i=1}^{\mathcal{N}_t}$ contains $\mathcal{N}_t$ samples, where $\boldsymbol{x}_i^t \in \mathcal{X}^t$ is an input and $y_i^t \in \mathcal{Y}^t$ is its corresponding label. The primary objective is for the model, after being trained on task $t$, to perform well on the set of all classes encountered up to task $t$, denoted by $\mathcal{Y}^{1:t} = \bigcup_{i=1}^t \mathcal{Y}^i$. We focus on the challenging class-incremental learning scenario (Van de Ven & Tolias, 2019; Wang et al., 2022b; 2023), where the label spaces of different tasks are disjoint, *i.e.,* $\mathcal{Y}^t \bigcap \mathcal{Y}^{t'} = \varnothing$ for any $t \neq t'$. A standard constraint in CL is that data from previous tasks $\mathcal{D}_1, \ldots, \mathcal{D}_{t-1}$ is unavailable when training on the current task $\mathcal{D}_t$. This sequential training process, without access to past data, makes the model susceptible to

*catastrophic forgetting* (McCloskey & Cohen, 1989; Nguyen et al., 2019; Mehta et al., 2023), wherein performance on previously learned tasks degrades significantly as the model adapts to new ones.

## 2.2 PROMPT-BASED CONTINUAL LEARNING

In vision tasks, prompting is typically applied to the *Vision Transformer* (ViT) (Dosovitskiy et al., 2020), which consists of a sequence of *Multi-Head Self-Attention* (MSA) (Vaswani et al., 2017) blocks. To illustrate how prompts are integrated, we first describe the standard MSA mechanism. Let the input to an MSA layer be a sequence of token embeddings $[\boldsymbol{x}_1, \ldots, \boldsymbol{x}_N]^\top \in \mathbb{R}^{N \times d}$, where $N$ is the sequence length and $d$ is the embedding dimension. The MSA operation is defined as follows:

$$\text{MSA}(\boldsymbol{X}^Q, \boldsymbol{X}^K, \boldsymbol{X}^V) = \text{Concat}(\boldsymbol{h}_1, \ldots, \boldsymbol{h}_m)W^O \in \mathbb{R}^{N \times d},$$

$$\boldsymbol{h}_i = \text{Attention}(\boldsymbol{X}^Q W_i^Q, \boldsymbol{X}^K W_i^K, \boldsymbol{X}^V W_i^V), \ i = 1, \ldots, m, \tag{1}$$

where $\boldsymbol{X}^Q = \boldsymbol{X}^K = \boldsymbol{X}^V = [\boldsymbol{x}_1, \ldots, \boldsymbol{x}_N]^\top$ are the query, key, and value matrices, respectively; $m$ is the number of attention heads; and $W^O \in \mathbb{R}^{md_v \times d}$, $W_i^Q \in \mathbb{R}^{d \times d_k}$, $W_i^K \in \mathbb{R}^{d \times d_k}$, and $W_i^V \in \mathbb{R}^{d \times d_v}$ are projection matrices, with $d_k = d_v = \frac{d}{m}$.

Most prompt-based methods (Wang et al., 2022a; 2023; Jiao et al., 2024) adopt *Prefix Tuning* (Li & Liang, 2021; Le et al., 2024b), which introduces learnable prefix key $\boldsymbol{P}^K \in \mathbb{R}^{N_p \times d}$ and prefix value $\boldsymbol{P}^V \in \mathbb{R}^{N_p \times d}$ parameters, prepended to the input key and value matrices of the MSA layer:

$$f_{\text{PreT}}(\boldsymbol{X}^Q, \boldsymbol{X}^K, \boldsymbol{X}^V) = \text{MSA}\left(\boldsymbol{X}^Q, \begin{bmatrix} \boldsymbol{P}^K \\ \boldsymbol{X}^K \end{bmatrix}, \begin{bmatrix} \boldsymbol{P}^V \\ \boldsymbol{X}^V \end{bmatrix}\right) = \text{Concat}(\hat{\boldsymbol{h}}_1, \ldots, \hat{\boldsymbol{h}}_m)W^O. \tag{2}$$

During training, only the prompt parameters $\boldsymbol{P}^K$ and $\boldsymbol{P}^V$ are updated, while the pre-trained ViT parameters, including $W^O$, $W_i^Q$, $W_i^K$, and $W_i^V$, remain frozen.

Conventional prompt-based continual learning methods, such as DualPrompt (Wang et al., 2022a), HiDe-Prompt (Wang et al., 2023), and NoRGa (Le et al., 2024a), typically allocate a separate set of prompt parameters for each task. However, recent work by Huang et al. (2024) demonstrates that a single prompt shared across tasks can achieve performance comparable to that of task-specific prompting, raising questions about the efficiency and utilization of prompts in prior approaches. While subsequent works have attempted to address related issues, they still present limitations. For instance, the mechanism proposed by Gao et al. (2024) to mitigate incorrect prompt selection at test time still results in a linear growth of prompts with the number of tasks. Similarly, the strategy of Roy et al. (2024) to manage prompt growth by employing Large Language Models (LLMs) introduces significant computational overhead due to its reliance on an external model.

## 2.3 MIXTURE OF EXPERTS

*Mixture of Experts* (MoE) extends classical mixture models by introducing an adaptive gating mechanism (Jacobs et al., 1991; Jordan & Jacobs, 1994). For a given input $\boldsymbol{h} \in \mathbb{R}^d$, the MoE model computes outputs from $N'$ *experts* $f_j : \mathbb{R}^d \rightarrow \mathbb{R}^{d_v}$, which are then combined using weights from a learned *gating function* $G : \mathbb{R}^d \rightarrow \mathbb{R}^{N'}$ as follows:

$$\hat{\mathbf{y}} = \sum_{j=1}^{N'} G(\boldsymbol{h})_j \cdot f_j(\boldsymbol{h}) = \sum_{j=1}^{N'} \frac{\exp\left(s_j(\boldsymbol{h})\right)}{\sum_{\ell=1}^{N'} \exp\left(s_\ell(\boldsymbol{h})\right)} \cdot f_j(\boldsymbol{h}), \tag{3}$$

where $s_j : \mathbb{R}^d \rightarrow \mathbb{R}$ denotes the *score function* for expert $f_j$. To enhance scalability, Shazeer et al. (2017) introduced the *Sparse Mixture of Experts* (SMoE), a variant that activates only the top-$K$ experts with the highest scores. The set of these indices, denoted $K_{\boldsymbol{h}}$, is formally defined as:

$$K_{\boldsymbol{h}} = \underset{S \subseteq \{1, \ldots, N'\} : |S| = K}{\arg\max} \sum_{j \in S} s_j(\boldsymbol{h}). \tag{4}$$

The final output is then computed as:

$$\hat{\mathbf{y}} = \sum_{j \in K_{\boldsymbol{h}}} \frac{\exp\left(s_j(\boldsymbol{h})\right)}{\sum_{\ell \in K_{\boldsymbol{h}}} \exp\left(s_\ell(\boldsymbol{h})\right)} \cdot f_j(\boldsymbol{h}). \tag{5}$$

By routing only a small subset of experts per input, SMoE achieves high model capacity with limited computational overhead. This property has driven adoption in large-scale systems, including language models (Fedus et al., 2022; Jiang et al., 2024; Comanici et al., 2025), computer vision (Riquelme et al., 2021; Xue et al., 2023), and multi-task learning (Ma et al., 2018; Yang et al., 2024).

## 3 METHODOLOGY

### 3.1 MIXTURE OF EXPERTS AND PREFIX TUNING

We adopt prefix tuning as our primary prompting strategy, following prior work. Furthermore, we use *a single prompt shared across all tasks*, similar to Huang et al. (2024). Recent findings by Le et al. (2024a; 2025) show that each attention head in a ViT can be viewed as a structured composition of multiple MoE models, framing prefix tuning as a way to add new experts to this structure.

Specifically, consider the output of the $l$-th head in Equation (2), denoted as $\hat{\boldsymbol{h}}_l = [\hat{\boldsymbol{h}}_{l,1}, \ldots, \hat{\boldsymbol{h}}_{l,N}]^\top \in \mathbb{R}^{N \times d_v}$. Let $\boldsymbol{X} = \left[\boldsymbol{x}_1^\top, \ldots, \boldsymbol{x}_N^\top\right]^\top \in \mathbb{R}^{Nd}$ denote the concatenated input embeddings to the MSA layer, and let $\boldsymbol{P}^K = \left[\boldsymbol{p}_1^K, \ldots, \boldsymbol{p}_{N_p}^K\right]^\top \in \mathbb{R}^{N_p \times d}$ and $\boldsymbol{P}^V = \left[\boldsymbol{p}_1^V, \ldots, \boldsymbol{p}_{N_p}^V\right]^\top \in \mathbb{R}^{N_p \times d}$.

We interpret each attention head as comprising $N$ *pre-trained experts* $f_j : \mathbb{R}^{Nd} \to \mathbb{R}^{d_v}$, together with $N_p$ *prompt experts* $f_{N+j'} : \mathbb{R}^{Nd} \to \mathbb{R}^{d_v}$ introduced via prefix tuning:

$$f_j(\boldsymbol{X}) = W_l^{V\top} \boldsymbol{x}_j = W_l^{V\top} E_j \boldsymbol{X}, \quad f_{N+j'}(\boldsymbol{X}) = W_l^{V\top} \boldsymbol{p}_{j'}^V,$$

for $j = 1, \ldots, N$, and $j' = 1, \ldots, N_p$. Here, $E_j \in \mathbb{R}^{d \times Nd}$ is a selection matrix such that $E_j \boldsymbol{X} = \boldsymbol{x}_j$. The corresponding score functions are defined as:

$$s_{i,j}(\boldsymbol{X}) = \frac{\boldsymbol{x}_i^\top W_l^Q W_l^{K\top} \boldsymbol{x}_j}{\sqrt{d_v}} = \frac{\boldsymbol{X}^\top E_i^\top W_l^Q W_l^{K\top} E_j \boldsymbol{X}}{\sqrt{d_v}},$$

$$s_{i,N+j'}(\boldsymbol{X}) = \frac{\boldsymbol{x}_i^\top W_l^Q W_l^{K\top} \boldsymbol{p}_{j'}^K}{\sqrt{d_v}} = \frac{\boldsymbol{X}^\top E_i^\top W_l^Q W_l^{K\top} \boldsymbol{p}_{j'}^K}{\sqrt{d_v}},$$

for $i = 1, \ldots, N$, $j = 1, \ldots, N$, and $j' = 1, \ldots, N_p$. With these definitions, the output of the $i$-th token in head $l$ can be expressed as an MoE model:

$$\hat{\boldsymbol{h}}_{l,i} = \sum_{j=1}^N \frac{\exp(s_{i,j}(\boldsymbol{X}))}{\sum_{k=1}^N \exp(s_{i,k}(\boldsymbol{X})) + \sum_{k'=1}^{N_p} \exp(s_{i,N+k'}(\boldsymbol{X}))} f_j(\boldsymbol{X})$$

$$+ \sum_{j'=1}^{N_p} \frac{\exp(s_{i,N+j'}(\boldsymbol{X}))}{\sum_{k=1}^N \exp(s_{i,k}(\boldsymbol{X})) + \sum_{k'=1}^{N_p} \exp(s_{i,N+k'}(\boldsymbol{X}))} f_{N+j'}(\boldsymbol{X}). \quad (6)$$

Thus, each attention head can be regarded as a multi-gate MoE model, where each output token $\hat{\boldsymbol{h}}_{l,i}$ is itself an MoE model. Crucially, only the prefix parameters $\boldsymbol{P}^K$ and $\boldsymbol{P}^V$ are learnable. Adaptation is therefore restricted to training the prompt experts $f_{N+j'}$ and their score functions $s_{i,N+j'}$. These experts complement the pre-trained experts in the attention head, enabling efficient task adaptation.

### 3.2 SPARSE MIXTURE OF PROMPT EXPERTS

The preceding analysis reveals that *even when a single prompt is shared across tasks, multiple prompt experts are still added to the pre-trained model*. Their simultaneous activation leads to repeated updates across tasks, inducing inter-task interference. To address this, we propose **SMoPE**, a novel sparse mixture of experts architecture built on prefix tuning. By selectively activating only a subset of relevant prompt experts, SMoPE introduces implicit parameter partitioning that mitigates interference.

**Prompt-Attention Score Aggregation.** Unlike standard MoEs where each expert has one score function, the MoE architecture within the attention head utilizes a multi-gate mechanism. Specifically, under prefix tuning, each prompt expert $f_{N+j'}$ has $N$ score functions $s_{i,N+j'}$ for $i = 1, \ldots, N$, rendering direct application of the standard SMoE framework non-trivial and computationally intensive. To reduce this complexity, we introduce a unified *proxy score* that aggregates these individual scores:

$$\tilde{s}_{j'}(\boldsymbol{X}) = \sum_{i=1}^N \frac{s_{i,N+j'}(\boldsymbol{X})}{N} = \frac{\boldsymbol{X}^\top \tilde{E}^\top W_l^Q W_l^{K\top} \boldsymbol{p}_{j'}^K}{\sqrt{d_v}} = \frac{\tilde{\boldsymbol{x}}^\top W_l^Q W_l^{K\top} \boldsymbol{p}_{j'}^K}{\sqrt{d_v}}, \quad (7)$$

where $\tilde{E} = \frac{1}{N} \sum_{i=1}^N E_i$ is the mean extraction matrix and $\tilde{\boldsymbol{x}} = \frac{1}{N} \sum_{i=1}^N \boldsymbol{x}_i$ denotes the average token representation. *Importantly, this formulation eliminates the need to compute all individual*

*scores $s_{i,N+j'}$; computing the single vector $\tilde{\boldsymbol{x}}$ suffices to obtain the proxy scores $\tilde{s}_{j'}$.* These proxy scores are then used to determine expert selection and output composition:

$$\hat{\boldsymbol{h}}_{l,i} = \sum_{j=1}^{N} \frac{\exp(s_{i,j}(\boldsymbol{X}))}{\sum_{k=1}^{N}\exp(s_{i,k}(\boldsymbol{X})) + \sum_{k'=1}^{N_p}\exp(\tilde{s}_{k'}(\boldsymbol{X}))} f_j(\boldsymbol{X})$$
$$+ \sum_{j' \in K_{\boldsymbol{X}}} \frac{\exp(\tilde{s}_{j'}(\boldsymbol{X}))}{\sum_{k=1}^{N}\exp(s_{i,k}(\boldsymbol{X})) + \sum_{k'=1}^{N_p}\exp(\tilde{s}_{k'}(\boldsymbol{X}))} f_{N+j'}(\boldsymbol{X}). \quad (8)$$

**Sample Complexity of Estimating Prompt Experts.** Although the prompt-attention score aggregation is designed to activate only a subset of relevant prompt experts, we show in Appendix A that the SMoPE model in Equation (8) maintains the same sample complexity for estimating prompt experts as the standard MoE in Equation (6). Specifically, estimating the prompt experts $f_{N+j'}(\boldsymbol{X})$ in each MoE variant with an estimation error of $\tau > 0$ requires at most a polynomial number of data points of the order $\mathcal{O}(\tau^{-4})$. Therefore, SMoPE is as sample-efficient as the standard MoE for prefix tuning.

**Sparse Prompt Expert Selection.** Each prompt expert $f_{N+j'}$ is now associated with a single unified score $\tilde{s}_{j'}$, enabling the efficient selection of the top-$K$ most relevant experts using Equation (4). This yields the active expert set $K_{\boldsymbol{X}}$, and Equation (8) becomes:

$$\hat{\boldsymbol{h}}_{l,i} = \sum_{j=1}^{N} \frac{\exp(s_{i,j}(\boldsymbol{X}))}{\sum_{k=1}^{N}\exp(s_{i,k}(\boldsymbol{X})) + \sum_{k' \in K_{\boldsymbol{X}}}\exp(\tilde{s}_{k'}(\boldsymbol{X}))} f_j(\boldsymbol{X})$$
$$+ \sum_{j' \in K_{\boldsymbol{X}}} \frac{\exp(\tilde{s}_{j'}(\boldsymbol{X}))}{\sum_{k=1}^{N}\exp(s_{i,k}(\boldsymbol{X})) + \sum_{k' \in K_{\boldsymbol{X}}}\exp(\tilde{s}_{k'}(\boldsymbol{X}))} f_{N+j'}(\boldsymbol{X}). \quad (9)$$

Similar to OVOR (Huang et al., 2024), SMoPE employs a single prompt shared across all tasks, ensuring high parameter efficiency. However, unlike OVOR, SMoPE activates only a sparse subset of $K$ prompt experts per input $\boldsymbol{X}$ within each MSA layer, introducing implicit parameter partitioning. *By updating only the relevant parameters*, SMoPE reduces knowledge interference and mitigates catastrophic forgetting. Importantly, this parameter selection relies solely on the current layer's input $\boldsymbol{X}$ and does not require precomputed query representations from a full model forward pass, as in prior task-specific prompting methods (Wang et al., 2022a; 2023; Smith et al., 2023; Le et al., 2024a). As a result, *SMoPE offers substantial reductions in computational overhead* compared to these approaches (Kim et al., 2024). Furthermore, by dynamically reusing relevant experts across tasks, SMoPE naturally promotes knowledge sharing and transfer.

**Implementation in Attention Layers.** We now describe how the MoE formulation in Equation (9) integrates with the attention mechanism. In prefix tuning, the attention matrix $A_l$ for the $l$-th head is constructed as follows:

$$A_l = \left[A_l^{\text{prompt}},\ A_l^{\text{pre-trained}}\right] = \frac{\boldsymbol{X}^Q W_l^Q W_l^{K\top} \left[\boldsymbol{P}^{K\top},\ \boldsymbol{X}^{K\top}\right]}{\sqrt{d_v}}$$
$$= \left[\frac{\boldsymbol{X}^Q W_l^Q W_l^{K\top} \boldsymbol{P}^{K\top}}{\sqrt{d_v}},\ \frac{\boldsymbol{X}^Q W_l^Q W_l^{K\top} \boldsymbol{X}^{K\top}}{\sqrt{d_v}}\right]. \quad (10)$$

This decomposition shows that the full attention matrix comprises scores from both the prompt experts ($A_l^{\text{prompt}}$) and the original pre-trained attention ($A_l^{\text{pre-trained}}$). Since SMoPE modifies only the prompt components, we adjust $A_l^{\text{prompt}}$ while keeping $A_l^{\text{pre-trained}}$ unchanged. Based on Equation (7), this adjustment requires only the computation of $\tilde{\boldsymbol{x}}$. The SMoPE-adjusted attention matrix is then:

$$\tilde{A}_l = \left[\tilde{A}_l^{\text{prompt}},\ A_l^{\text{pre-trained}}\right], \quad \tilde{A}_l^{\text{prompt}} = \text{TopK}\left(\frac{\tilde{\boldsymbol{x}}^\top W_l^Q W_l^{K\top} \boldsymbol{P}^K}{\sqrt{d_v}}\right).\text{expand}(N, -1). \quad (11)$$

Here, the Top-$K$ operator selects the $K$ most relevant prompt experts based on the proxy scores $\tilde{s}_{j'}$, which are computed once per input and shared across all output tokens $\hat{\boldsymbol{h}}_{l,i}$. Notably, while conventional prefix tuning computes $N$ scores per prompt expert at a cost of $\mathcal{O}(Nd_k)$, SMoPE reduces this to a single proxy score at $\mathcal{O}(d_k)$, yielding up to an $N$-fold reduction in complexity.

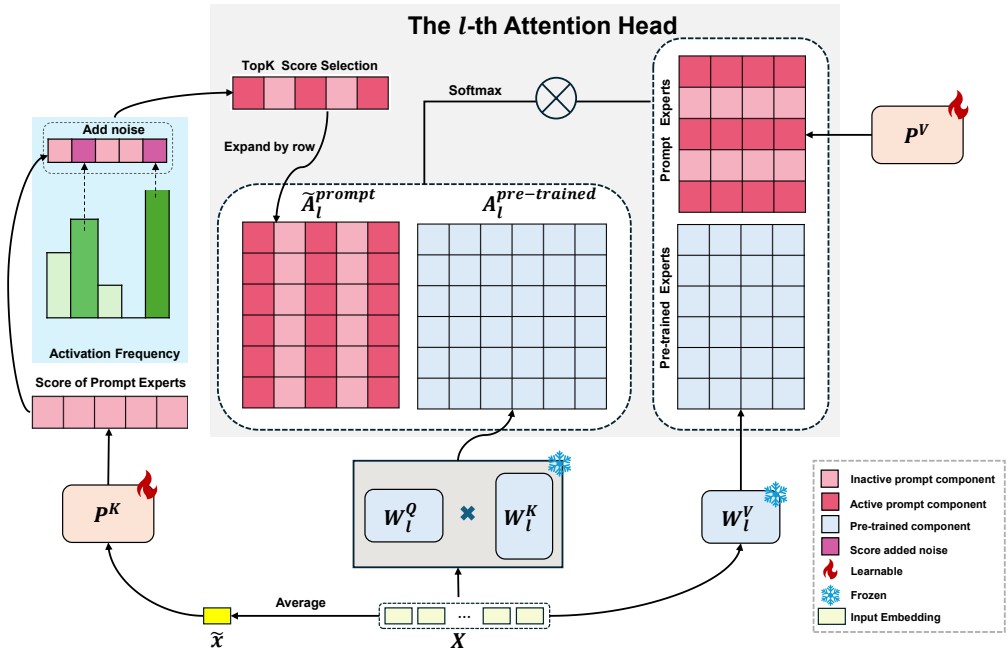

Figure 1: **SMoPE Implementation in Attention Layers.** The attention mechanism for each head is composed of both pre-trained and prompt components. The pre-trained attention matrix $A_l^{\text{pre-trained}}$ is computed using standard self-attention. To construct the prompt attention matrix $\tilde{A}_l^{\text{prompt}}$, we first calculate the average input representation $\tilde{x}$, and evaluate the scores for all prompt experts. During training, frequently activated prompt experts are penalized by applying an adaptive noise to their scores, which promotes exploration of underutilized experts for new tasks while preserving essential knowledge in critical experts. A Top-$K$ selection operator then identifies the most relevant experts based on these adjusted scores. The selected scores are row-expanded to form $\tilde{A}_l^{\text{prompt}}$. Finally, $\tilde{A}_l^{\text{prompt}}$ is concatenated with $A_l^{\text{pre-trained}}$ to produce the final attention matrix, which is applied to the expert representations via a dot product, similar to the standard self-attention mechanism.

## 3.3 BALANCING EXPERT UTILIZATION WITH ADAPTIVE NOISE

A well-known challenge in training SMoE models is the *imbalance in expert utilization*, where a small subset of experts dominates routing decisions while others remain underutilized or inactive (Mu & Lin, 2025). This imbalance reduces the diversity of learned representations and ultimately limits model effectiveness. We find that SMoPE exhibits a similar issue: during training, only a small group of prompt experts is consistently activated across tasks. While SMoPE mitigates interference by selectively activating experts, repeatedly relying on the same subset still risks knowledge interference.

**Adaptive Noise Mechanism.** To address this issue, we introduce a novel adaptive noise mechanism that promotes more balanced expert utilization in continual learning, as follows:

$$K_{\boldsymbol{X}} = \underset{S \subseteq \{1,\ldots,N_p\}:|S|=K}{arg\,max} \sum_{j' \in S} \left( \tilde{s}_{j'}(\boldsymbol{X}) - \epsilon_{j'} \right), \tag{12}$$

$$\epsilon_{j'} = \begin{cases} \epsilon \cdot (\max_j \tilde{s}_j(\boldsymbol{X}) - \min_j \tilde{s}_j(\boldsymbol{X})) & \text{if } \texttt{train} \text{ and } F_{j'} \geq \frac{1}{N_p} \sum_{j=1}^{N_p} F_j \\ 0 & \text{otherwise} \end{cases} \tag{13}$$

where $\epsilon \in [0,1]$ is a hyperparameter, and $F_{j'}$ denotes the accumulated activation frequency of prompt expert $f_{N+j'}$, *i.e.,* the proportion of instances in which expert $f_{N+j'}$ was selected across previous tasks. We define a prompt expert as *important* if its activation frequency exceeds the average across all prompt experts within its attention head. These experts, having been frequently activated, are expected to encode essential prior knowledge. The adaptive noise mechanism specifically targets these important experts, applying the noise penalty only to those with activation frequencies above the mean. *This penalization encourages the selection of underutilized experts when adapting to new tasks, while protecting knowledge stored in the important experts that are already highly activated.* Please refer to Figure 1 for an illustration.

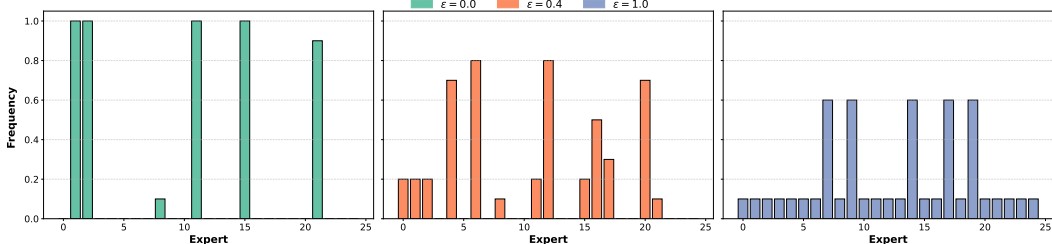

Figure 2: **Activation Frequencies of Prompt Experts.** Results on CUB-200 with the prompt length $N_p = 25$ and $K = 5$. We show one representative attention head and visualize the frequency with which prompt experts are activated after training on all tasks under different values of $\epsilon$.

As shown in Figure 2, setting $\epsilon = 0.0$ results in the repeated activation of a small subset of experts, while $\epsilon = 1.0$ spans the entire dynamic score range, heavily penalizing frequently used experts and enforcing full expert utilization. Intermediate values provide a smooth trade-off between exploration and stability. By balancing expert utilization in this way, SMoPE can more effectively adapt to new tasks while preserving the knowledge encoded in important experts, thereby mitigating catastrophic forgetting. Further discussion and experimental results can be found in Appendix D.3.

### 3.4 PREFIX KEY PROTOTYPES FOR EXPERT SPECIALIZATION

**Promoting Expert Specialization.** During both training and inference, each MSA layer receives an input $\boldsymbol{X}$ and selects a subset of prompt experts. To facilitate this selection, we encourage expert specialization, whereby each expert focuses on distinct regions of the input space. This enables more reliable routing, as specialized experts are more easily identifiable for a given input. To promote such specialization, we introduce the following objective:

$$\mathcal{L}_{\text{router}} = -\sum_{j' \in K_{\boldsymbol{X}}} \frac{\exp\left(\tilde{s}_{j'}(\boldsymbol{X})\right)}{\sum_{k' \in K_{\boldsymbol{X}}} \exp\left(\tilde{s}_{k'}(\boldsymbol{X})\right) + \sum_{k' \notin K_{\boldsymbol{X}}} \exp\left(\tilde{s}_{k'}(\boldsymbol{X})\right)}. \tag{14}$$

This objective encourages higher scores $\tilde{s}_{j'}(\boldsymbol{X})$ for experts in the selected set $K_{\boldsymbol{X}}$ while suppressing the scores of unselected experts. As a result, experts develop more distinct input-space specializations, reducing redundancy and improving routing accuracy.

**Prefix Keys as Prototypes.** From Equation (7), optimizing $\tilde{s}_{j'}(\boldsymbol{X})$ corresponds to updating the prefix key $\boldsymbol{p}_{j'}^K$. Since the score function determines expert activation, *the prefix key effectively defines the region of the input space in which each expert specializes*. Thus, optimizing $\mathcal{L}_{\text{router}}$ requires updating both selected and unselected prefix keys.

However, in continual learning, this poses a challenge: past-task data are unavailable, and updating unselected prefix keys may lead to overwriting prior specializations. To address this, we propose treating *prefix keys from earlier tasks as prototypes*, which serve as implicit memory representations of past input distributions. Specifically, we define the prototype set: $\mathcal{D}_{\text{proto}} = \left\{ \boldsymbol{p}_{\text{old},j'}^K \mid 1 \leq j' \leq N_p,\ F_{j'} \geq \frac{1}{N_p} \sum_{j=1}^{N_p} F_j \right\}$, where $\boldsymbol{p}_{\text{old},j'}^K$ denotes the prefix key after training on the previous task. Only frequently activated experts are retained to avoid noisy or uninformative prototypes. We then define the prototype-based objective:

$$\mathcal{L}_{\text{proto}} = -\sum_{\boldsymbol{p} \in \mathcal{D}_{\text{proto}}} \sum_{j' \in K_{\boldsymbol{p}}} \frac{\exp\left(\boldsymbol{p}^\top \boldsymbol{p}_{j'}^K\right)}{\sum_{k'=1}^{N_p} \exp\left(\boldsymbol{p}^\top \boldsymbol{p}_{k'}^K\right)}, \tag{15}$$

where $K_{\boldsymbol{p}} = \underset{S \subseteq \{1,\ldots,N_p\}:|S|=K}{arg\,max} \sum_{j' \in S} \left(\boldsymbol{p}^\top \boldsymbol{p}_{j'}^K\right)$ denotes the top-$K$ experts activated by prototype $\boldsymbol{p}$. While $\mathcal{L}_{\text{router}}$ promotes specialization using current-task data, $\mathcal{L}_{\text{proto}}$ preserves specializations learned from previous tasks, thereby mitigating catastrophic forgetting without relying on past inputs.

**Additional Training Strategies.** In continual learning, the MLP classifier head often develops a bias toward newly introduced classes (Hou et al., 2019; Belouadah & Popescu, 2019). To mitigate this, we adopt *task-adaptive prediction*, following prior work (Wang et al., 2023; Jiao et al., 2024; Le et al., 2024a). This technique adjusts predictions using the representation statistics of previously learned

Table 1: Performance comparison on ImageNet-R, CIFAR-100, and CUB-200 (10-task splits). ↑ indicates higher is better. FAA and CAA are averaged over 5 runs. **Bold** denotes the best results, excluding joint training.

| Method | ImageNet-R | | CIFAR-100 | | CUB-200 | |
|---|---|---|---|---|---|---|
| | FAA (↑) | CAA (↑) | FAA (↑) | CAA (↑) | FAA (↑) | CAA (↑) |
| Joint-Train | 82.06 | | 91.38 | | 88.00 | |
| L2P++ | $69.29 \pm 0.73$ | $78.30 \pm 0.69$ | $82.50 \pm 1.10$ | $88.96 \pm 0.82$ | $77.18 \pm 0.71$ | $84.31 \pm 0.17$ |
| Deep L2P++ | $71.66 \pm 0.64$ | $79.63 \pm 0.90$ | $84.30 \pm 1.03$ | $90.50 \pm 0.69$ | $78.64 \pm 0.60$ | $86.14 \pm 0.57$ |
| DualPrompt | $71.32 \pm 0.62$ | $78.94 \pm 0.72$ | $82.37 \pm 0.29$ | $87.10 \pm 0.55$ | $77.27 \pm 0.31$ | $84.82 \pm 0.15$ |
| CODA-Prompt | $75.45 \pm 0.56$ | $81.59 \pm 0.82$ | $87.02 \pm 0.10$ | $91.61 \pm 0.91$ | $76.65 \pm 0.70$ | $84.16 \pm 0.23$ |
| OVOR | $75.25 \pm 0.21$ | $79.78 \pm 0.65$ | $86.91 \pm 0.35$ | $91.02 \pm 1.00$ | $77.45 \pm 0.69$ | $85.81 \pm 1.56$ |
| HiDe-Prompt | $74.25 \pm 0.19$ | $79.64 \pm 0.53$ | $88.27 \pm 0.59$ | $91.38 \pm 0.78$ | $85.60 \pm 0.05$ | $90.16 \pm 0.64$ |
| NoRGa | $74.39 \pm 0.08$ | $79.68 \pm 0.41$ | $88.79 \pm 0.13$ | $92.06 \pm 0.51$ | $85.78 \pm 0.24$ | $90.63 \pm 0.58$ |
| ConvPrompt | $77.30 \pm 0.45$ | $81.46 \pm 0.67$ | $88.76 \pm 0.36$ | $92.49 \pm 0.89$ | $82.56 \pm 0.61$ | $86.68 \pm 0.95$ |
| CPrompt | $77.14 \pm 0.11$ | $82.92 \pm 0.70$ | $87.82 \pm 0.21$ | $92.53 \pm 0.23$ | $80.35 \pm 0.44$ | $85.66 \pm 0.59$ |
| VQ-Prompt | $78.71 \pm 0.22$ | $83.24 \pm 0.68$ | $88.73 \pm 0.27$ | $92.84 \pm 0.73$ | $86.72 \pm 0.94$ | $90.33 \pm 1.03$ |
| **SMoPE** | $\mathbf{79.32} \pm 0.42$ | $\mathbf{84.39} \pm 0.77$ | $\mathbf{89.23} \pm 0.12$ | $\mathbf{93.67} \pm 0.82$ | $\mathbf{87.43} \pm 0.39$ | $\mathbf{91.11} \pm 0.55$ |

Table 2: Performance comparison on ImageNet-R (10-task split) for self-supervised pre-training, iBOT-1K, and DINO-1K. ↑ indicates higher is better. FAA and CAA are averaged over 5 runs. **Bold** highlights the best results.

| Method | iBOT-1K | | DINO-1K | |
|---|---|---|---|---|
| | FAA (↑) | CAA (↑) | FAA (↑) | CAA (↑) |
| DualPrompt | $61.51 \pm 1.05$ | $67.11 \pm 0.08$ | $58.57 \pm 0.45$ | $64.89 \pm 0.15$ |
| CODA-Prompt | $66.56 \pm 0.68$ | $73.14 \pm 0.57$ | $63.15 \pm 0.39$ | $69.73 \pm 0.25$ |
| HiDe-Prompt | $65.34 \pm 0.37$ | $71.67 \pm 0.56$ | $62.94 \pm 0.22$ | $70.04 \pm 0.70$ |
| NoRGA | $65.81 \pm 0.52$ | $72.11 \pm 0.52$ | $63.02 \pm 0.11$ | $70.25 \pm 0.64$ |
| OVOR | $69.46 \pm 0.46$ | $74.95 \pm 0.63$ | $66.55 \pm 0.30$ | $72.89 \pm 0.71$ |
| VQ-Prompt | $71.68 \pm 0.72$ | $76.66 \pm 0.40$ | $68.42 \pm 0.28$ | $74.43 \pm 0.58$ |
| **SMoPE** | $\mathbf{72.17} \pm 0.36$ | $\mathbf{77.24} \pm 0.57$ | $\mathbf{68.61} \pm 0.41$ | $\mathbf{75.14} \pm 0.54$ |

classes to correct classifier bias and stabilize prompt learning. Additionally, inspired by advances in SMoE training (Wu et al., 2022; Lin et al., 2024; Cai et al., 2025), we apply *dense expert training* (*i.e.,* without sparse selection) during the initial epochs of the first task. This aids in establishing stable expert representations before enabling sparse routing.

**Final Optimization Objective.** The complete loss function for our SMoPE framework is:

$$\mathcal{L} = \mathcal{L}_{\text{ce}} + \alpha_{\text{router}} \cdot \mathcal{L}_{\text{router}} + \alpha_{\text{proto}} \cdot \mathcal{L}_{\text{proto}}, \tag{16}$$

where $\mathcal{L}_{\text{ce}}$ is the standard cross-entropy loss, and $\alpha_{\text{router}}$ and $\alpha_{\text{proto}}$ are weighting hyperparameters. During training, only the prefix parameters and classifier head are updated, while the pre-trained backbone remains frozen. Please refer to Appendix B for further details on the training algorithm.

# 4 EXPERIMENTS

## 4.1 EXPERIMENTAL DETAILS

**Datasets.** Following Jiao et al. (2024), we evaluate SMoPE on three representative CL benchmarks: ImageNet-R (Boschini et al., 2022), CIFAR-100 (Krizhevsky et al., 2009), and CUB-200 (Wah et al., 2011). ImageNet-R consists of 200 challenging classes, including difficult samples from ImageNet and newly collected data with diverse stylistic variations, and is split into 5, 10, or 20 disjoint tasks. CIFAR-100 contains 100 classes, which we randomly partition into 10 tasks. CUB-200 comprises fine-grained images of 200 bird species, randomly divided into 10 tasks with 20 classes each.

**Evaluation Metrics.** We adopt two widely used CL metrics: Final Average Accuracy (FAA) and Cumulative Average Accuracy (CAA). FAA is the average accuracy after training on all tasks, while CAA represents the mean of the FAA values recorded after each task.

**Baselines.** We compare our approach against several representative prompt-based continual learning methods, including L2P (Wang et al., 2022b), DualPrompt (Wang et al., 2022a) and CODA-Prompt (Smith et al., 2023). Additionally, we evaluate against recent state-of-the-art methods, such

as ConvPrompt (Roy et al., 2024), CPrompt (Gao et al., 2024), and VQ-Prompt (Jiao et al., 2024). We also consider OVOR (Huang et al., 2024), which employs a single prompt, and state-of-the-art task-specific prompt methods such as HiDe-Prompt (Wang et al., 2023) and NoRGa (Le et al., 2024a). For L2P, we follow Smith et al. (2023) and include two variants: "L2P++", which replaces prompt tuning with prefix tuning, and "Deep L2P++", which extends L2P++ by inserting prompts into the first five MSA blocks. Finally, we report the performance of "Joint-Train", an offline upper bound obtained by training on all tasks simultaneously.

**Implementation Details.** All experiments are conducted on a single NVIDIA A100 GPU. To ensure a fair comparison, all baseline methods are re-implemented using the configurations reported in their original papers. Following Jiao et al. (2024), we adopt a ViT-B/16 backbone pretrained on ImageNet-1K (Russakovsky et al., 2015) and ImageNet-21K (Ridnik et al., 2021) as the backbone. We also evaluate self-supervised ViT-B/16 models from iBOT-1K (Zhou et al., 2021) and DINO-1K (Caron et al., 2021). The prompt length is fixed at $N_p = 25$, and the number of selected prompt experts is set to $K = 5$ for all experiments. Prompts are inserted into the first six MSA blocks. SMoPE is optimized using AdamW (Loshchilov & Hutter, 2017) with a cosine learning rate decay schedule. The batch size is set to 64 for ImageNet-R, and 128 for both CIFAR-100 and CUB-200. Additional implementation details are provided in Appendix C.

## 4.2 EXPERIMENTAL RESULTS

**Overall Comparison.** Table 1 summarizes the performance of SMoPE compared to prompt-based continual learning baselines. *SMoPE achieves the best overall results across all three benchmarks, consistently surpassing prior methods on both FAA and CAA*, demonstrating strong resistance to forgetting while effectively adapting to new tasks. On CIFAR-100 and CUB-200, OVOR, which employs a single shared prompt, performs competitively with L2P, DualPrompt, and CODA-Prompt but falls short of task-specific methods such as HiDe-Prompt and NoRGa. Notably, SMoPE breaks this trade-off: despite using only a single shared prompt, it not only outperforms OVOR by a large margin but also surpasses task-specific methods, indicating that its structured design overcomes the limitations of shared prompting. Its performance is also comparable to that of VQ-Prompt, a recent state-of-the-art method. These results highlight that SMoPE mitigates forgetting while maintaining the efficiency of a shared prompt, achieving a favorable balance between effectiveness and efficiency without relying on task-specific model expansions.

**Performance with Different Pre-training Paradigms.** In line with prior work (Wang et al., 2023; Jiao et al., 2024), we conduct experiments on ImageNet-R using two self-supervised pre-training paradigms: iBOT-1K (Zhou et al., 2021) and DINO-1K (Caron et al., 2021). The results, presented in Table 2, demonstrate that SMoPE surpasses all other prompt-based continual learning baselines, highlighting the generalizability and robustness of SMoPE's design across different pre-training paradigms.

**Computational Cost Analysis.** To evaluate the efficiency of SMoPE, we compare the number of learnable parameters, training GFLOPs, and inference GFLOPs, as shown in Table 3. By using a single prompt for all tasks, SMoPE significantly reduces the number of parameters. It also avoids passing the input through the full pre-trained model to compute a query, unlike prior approaches. Instead, each MSA layer selects prompt experts based on the input at that layer, resulting in a substantial reduction in computational cost up to **50%**. These design choices highlight SMoPE's efficiency over task-specific prompting methods while maintaining competitive performance.

Table 3: Comparison of computational cost on ImageNet-R (10-task split), including learnable parameters (millions), training and inference costs (GFLOPs), and relative cost (%) to L2P.

| Method | Params (M) | Train GFLOPs | Test GFLOPs |
|---|---|---|---|
| Deep L2P++ | 4.78 | 67.44 (100%) | 67.44 (100%) |
| DualPrompt | 1.10 | 33.72 (50%) | 67.44 (100%) |
| CODA-Prompt | 3.99 | 67.44 (100%) | 67.44 (100%) |
| HiDe-Prompt | 4.21 | 33.72 (50%) | 67.45 (100%) |
| NoRGa | 4.21 | 33.72 (50%) | 67.45 (100%) |
| OVOR | 0.26 | 33.72 (50%) | 33.72 (50%) |
| VQ-Prompt | 0.50 | 67.44 (100%) | 67.44 (100%) |
| **SMoPE** | **0.38** | **33.72 (50%)** | **33.72 (50%)** |

Table 4: Ablation study of the proposed SMoPE on CUB-200, split into 10 tasks. FAA and CAA are averaged over 5 runs. ↑ indicates that higher values are better. **Bold** highlights the best results.

| Training Strategy | FAA (↑) | CAA (↑) |
|---|---|---|
| One Prompt | $75.23 \pm 0.17$ | $83.61 \pm 0.48$ |
| + Prompt Score Aggregation | $75.49 \pm 0.37$ | $83.65 \pm 0.70$ |
| + Sparse Expert Selection | $79.12 \pm 0.20$ | $87.16 \pm 0.56$ |
| + Adaptive Noise | $85.36 \pm 0.21$ | $89.12 \pm 0.38$ |
| + Task-Adaptive Prediction | $86.03 \pm 0.29$ | $90.09 \pm 0.43$ |
| + Initial Dense Training | $86.27 \pm 0.46$ | $90.23 \pm 0.52$ |
| + Router Loss $\mathcal{L}_{router}$ | $87.05 \pm 0.40$ | $90.47 \pm 0.40$ |
| + Prototype Loss $\mathcal{L}_{proto}$ | $\mathbf{87.43 \pm 0.39}$ | $\mathbf{91.11 \pm 0.55}$ |

**Ablation Studies.** To evaluate the contribution of individual components in SMoPE, we conducted a series of ablation experiments (see Table 4). The baseline configuration, "One Prompt", employs standard prefix tuning with a single prompt shared across all tasks. Removing all SMoPE components results in a significant performance drop, primarily due to the continual updating of shared parameters. Introducing the prompt-attention score aggregation mechanism alters the attention computation and achieves performance comparable to the baseline, consistent with our analysis in Appendix A. Adding sparse expert selection further improves performance, although the gains are limited by imbalanced expert utilization. Notably, integrating the adaptive noise mechanism leads to substantial performance improvements, underscoring its role in reducing interference and mitigating forgetting. Incorporating all components yields the highest overall performance, confirming the complementary nature of SMoPE's design. These ablation results collectively validate that each component contributes meaningfully to the final model performance, and their integration is essential for SMoPE's effectiveness.

## 5   DISCUSSION AND CONCLUSION

In this paper, we introduced SMoPE, a prompt-based CL method that balances the paradigms of using single shared and task-specific prompts. By integrating a sparse MoE architecture into prefix tuning, SMoPE reduces knowledge interference by updating only task-relevant parameters. We also proposed an adaptive noise mechanism and a prototype loss function that leverage prefix keys as an implicit memory of previous tasks, further enhancing performance. Extensive experiments demonstrate that SMoPE effectively reduces catastrophic forgetting while maintaining adaptability to new tasks, with significantly improved parameter and computational efficiency compared to existing methods.

Despite these promising results, several avenues for future work remain. Although our approach uses a shared prompt structure across tasks, which mitigates forgetting more effectively than prior methods, interference may still arise as the number of tasks grows. Future research could explore dynamically expanding the prompt length or increasing the number of prompt experts to better address continual learning demands. Investigating how SMoPE scales with the number of prompt experts may also yield valuable insights. While our experiments focused on ViT, extending SMoPE to other foundational models would help assess its generalizability. Finally, though designed for continual learning, SMoPE's architecture may have broader applications warranting further exploration.

REPRODUCIBILITY STATEMENT

In order to facilitate the reproduction of our empirical results, we provide detailed descriptions of the experimental setup in Section 4 and Appendix C. All datasets used in this study are publicly available, enabling full replication of our experiments.

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

# Supplement to "One-Prompt Strikes Back: Sparse Mixture of Experts for Prompt-based Continual Learning"

This supplementary material provides: a theoretical analysis of prompt estimation rates when using prompt-attention score aggregation (Appendix A); the detailed training algorithm for SMoPE (Appendix B); additional experimental details (Appendix C) and results (Appendix D); and a discussion on the use of large language models in this work (Appendix E).

## A  THEORETICAL ANALYSIS OF PROMPT-ATTENTION SCORE AGGREGATION

In this appendix, we aim to compare the sample complexity of estimating prompt experts with and without prompt-attention score aggregation. To this end, we conduct a convergence analysis of prompt expert estimation. The results of the analysis reveal that employing prompt-attention score aggregation is as sample-efficient as not using it, which indicates that this choice does not affect the model's ability to adapt to new tasks. Before stating the problem formally, let us introduce the notation that we use throughout this appendix.

**Notation.** For any natural number $n \in \mathbb{N}$, let $[n] = \{1, 2, \ldots, n\}$. For a vector $u \in \mathbb{R}^d$, we use both $u = (u^{(1)}, u^{(2)}, \ldots, u^{(d)})$ and $u = (u_1, u_2, \ldots, u_d)$ interchangeably. Given a multi-index $\alpha = (\alpha_1, \alpha_2, \ldots, \alpha_d) \in \mathbb{N}^d$, we write $u^\alpha = u_1^{\alpha_1} u_2^{\alpha_2} \cdots u_d^{\alpha_d}$, $|\alpha| = \alpha_1 + \alpha_2 + \cdots + \alpha_d$, and $\alpha! = \alpha_1! \alpha_2! \cdots \alpha_d!$. The Euclidean norm of $u$ is denoted by the term $\|u\|$, while $|S|$ refers to the cardinality of any set $S$. For two positive sequences $(a_n)_{n \geq 1}$ and $(b_n)_{n \geq 1}$, we write $a_n = \mathcal{O}(b_n)$ or $a_n \lesssim b_n$ if there exists a constant $C > 0$ such that $a_n \leq C b_n$ for all $n$. The notation $a_n = \mathcal{O}_P(b_n)$ means that $a_n/b_n$ is stochastically bounded. We further write $a_n = \widetilde{\mathcal{O}}_P(b_n)$ when $a_n = \mathcal{O}_P(b_n \log^c(b_n))$, for some $c > 0$.

As established in Section 3.2, the MoE models $\hat{h}_{l,1}, \ldots, \hat{h}_{l,N}$ in each attention head share a common structure of experts and score functions. Therefore, to simplify our analysis while maintaining rigor, we focus on the first head (*i.e.*, $l = 1$) and the first row of its attention matrix (*i.e.*, $i = 1$). Within this simplified setting, we present a regression-based framework to analyze the convergence of prompt expert estimation in an MoE model for prefix tuning. This framework has previously been used to study the asymptotic behavior of other MoE-based parameter-efficient fine-tuning methods, namely, low-rank adaptation (Truong et al., 2025) and LLaMA-Adapter (Diep et al., 2025). We begin by defining the problem setting with prompt-attention score aggregation.

**Problem Setting with Prompt-Attention Score Aggregation.**  Suppose we have i.i.d. data $(\boldsymbol{X}_1, Y_1), (\boldsymbol{X}_2, Y_2), \ldots, (\boldsymbol{X}_n, Y_n) \in \mathbb{R}^{Nd} \times \mathbb{R}$ generated from the following model:

$$Y_i = g_{G^*}(\boldsymbol{X}_i) + \varepsilon_i, \quad i = 1, \ldots, n, \tag{17}$$

where the noise terms $\varepsilon_1, \ldots, \varepsilon_n$ are independent Gaussian random variables with mean zero and variance $\nu^2$. The covariates $\boldsymbol{X}_1, \ldots, \boldsymbol{X}_n$ are drawn i.i.d. from some probability distribution $\mu$. The regression function $g_{G^*}$ is composed of $N$ pre-trained experts and $N_p$ learnable prompt experts:

$$g_{G^*}(\boldsymbol{X}) = \frac{\sum_{j=1}^{N} \exp\left(\boldsymbol{X}^\top B_j^0 \boldsymbol{X} + c_j^0\right) h(\boldsymbol{X}, \eta_j^0)}{\sum_{k=1}^{N} \exp\left(\boldsymbol{X}^\top B_k^0 \boldsymbol{X} + c_k^0\right) + \sum_{k'=1}^{N_p} \exp\left((\beta_{1k'}^*)^\top W^\top \boldsymbol{X} + \beta_{0k'}^*\right)}$$
$$+ \frac{\sum_{j'=1}^{N_p} \exp\left((\beta_{1j'}^*)^\top W^\top \boldsymbol{X} + \beta_{0j'}^*\right) h(\boldsymbol{X}, \eta_{j'}^*)}{\sum_{k=1}^{N} \exp\left(\boldsymbol{X}^\top B_k^0 \boldsymbol{X} + c_k^0\right) + \sum_{k'=1}^{N_p} \exp\left((\beta_{1k'}^*)^\top W^\top \boldsymbol{X} + \beta_{0k'}^*\right)}, \tag{18}$$

Here, $G^* = \sum_{j'=1}^{N_p} \exp(\beta_{0j'}^*) \delta_{(\beta_{1j'}^*, \eta_{j'}^*)}$ represents the *mixing measure*, *i.e.,* a weighted sum of Dirac measures associated with the parameters $(\beta_{1j'}^*, \beta_{0j'}^*, \eta_{j'}^*)$ of the prompt experts. The matrix $B_j^0$ plays the same role as the matrix $\frac{E_1^\top W_1^Q W_1^{K^\top} E_j}{\sqrt{d_v}}$ in the score function $s_{1,j}(\boldsymbol{X})$, while the vector $\beta_{1j'}^*$ and the matrix $W$, respectively, correspond to the components $\frac{\bar{E}^\top W_1^Q W_1^{K^\top} \boldsymbol{p}_{j'}^K}{\sqrt{d_v}}$ used in the modified score function $\tilde{s}_{j'}(\boldsymbol{X})$. The expert functions $h(\boldsymbol{X}, \eta_j^0)$ correspond to the pre-trained experts $f_j(\boldsymbol{X})$, while $h(\boldsymbol{X}, \eta_{j'}^*)$ correspond to the new prompt experts $f_{N+j'}(\boldsymbol{X})$. In general, the expert functions can have flexible parametric forms, not restricted to the simple forms used in the linear-gating-prefix MoE model (Le et al., 2024a).

**Least Squares Estimation.** We use the least squares method (van de Geer, 2000) to estimate the unknown parameters $(\beta_{0j'}^*, \beta_{1j'}^*, \eta_{j'}^*)_{j'=1}^{N_p}$, or, equivalently, the true mixing measure $G^*$. The estimator, denoted by $\widehat{G}_n$, is defined as:

$$\widehat{G}_n = \arg \min_{G \in \mathcal{G}_{N_p'}(\Theta)} \sum_{i=1}^n \big(Y_i - g_G(\boldsymbol{X}_i)\big)^2, \tag{19}$$

where

$$\mathcal{G}_{N_p'}(\Theta) = \Big\{ G = \sum_{i=1}^\ell \exp(\beta_{0i}) \delta_{(\beta_{1i}, \eta_i)} : 1 \le \ell \le N_p', (\beta_{0i}, \beta_{1i}, \eta_i) \in \Theta \Big\}$$

is the set of mixing measures with at most $N_p'$ components. Since the true number of experts, $N_p$, is typically unknown in practice, we assume that the maximum number of components, $N_p'$, is chosen to be an upper bound such that $N_p' \ge N_p$.

**Definition A.1** (Strong Identifiability). We say an expert function $h(\cdot, \eta)$ is *strongly identifiable* if and only if it is twice differentiable with respect to its parameters, and if for any $k \ge 1$ and any set of pairwise distinct parameters $\eta_1, \ldots, \eta_k$, the collection of functions

$$\Big\{ \boldsymbol{X}^\nu \cdot \frac{\partial^{|\gamma|} h}{\partial \eta^\gamma}(\boldsymbol{X}, \eta_j) : j \in [k], \ \nu \in \mathbb{N}^{Nd}, \ \gamma \in \mathbb{N}^q, \ 0 \le |\nu| + |\gamma| \le 2 \Big\}$$

is linearly independent for almost every $\boldsymbol{X} \in \mathbb{R}^{Nd}$.

Intuitively, this condition prevents unwanted interactions between the parameters of the expert function $h(\cdot, \eta)$. In practice, it guarantees that the regression difference $\hat{g}_G(\boldsymbol{X}) - g_{G^*}(\boldsymbol{X})$ can be expanded into linearly independent terms via a Taylor expansion of expressions *i.e.,* $\exp(\beta_1^\top W^\top \boldsymbol{X}) h(\boldsymbol{X}, \eta)$. Thus, the above condition guarantees that all the derivative terms in the Taylor expansion are linearly independent. The following example illustrates a class of expert functions $h(\cdot, \eta)$ that satisfy this condition.

**Example.** Consider a neural network expert of the form $h(\boldsymbol{X}, (a, b)) = \phi(a^\top \boldsymbol{X} + b)$, where the parameters are $\eta = (a, b) \in \mathbb{R}^{Nd} \times \mathbb{R}$ and the activation function is $\phi \in \{\text{ReLU}, \text{GELU}, z \mapsto z^p\}$. This form of expert function satisfies the strong identifiability condition under mild assumptions on the activation function $\phi$.

**Voronoi Loss.** To quantify the discrepancy between an estimated measure $G$ and the target measure $G^*$, we define

$$\mathcal{D}(G, G^*) = \sum_{j' \in [N_p] : |\mathcal{V}_{j'}| > 1} \sum_{i \in \mathcal{V}_{j'}} \exp(\beta_{0i}) \Big( \|\Delta\beta_{1ij'}\|^2 + \|\Delta\eta_{ij'}\|^2 \Big)$$

$$+ \sum_{j' \in [N_p] : |\mathcal{V}_{j'}| = 1} \sum_{i \in \mathcal{V}_{j'}} \exp(\beta_{0i}) \Big( \|\Delta\beta_{1ij'}\| + \|\Delta\eta_{ij'}\| \Big)$$

$$+ \sum_{j'=1}^{N_p} \Big| \sum_{i \in \mathcal{V}_{j'}} \exp(\beta_{0i}) - \exp(\beta_{0j'}^*) \Big|. \tag{20}$$

Here, the notation $\Delta\beta_{1ij'} := \beta_{1i} - \beta_{1j'}^*$ and $\Delta\eta_{ij'} := \eta_i - \eta_{j'}^*$ represents the difference between estimated and true parameters. The partitioning of the estimated components is based on Voronoi cells. For each true component $\omega_{j'}^* := (\beta_{1j'}^*, \eta_{j'}^*)$, the corresponding Voronoi cell $\mathcal{V}_{j'} \equiv \mathcal{V}_{j'}(G)$ contains the indices of the estimated components that are closest to it. Formally, it is defined as

$$\mathcal{V}_{j'} := \Big\{ i \in \{1, 2, \ldots, N_p'\} : \|\omega_i - \omega_{j'}^*\| \le \|\omega_i - \omega_\ell^*\|, \ \forall \ell \ne j' \Big\}, \tag{21}$$

where $\omega_i := (\beta_{1i}, \eta_i)$ denotes an estimated component from $G$. Consequently, the cardinality of a cell $|\mathcal{V}_{j'}|$ indicates how many components $\omega_i$ from $G$ are used to approximate the true component $\omega_{j'}^*$ in $G^*$.

By construction, a small value of $\mathcal{D}(G, G^*)$ implies that the estimated parameters are close to the true parameters. Based on the observation, although this loss function $\mathcal{D}(G, G_*)$ is not symmetric, it

serves as a suitable measure for analyzing the convergence of the least squares estimator $\widehat{G}_n$. With this loss function defined, we can now present the sample complexity of estimating prompt experts with prompt-attention score aggregation.

**Theorem A.2.** *Under the strong identifiability condition for the expert function $h(\boldsymbol{X}, \eta)$, the least squares estimator $\widehat{G}_n$ converges to the true measure $G^*$ at the following rate:*

$$\mathcal{D}(\widehat{G}_n, G^*) = \widetilde{\mathcal{O}}_P(n^{-1/2}). \tag{22}$$

The proof of Theorem A.2 is deferred to Appendix A.1. Several remarks on this result are in order.

*(i) Convergence rate of prompt parameter estimation.* This theorem, combined with the definition of the Voronoi loss in Equation (20), implies that the convergence rates for the prompt parameters $\eta_{j'}^*$ range from $\widetilde{\mathcal{O}}_P(n^{-1/4})$ to $\widetilde{\mathcal{O}}_P(n^{-1/2})$, depending on the cardinality of the corresponding Voronoi cells $\mathcal{V}_{j'}$.

*(ii) Convergence rate of prompt expert estimation.* Denote $\widehat{G}_n := \sum_{i=1}^{N_p} \exp(\hat{\beta}_{n,0i}) \delta_{(\hat{\beta}_{n,1i}, \hat{\eta}_{n,i})}$. If the expert function $h(\boldsymbol{X}, \eta)$ is Lipschitz continuous with respect to its parameter $\eta$ for almost every $\boldsymbol{X}$, *i.e.*,

$$|h(\boldsymbol{X}, \hat{\eta}_{n,i}) - h(\boldsymbol{X}, \eta_{j'}^*)| \lesssim \|\hat{\eta}_{n,i} - \eta_{j'}^*\|, \tag{23}$$

then this property, combined with the parameter convergence rates from remark (i), implies that the rates for estimating prompt experts $h(\boldsymbol{X}, \eta_{j'}^*)$ admit the same orders as those for estimating prompt parameters $\eta_{j'}^*$, standing at $\widetilde{\mathcal{O}}_P(n^{-1/4})$ or $\widetilde{\mathcal{O}}_P(n^{-1/2})$.

*(iii) Sample complexity of estimating prompt experts.* As a consequence, when using the prompt-attention score aggregation, we need a polynomial number of data points, either $\mathcal{O}(\tau^{-4})$ or $\mathcal{O}(\tau^{-2})$, to estimate the prompt experts with a given error $\tau > 0$.

**Problem Setting without Prompt-Attention Score Aggregation.** In this setting, we assume the i.i.d. data $(\boldsymbol{X}_1, Y_1), (\boldsymbol{X}_2, Y_2), \ldots, (\boldsymbol{X}_n, Y_n) \in \mathbb{R}^{Nd} \times \mathbb{R}$ are generated from the same regression model (17), but with the following modified regression function:

$$\bar{g}_{G^*}(\boldsymbol{X}) = \frac{\sum_{j=1}^{N} \exp(\boldsymbol{X}^\top B_j^0 \boldsymbol{X} + c_j^0) \, h(\boldsymbol{X}, \eta_j^0)}{\sum_{k=1}^{N} \exp(\boldsymbol{X}^\top B_k^0 \boldsymbol{X} + c_k^0) + \sum_{k'=1}^{N_p} \exp((\beta_{1k'}^*)^\top W_{k'}^\top \boldsymbol{X} + \beta_{0k'}^*)}$$
$$+ \frac{\sum_{j'=1}^{N_p} \exp((\beta_{1j'}^*)^\top W_{j'}^\top \boldsymbol{X} + \beta_{0j'}^*) \, h(\boldsymbol{X}, \eta_{j'}^*)}{\sum_{k=1}^{N} \exp(\boldsymbol{X}^\top B_k^0 \boldsymbol{X} + c_k^0) + \sum_{k'=1}^{N_p} \exp((\beta_{1k'}^*)^\top W_{k'}^\top \boldsymbol{X} + \beta_{0k'}^*)}. \tag{24}$$

The key difference between this function and the one in Equation (18) is that the weight matrix $W$ is no longer shared across prompt experts. Instead, because there is no prompt-attention score aggregation, each expert $j'$ has its own matrix, $W_{j'}$. Accordingly, the least squares estimator is now defined as:

$$\overline{G}_n = \arg \min_{G \in \mathcal{G}_{N_p'}(\Theta)} \sum_{i=1}^{n} (Y_i - \bar{g}_G(\boldsymbol{X}_i))^2.$$

**Theorem A.3.** *Under the strong identifiability condition for the expert function $h(\boldsymbol{X}, \eta)$, the least squares estimator $\overline{G}_n$ converges to the true measure $G^*$ at the following rate:*

$$\mathcal{D}(\overline{G}_n, G^*) = \widetilde{\mathcal{O}}_P(n^{-1/2}). \tag{25}$$

It should be noted that the proof arguments for the dense gating have been included in Appendix A.1. Furthermore, since the weight matrices $W_{j'}$ are frozen during training, their dependence on the expert index $j'$ do not affect the proof arguments in Appendix A.1. In other words, the proof of Theorem A.3 can be done similarly to that of Theorem A.2, so it is omitted here.

**Comparison of Sample Complexity with and without Prompt-Attention Score Aggregation.** Comparing the convergence rates in Equations (22) and (25), we find that they are identical. This indicates that the estimation rate for the prompt parameters is unaffected by the use of prompt-attention score aggregation. As a direct consequence of the Lipschitz continuity in Equation (23), the estimation rates for the prompt experts are also preserved. We therefore conclude that incorporating prompt-attention score aggregation into the MoE model for prefix tuning does not change the sample complexity of prompt expert estimation.

## A.1 PROOF OF THEOREM A.2

**Roadmap.** The result follows once we establish the following two inequalities:

$$\inf_{G \in \mathcal{G}_{N_p'}(\Theta)} \frac{\|g_G - g_{G_*}\|_{L^2(\mu)}}{\mathcal{D}(G, G_*)} > 0, \tag{26}$$

$$\|g_{\widehat{G}_n} - g_{G_*}\|_{L^2(\mu)} = \mathcal{O}_P\left(\sqrt{\log(n)/n}\right). \tag{27}$$

We will prove Equations (26) and (27) in turn.

**Proof of Equation** (26). We split the argument into a *local* and a *global* part.

*Local part.* We use proof by contradiction to show that:

$$\lim_{\varepsilon \to 0} \inf_{G \in \mathcal{G}_{N_p'}(\Theta): \, \mathcal{D}(G, G_*) \leq \varepsilon} \frac{\|g_G - g_{G_*}\|_{L^2(\mu)}}{\mathcal{D}(G, G_*)} > 0. \tag{28}$$

Suppose, for the sake of contradiction, that the claim is false. Then there exists a sequence of mixing measures, $G_n = \sum_{i=1}^{N_p} \exp(\beta_{0i}^n) \delta_{(\beta_{1i}^n, \eta_i^n)} \in \mathcal{G}_{N_p'}(\Theta)$, such that $\mathcal{D}_n := \mathcal{D}(G_n, G_*) \to 0$ and

$$\frac{\|g_{G_n} - g_{G_*}\|_{L^2(\mu)}}{\mathcal{D}_n} \to 0. \tag{29}$$

Let $\mathcal{V}_j^n := \mathcal{V}_j(G_n)$ be the Voronoi cell corresponding to the $j$-th true component. Since $\mathcal{D}_n \to 0$, the components of $G_n$ converge to those of $G_*$. For $n$ sufficiently large, the Voronoi partition stabilizes, so we can drop the superscript $n$ and write $\mathcal{V}_j = \mathcal{V}_j^n$ for simplicity. The loss $\mathcal{D}_n$ is then given by

$$\mathcal{D}_n = \sum_{j:|\mathcal{V}_j|>1} \sum_{i \in \mathcal{V}_j} \exp(\beta_{0i}^n)\left(\|\Delta\beta_{1ij}^n\|^2 + \|\Delta\eta_{ij}^n\|^2\right)$$

$$+ \sum_{j:|\mathcal{V}_j|=1} \sum_{i \in \mathcal{V}_j} \exp(\beta_{0i}^n)\left(\|\Delta\beta_{1ij}^n\| + \|\Delta\eta_{ij}^n\|\right) + \sum_{j=1}^{N_p} \left|\sum_{i \in \mathcal{V}_j} \exp(\beta_{0i}^n) - \exp(\beta_{0j}^*)\right|, \tag{30}$$

where $\Delta\beta_{1ij}^n := \beta_{1i}^n - \beta_{1j}^*$ and $\Delta\eta_{ij}^n := \eta_i^n - \eta_j^*$. Since $\mathcal{D}_n \to 0$, we have $(\beta_{1i}^n, \eta_i^n) \to (\beta_{1j}^*, \eta_j^*)$ for $i \in \mathcal{V}_j$, and the weights aggregate correctly: $\sum_{i \in \mathcal{V}_j} \exp(\beta_{0i}^n) \to \exp(\beta_{0j}^*)$. We now split the proof of the local part into three steps.

**Step 1 — Taylor expansion.** In this step, we want to decompose the following quantity into a combination of linearly independent elements:

$$Q_n(\boldsymbol{X}) := \left[\sum_{i'=1}^N \exp(\boldsymbol{X}^\top B_{i'}^0 \boldsymbol{X} + c_{i'}^0) + \sum_{j'=1}^{N_p} \exp((\beta_{1j'}^*)^\top W^\top \boldsymbol{X} + \beta_{0j'}^*)\right] [g_{G_n}(\boldsymbol{X}) - g_{G_*}(\boldsymbol{X})] \tag{31}$$

We first rewrite it as follows:

$$Q_n(\boldsymbol{X}) = \sum_{j=1}^{N_p} \sum_{i \in \mathcal{V}_j} \exp(\beta_{0i}^n)\left[\exp((\beta_{1i}^n)^\top W^\top \boldsymbol{X})h(\boldsymbol{X}; \eta_i^n) - \exp((\beta_{1j}^*)^\top W^\top \boldsymbol{X})h(\boldsymbol{X}; \eta_j^*)\right]$$

$$- \sum_{j=1}^{N_p} \sum_{i \in \mathcal{V}_j} \exp(\beta_{0i}^n)\left[\exp((\beta_{1i}^n)^\top W^\top \boldsymbol{X}) - \exp((\beta_{1j}^*)^\top W^\top \boldsymbol{X})\right]g_{G_n}(\boldsymbol{X})$$

$$+ \sum_{j=1}^{N_p} \left(\sum_{i \in \mathcal{V}_j} \exp(\beta_{0i}^n) - \exp(\beta_{0j}^*)\right) \exp((\beta_{1j}^*)^\top W^\top \boldsymbol{X})\left[h(\boldsymbol{X}; \eta_j^*) - g_{G_n}(\boldsymbol{X})\right]$$

$$:= A_n(\boldsymbol{X}) - B_n(\boldsymbol{X}) + C_n(\boldsymbol{X}). \tag{32}$$

**Decomposition of $A_n(\boldsymbol{X})$.** Let us denote $E(\boldsymbol{X};\beta_1) := \exp(\beta_1^\top W^\top \boldsymbol{X})$, then $A_n$ can be separated into two terms as follows:

$$
A_n(\boldsymbol{X}) := \sum_{j:|\mathcal{V}_j|=1} \sum_{i\in\mathcal{V}_j} \exp(\beta_{0i}^n)\Big[E(\boldsymbol{X};\beta_{1i}^n)h(\boldsymbol{X};\eta_i^n) - E(\boldsymbol{X};\beta_{1j}^*)h(\boldsymbol{X};\eta_j^*)\Big]
$$
$$
+ \sum_{j:|\mathcal{V}_j|>1} \sum_{i\in\mathcal{V}_j} \exp(\beta_{0i}^n)\Big[E(\boldsymbol{X};\beta_{1i}^n)h(\boldsymbol{X};\eta_i^n) - E(\boldsymbol{X};\beta_{1j}^*)h(\boldsymbol{X};\eta_j^*)\Big]
$$
$$
:= A_{n,1}(\boldsymbol{X}) + A_{n,2}(\boldsymbol{X}).
$$

By means of the first-order Taylor expansion, we have

$$
A_{n,1}(\boldsymbol{X}) = \sum_{j:|\mathcal{V}_j|=1} \sum_{i\in\mathcal{V}_j} \frac{\exp(\beta_{0i}^n)}{\alpha!} \sum_{|\alpha|=1} (\Delta\beta_{1ij}^n)^{\alpha_1}(\Delta\eta_{ij}^n)^{\alpha_2} \frac{\partial^{|\alpha_1|}E}{\partial\beta_1^{\alpha_1}}(\boldsymbol{X};\beta_{1j}^*)\frac{\partial^{|\alpha_2|}h}{\partial\eta^{\alpha_2}}(\boldsymbol{X};\eta_j^*) + R_{n,1}(\boldsymbol{X})
$$
$$
= \sum_{j:|\mathcal{V}_j|=1} \sum_{|\alpha_1|+|\alpha_2|=1} S_{n,j,\alpha_1,\alpha_2} \frac{\partial^{|\alpha_1|}E}{\partial\beta_1^{\alpha_1}}(\boldsymbol{X};\beta_{1j}^*)\frac{\partial^{|\alpha_2|}h}{\partial\eta^{\alpha_2}}(\boldsymbol{X};\eta_j^*) + R_{n,1}(\boldsymbol{X}),
$$

where $R_{n,1}(\boldsymbol{X})$ is a Taylor remainder such that $R_{n,1}(\boldsymbol{X})/\mathcal{D}_n \to 0$ as $n \to \infty$, and

$$
S_{n,j,\alpha_1,\alpha_2} := \sum_{i\in\mathcal{V}_j} \frac{\exp(\beta_{0i}^n)}{\alpha!}(\Delta\beta_{1ij}^n)^{\alpha_1}(\Delta\eta_{ij}^n)^{\alpha_2}.
$$

On the other hand, by applying the second-order Taylor expansion, we get that

$$
A_{n,2}(\boldsymbol{X}) = \sum_{j:|\mathcal{V}_j|>1} \sum_{1\leq|\alpha_1|+|\alpha_2|\leq 2} S_{n,j,\alpha_1,\alpha_2} \frac{\partial^{|\alpha_1|}E}{\partial\beta_1^{\alpha_1}}(\boldsymbol{X};\beta_{1j}^*)\frac{\partial^{|\alpha_2|}h}{\partial\eta^{\alpha_2}}(\boldsymbol{X};\eta_j^*) + R_{n,2}(\boldsymbol{X}),
$$

in which $R_{n,2}(\boldsymbol{X})$ is a Taylor remainder such that $R_{n,2}(\boldsymbol{X})/\mathcal{D}_n \to 0$ as $n \to \infty$.

**Decomposition of $B_n$.** Recall that we have

$$
B_n(\boldsymbol{X}) = \sum_{j:|\mathcal{V}_j|=1} \sum_{i\in\mathcal{V}_j} \exp(\beta_{0i}^n)\Big[E(\boldsymbol{X};\beta_{1i}^n) - E(\boldsymbol{X};\beta_{1j}^*)\Big]g_{G_n}(\boldsymbol{X})
$$
$$
+ \sum_{j:|\mathcal{V}_j|>1} \sum_{i\in\mathcal{V}_j} \exp(\beta_{0i}^n)\Big[E(\boldsymbol{X};\beta_{1i}^n) - E(x;\beta_{1j}^*)\Big]g_{G_n}(\boldsymbol{X})
$$
$$
:= B_{n,1}(\boldsymbol{X}) + B_{n,2}(\boldsymbol{X}).
$$

By invoking first-order and second-order Taylor expansions to $B_{n,1}(\boldsymbol{X})$ and $B_{n,2}(\boldsymbol{X})$, it follows that

$$
B_{n,1}(\boldsymbol{X}) = \sum_{j:|\mathcal{V}_j|=1} \sum_{|\ell|=1} T_{n,j,\ell} \cdot \frac{\partial^{|\ell|}E}{\partial\beta_1^\ell}(\boldsymbol{X};\beta_{1j}^*)g_{G_n}(\boldsymbol{X}) + R_{n,3}(\boldsymbol{X}),
$$
$$
B_{n,2}(\boldsymbol{X}) = \sum_{j:|\mathcal{V}_j|>1} \sum_{1\leq|\ell|\leq 2} T_{n,j,\ell} \cdot \frac{\partial^{|\ell|}E}{\partial\beta_1^\ell}(\boldsymbol{X};\beta_{1j}^*)g_{G_n}(\boldsymbol{X}) + R_{n,4}(\boldsymbol{X}),
$$

where we define

$$
T_{n,j,\ell} := \sum_{i\in\mathcal{V}_j} \frac{\exp(\beta_{0i}^n)}{\ell!}(\Delta\beta_{1ij}^n)^\ell.
$$

Additionally, $R_{n,3}(\boldsymbol{X})$ and $R_{n,4}(\boldsymbol{X})$ are Taylor remainders such that $R_{n,3}(\boldsymbol{X})/\mathcal{D}_n \to 0$ and $R_{n,3}(\boldsymbol{X})/\mathcal{D}_n \to 0$ as $n \to \infty$.

Collect the above results together, we can represent $Q_n(x)$ as

$$
Q_n(\boldsymbol{X}) = \sum_{j=1}^{N_p} \sum_{0\leq|\alpha_1|+|\alpha_2|\leq 2} S_{n,j,\alpha_1,\alpha_2} \frac{\partial^{|\alpha_1|}E}{\partial\beta_1^{\alpha_1}}(\boldsymbol{X};\beta_{1j}^*)\frac{\partial^{|\alpha_2|}h}{\partial\eta^{\alpha_2}}(\boldsymbol{X};\eta_j^*),
$$
$$
- \sum_{j=1}^{N_p} \sum_{0\leq|\ell|\leq 2} T_{n,j,\ell} \cdot \frac{\partial^{|\ell|}E}{\partial\beta_1^\ell}(\boldsymbol{X};\beta_{1j}^*)g_{G_n}(\boldsymbol{X}) + \sum_{i=1}^4 R_{n,i}(\boldsymbol{X}), \tag{33}
$$

where we define $S_{n,j,\mathbf{0}_{d\times d},\mathbf{0}_q} = T_{n,j,\mathbf{0}_{d\times d}} = \sum_{i\in\mathcal{V}_j} \exp(\beta_{0i}^n) - \exp(\beta_{0j}^*)$ for any $j \in [N_p]$.

**Step 2 — Non-vanishing coefficients.** Here, we show that *not all* normalized coefficients can vanish. Specifically, we prove that at least one of the ratios $S_{n,j,\alpha_1,\alpha_2}/\mathcal{D}_n$ or $T_{n,j,\ell}/\mathcal{D}_n$ does *not* converge to $0$ as $n \to \infty$. Assume by the contrary: for every $j \in [N_p]$ and $0 \le |\alpha_1|, |\alpha_2|, |\ell| \le 2$,

$$\frac{S_{n,j,\alpha_1,\alpha_2}}{\mathcal{D}_n} \to 0, \qquad \frac{T_{n,j,\ell}}{\mathcal{D}_n} \to 0.$$

In particular,

$$\frac{1}{\mathcal{D}_n}\sum_{j=1}^{N_p}\Big|\sum_{i\in\mathcal{V}_j}\exp(\beta_{0i}^n) - \exp(\beta_{0j}^*)\Big| = \sum_{j=1}^{N_p}\Big|\frac{S_{n,j,\mathbf{0}_{d\times d},\mathbf{0}_q}}{\mathcal{D}_n}\Big| \to 0. \tag{34}$$

First, consider indices $j$ for which the Voronoi cell is a singleton, *i.e.,* $|\mathcal{V}_j| = 1$.

- Fix $u, v \in [Nd]$, take $\alpha_1 \in \mathbb{N}^{Nd \times Nd}$ with $\alpha_1^{(uv)} = 1$ and $\alpha_2 = \mathbf{0}_q$. Then

$$\frac{1}{\mathcal{D}_n}\sum_{i\in\mathcal{V}_j}\exp(\beta_{0i}^n)\,|(\Delta\beta_{1ij}^n)^{(uv)}| = \Big|\frac{S_{n,j,\alpha_1,\alpha_2}}{\mathcal{D}_n}\Big| \to 0.$$

  Summing over $u, v$ and using equivalence of $\ell_1$ and $\ell_2$ norms yields

$$\frac{1}{\mathcal{D}_n}\sum_{i\in\mathcal{V}_j}\exp(\beta_{0i}^n)\,\|\Delta\beta_{1ij}^n\| \to 0. \tag{35}$$

- Fix $u \in [q]$, take $\alpha_1 = \mathbf{0}_{Nd\times Nd}$ and $\alpha_2 \in \mathbb{N}^q$ with $\alpha_2^{(u)} = 1$. Then

$$\frac{1}{\mathcal{D}_n}\sum_{i\in\mathcal{V}_j}\exp(\beta_{0i}^n)\,|(\Delta\eta_{ij}^n)^{(u)}| = \Big|\frac{S_{n,j,\alpha_1,\alpha_2}}{\mathcal{D}_n}\Big| \to 0,$$

  hence

$$\frac{1}{\mathcal{D}_n}\sum_{i\in\mathcal{V}_j}\exp(\beta_{0i}^n)\,\|\Delta\eta_{ij}^n\| \to 0. \tag{36}$$

Combining (35) and (36) yields

$$\frac{1}{\mathcal{D}_n}\sum_{j:\,|\mathcal{V}_j|=1}\sum_{i\in\mathcal{V}_j}\exp(\beta_{0i}^n)\big[\|\Delta\beta_{1ij}^n\| + \|\Delta\eta_{ij}^n\|\big] \to 0. \tag{37}$$

Next, consider indices $j$ with multi-element cells, *i.e.,* $|\mathcal{V}_j| > 1$.

- Fix $u, v \in [Nd]$, let $\alpha_1^{(uv)} = 2$ and $\alpha_2 = \mathbf{0}_q$. Then

$$\frac{1}{\mathcal{D}_n}\sum_{i\in\mathcal{V}_j}\exp(\beta_{0i}^n)\,|(\Delta\beta_{1ij}^n)^{(uv)}|^2 = \Big|\frac{S_{n,j,\alpha_1,\alpha_2}}{\mathcal{D}_n}\Big| \to 0,$$

  hence

$$\frac{1}{\mathcal{D}_n}\sum_{i\in\mathcal{V}_j}\exp(\beta_{0i}^n)\,\|\Delta\beta_{1ij}^n\|^2 \to 0. \tag{38}$$

- Fix $u \in [q]$, take $\alpha_1 = \mathbf{0}_{Nd\times Nd}$ and $\alpha_2^{(u)} = 2$. Then

$$\frac{1}{\mathcal{D}_n}\sum_{i\in\mathcal{V}_j}\exp(\beta_{0i}^n)\,|(\Delta\eta_{ij}^n)^{(u)}|^2 = \Big|\frac{S_{n,j,\alpha_1,\alpha_2}}{\mathcal{D}_n}\Big| \to 0,$$

  giving

$$\frac{1}{\mathcal{D}_n}\sum_{i\in\mathcal{V}_j}\exp(\beta_{0i}^n)\,\|\Delta\eta_{ij}^n\|^2 \to 0. \tag{39}$$

Combining these gives

$$\frac{1}{\mathcal{D}_n} \sum_{j: |\mathcal{V}_j|>1} \sum_{i \in \mathcal{V}_j} \exp(\beta_{0i}^n) \big[ \|\Delta\beta_{1ij}^n\| + \|\Delta\eta_{ij}^n\| \big] \to 0. \tag{40}$$

Finally, summing the terms in (34), (37), and (40) yields

$$\mathcal{D}_n / \mathcal{D}_n \to 0,$$

which implies $1 \to 0$, a contradiction. Therefore, our initial assumption was false: *at least one* ratio among $S_{n,j,\alpha_1,\alpha_2}/\mathcal{D}_n$ and $T_{n,j,\ell}/\mathcal{D}_n$ does *not* converge to 0 as $n \to \infty$.

**Step 3 — Application of Fatou's lemma.** In this step, we show that *all* normalized coefficients $S_{n,j,\alpha_1,\alpha_2}/\mathcal{D}_n$ and $T_{n,j,\ell}/\mathcal{D}_n$ must, in fact, converge to zero as $n \to \infty$. This will contradict the conclusion of Step 2, completing the proof by contradiction. We denote $m_n$ as the maximum of the absolute values of those ratios. The result of Step 2 implies that $1/m_n \nrightarrow \infty$.

From the hypothesis in Equation (29), we have $\|g_{G_n} - g_{G_*}\|_{L^2(\mu)}/\mathcal{D}_n \to 0$ as $n \to \infty$, which indicates that $\|g_{G_n} - g_{G_*}\|_{L^1(\mu)}/\mathcal{D}_n \to 0$. Applying Fatou's lemma, we get

$$0 = \lim_{n \to \infty} \frac{\|g_{G_n} - g_{G_*}\|_{L^1(\mu)}}{m_n \mathcal{D}_n} \geq \int \liminf_{n \to \infty} \frac{|g_{G_n}(\boldsymbol{X}) - g_{G_*}(\boldsymbol{X})|}{m_n \mathcal{D}_n} \mathrm{d}\mu(\boldsymbol{X}) \geq 0.$$

This implies that $\frac{1}{m_n \mathcal{D}_n} \cdot [g_{G_n}(\boldsymbol{X}) - g_{G_*}(\boldsymbol{X})] \to 0$ as $n \to \infty$ for $\mu$-almost surely $\boldsymbol{X}$. Looking at the formulation of $Q_n(\boldsymbol{X})$ in equation (31), since the term $\Big[ \sum_{i'=1}^{k_0} \exp(\boldsymbol{X}^\top B_{i'}^0 \boldsymbol{X} + c_{i'}^0) + \sum_{j'=1}^{k_*} \exp((\beta_{1j'}^*)^\top W^\top \boldsymbol{X} + \beta_{0j'}^*) \Big]$ is bounded, we deduce that the term $\frac{Q_n(\boldsymbol{X})}{m_n \mathcal{D}_n} \to 0$ for $\mu$-almost surely $\boldsymbol{X}$.

Let us define the normalized limits

$$\frac{S_{n,j,\alpha_1,\alpha_2}}{m_n \mathcal{D}_n} \to \phi_{j,\alpha_1,\alpha_2}, \qquad \frac{T_{n,j,\ell}}{m_n \mathcal{D}_n} \to \varphi_{j,\ell},$$

with a note that at least one among them is non-zero. Then, from the decomposition of $Q_n(\boldsymbol{X})$ in Equation (33), we have

$$\sum_{j=1}^{N_p} \sum_{|\alpha_1|+|\alpha_2|=0}^{1+\mathbf{1}_{\{|\mathcal{V}_j|>1\}}} \phi_{j,\alpha_1,\alpha_2} \cdot \frac{\partial^{|\alpha_1|}E}{\partial\beta_1^{\alpha_1}}(\boldsymbol{X};\beta_{1j}^*) \frac{\partial^{|\alpha_2|}h}{\partial\eta^{\alpha_2}}(\boldsymbol{X};\eta_j^*),$$

$$-\sum_{j=1}^{N_p} \sum_{|\ell|=0}^{1+\mathbf{1}_{\{|\mathcal{V}_j|>1\}}} \varphi_{j,\ell} \cdot \frac{\partial^{|\ell|}E}{\partial\beta_1^\ell}(\boldsymbol{X};\beta_{1j}^*) g_{G_*}(\boldsymbol{X}) = 0,$$

for $\mu$-almost surely $\boldsymbol{X}$. It is worth noting that the term $\frac{\partial^{|\alpha_1|}E}{\partial\beta_1^{\alpha_1}}(\boldsymbol{X};\beta_{1j}^*) \cdot \frac{\partial^{|\alpha_2|}h}{\partial\eta^{\alpha_2}}(\boldsymbol{X};\eta_j^*)$ can be explicitly expressed as

- When $|\alpha_1| = 0, |\alpha_2| = 0$: $\exp((\beta_{1j}^*)^\top W^\top \boldsymbol{X}) h(\boldsymbol{X};\eta_j^*)$;

- When $|\alpha_1| = 1, |\alpha_2| = 0$: $(W^\top \boldsymbol{X})^{(u)} \exp((\beta_{1j}^*)^\top \boldsymbol{X}) h(\boldsymbol{X};\eta_j^*)$;

- When $|\alpha_1| = 0, |\alpha_2| = 1$: $\exp((\beta_{1j}^*)^\top W^\top \boldsymbol{X}) \frac{\partial h}{\partial\eta^{(w)}}(\boldsymbol{X};\eta_j^*)$;

- When $|\alpha_1| = 1, |\alpha_2| = 1$: $(W^\top \boldsymbol{X})^{(u)} \exp((\beta_{1j}^*)^\top W^\top \boldsymbol{X}) \frac{\partial h}{\partial\eta^{(w)}}(\boldsymbol{X};\eta_j^*)$;

- When $|\alpha_1| = 2, |\alpha_2| = 0$: $(W^\top \boldsymbol{X})^{(u)}(W^\top \boldsymbol{X})^{(v)} \exp((\beta_{1j}^*)^\top \boldsymbol{X}) h(\boldsymbol{X};\eta_j^*)$;

- When $|\alpha_1| = 0, |\alpha_2| = 2$: $\exp((\beta_{1j}^*)^\top \boldsymbol{X}) \frac{\partial^2 h}{\partial\eta^{(w)}\partial\eta^{(w')}}(\boldsymbol{X};\eta_j^*)$.

Recall that the expert function $h$ satisfies the strong identifiability condition in Definition A.1. This condition implies that the set of functions

$$\left\{ (W^\top \boldsymbol{X})^\nu \frac{\partial^{|\gamma|}h}{\partial\eta^\gamma}(\boldsymbol{X},\eta_j^*) : j \in [N_p], \, 0 \leq |\nu| + |\gamma| \leq 2 \right\}$$

is linearly independent for almost every $\boldsymbol{X}$. Therefore, we obtain that $\phi_{j,\alpha_1,\alpha_2} = \varphi_{j,\ell} = 0$ for all $j \in [N_p]$, $0 \leq |\alpha_1| + |\alpha_2|, |\ell| \leq 1 + \mathbf{1}_{\{|\mathcal{V}_j|>1\}}$. This contradicts the fact that at least one of these coefficients is non-zero. This completes the proof by contradiction for the local part, thereby establishing the inequality in Equation (28).

*Global part.* From the local inequality (28), we can infer that there exists a constant $\varepsilon' > 0$ such that

$$\inf_{G \in \mathcal{G}_{N_p'}(\Theta):\mathcal{D}(G,G_*)\leq\varepsilon'} \frac{\|g_G - g_{G_*}\|}{\mathcal{D}(G,G_*)} > 0.$$

It remains to show that the inequality also holds for measures far from $G_*$:

$$\inf_{G \in \mathcal{G}_{N_p'}(\Theta):\mathcal{D}(G,G_*)>\varepsilon'} \frac{\|g_G - g_{G_*}\|}{\mathcal{D}(G,G_*)} > 0. \tag{41}$$

Suppose, for the sake of contradiction, that (41) is false. Then there exists a sequence of mixing measures $\{G_n'\} \subset \mathcal{G}_{N_p'}(\Theta)$ with $\mathcal{D}(G_n', G_*) > \varepsilon'$ for all $n$, such that

$$\lim_{n\to\infty} \frac{\|g_{G_n'} - g_{G_*}\|}{\mathcal{D}(G_n', G_*)} = 0,$$

which in turn implies that $\|g_{G_n'} - g_{G_*}\| \to 0$ as $n \to \infty$.

Since the parameter space $\Theta$ is compact, the sequence $\{G_n'\}$ admits a convergent subsequence with limit $G' \in \mathcal{G}_{N_p'}(\Theta)$. Moreover, because $\mathcal{D}(G_n', G_*) > \varepsilon'$, it follows that the limit satisfies $\mathcal{D}(G', G_*) \geq \varepsilon'$.

Applying Fatou's lemma, we obtain

$$0 = \lim_{n\to\infty} \|g_{G_n'} - g_{G_*}\|^2 \geq \int \liminf_{n\to\infty} |g_{G_n'}(\boldsymbol{X}) - g_{G_*}(\boldsymbol{X})|^2 \, d\mu(\boldsymbol{X}).$$

This implies that $g_{G'}(\boldsymbol{X}) = g_{G_*}(\boldsymbol{X})$ for $\mu$-almost every $\boldsymbol{X}$.

By an identifiability property of the MoE model (to be established at the end of this proof), this functional equality implies that the measures themselves must be identical, *i.e.,* $G' \equiv G_*$. This leads to $\mathcal{D}(G', G_*) = 0$, which contradicts our earlier conclusion that $\mathcal{D}(G', G_*) \geq \varepsilon' > 0$. This contradiction establishes (41) and completes the proof of Equation (26).

**Identifiable.** We now establish the identifiability of the MoE regression function $g_G$. Specifically, we show that if $g_G(\boldsymbol{X}) = g_{G_*}(\boldsymbol{X})$ for almost every $\boldsymbol{X}$, then the measures must be identical, *i.e.,* $G \equiv G_*$.

For clarity, define

$$\text{softmax}_G(u) := \frac{\exp(u)}{\sum_{i'=1}^{N} \exp(\boldsymbol{X}^\top B_{i'}^0 \boldsymbol{X} + c_{i'}^0) + \sum_{j'=1}^{N_p} \exp((\beta_{1j'})^\top W^\top \boldsymbol{X} + \beta_{0j'})},$$

$$\text{softmax}_{G_*}(u^*) := \frac{\exp(u^*)}{\sum_{i'=1}^{N} \exp(\boldsymbol{X}^\top B_{i'}^0 \boldsymbol{X} + c_{i'}^0) + \sum_{j'=1}^{N_p} \exp((\beta_{1j'}^*)^\top W^\top \boldsymbol{X} + \beta_{0j'}^*)},$$

with

$$u \in \{\boldsymbol{X}^\top B_{i'}^0 \boldsymbol{X} + c_{i'}^0, (\beta_{1j'})^\top W^\top \boldsymbol{X} + \beta_{0j'} : i' \in [N], j' \in [N_p']\},$$

$$u^* \in \{\boldsymbol{X}^\top B_{i'}^0 \boldsymbol{X} + c_{i'}^0, (\beta_{1j'}^*)^\top W^\top \boldsymbol{X} + \beta_{0j'}^* : i' \in [N], j' \in [N_p]\}.$$

Since $g_G(\boldsymbol{X}) = g_{G_*}(\boldsymbol{X})$ for almost every $\boldsymbol{X}$, we have

$$\sum_{i=1}^{N} \text{softmax}_G(\boldsymbol{X}^\top B_i \boldsymbol{X} + c_i^0) \cdot h(\boldsymbol{X}, \eta_i^0) + \sum_{j=1}^{N_p'} \text{softmax}_G\left((\beta_{1j})^\top \boldsymbol{X} + \beta_{0j}\right) \cdot h(\boldsymbol{X}, \eta_j)$$

$$= \sum_{i=1}^{N} \text{softmax}_{G_*}(\boldsymbol{X}^\top B_i \boldsymbol{X} + c_i^0) \cdot h(\boldsymbol{X}, \eta_i^0) + \sum_{j=1}^{N_p} \text{softmax}_{G_*}\left((\beta_{1j}^*)^\top \boldsymbol{X} + \beta_{0j}^*\right) \cdot h(\boldsymbol{X}, \eta_j^*).$$

$$\tag{42}$$

As the expert function $h$ satisfies the conditions in Definition A.1, the set $\{h(\boldsymbol{X}, \eta_i') : i \in [k']\}$, where $\eta_1', \ldots, \eta_{k'}'$ are distinct vectors for some $k' \in \mathbb{N}$, is linearly independent. If $N_p' \neq N_p$, then there exists some $i \in [N_p']$ such that $\eta_i \neq \eta_j^*$ for any $j \in [N_p]$. This implies that $\sum_{j=1}^{N_p'} \mathrm{softmax}_G\left((\beta_{1j})^\top \boldsymbol{X} + \beta_{0j}\right) \cdot h(\boldsymbol{X}, \eta_j) = 0$, which is a contradiction. Thus, we must have that $N_p = N_p'$. As a result,

$$\left\{ \mathrm{softmax}_G\left((\beta_{1j})^\top \boldsymbol{X} + \beta_{0j}\right) : j \in [N_p'] \right\} = \left\{ \mathrm{softmax}_{G_*}\left((\beta_{1j}^*)^\top \boldsymbol{X} + \beta_{0j}^*\right) : j \in [N_p] \right\},$$

for almost every $\boldsymbol{X}$. Without loss of generality,

$$\mathrm{softmax}_G((\beta_{1j})^\top \boldsymbol{X} + \beta_{0j}) = \mathrm{softmax}_{G_*}((\beta_{1j}^*)^\top \boldsymbol{X} + \beta_{0j}^*), \tag{43}$$

for each $j \in [N_p]$. Since the softmax function is invariant to translation, this forces $\beta_{1j} = \beta_{1j}^*$ and $\beta_{0j} = \beta_{0j}^* + v_0$ for some $v_0 \in \mathbb{R}$. Given the normalization $\beta_{0N_p'} = \beta_{0N_p} = 0$, we conclude $\beta_{0j} = \beta_{0j}^*$.

Substituting back, the equation becomes:

$$\sum_{j=1}^{N_p} \exp(\beta_{0j}) \exp((\beta_{1j})^\top \boldsymbol{X}) h(\boldsymbol{X}, \eta_j) = \sum_{j=1}^{N_p} \exp(\beta_{0j}^*) \exp((\beta_{1j}^*)^\top \boldsymbol{X}) h(\boldsymbol{X}, \eta_j^*).$$

for almost every $\boldsymbol{X}$.

Partition the index set $[N_p]$ into groups $P_1, \ldots, P_m$ (with $m \leq N_p$) such that within each group the weights $\exp(\beta_{0i})$ agree, and across groups they differ. Then the above equality reduces groupwise to

$$\sum_{i \in P_j} \exp(\beta_{0i}) \exp((\beta_{1i})^\top \boldsymbol{X}) h(\boldsymbol{X}, \eta_i) = \sum_{i \in P_j} \exp(\beta_{0i}^*) \exp((\beta_{1i}^*)^\top \boldsymbol{X}) h(\boldsymbol{X}, \eta_i^*),$$

for almost every $\boldsymbol{X}$. Since $\beta_{1i} = \beta_{1i}^*$ and $\beta_{0i} = \beta_{0i}^*$, this equality implies that $\{\eta_i : i \in P_j\} = \{\eta_i^* : i \in P_j\}$ for all $j$.

Therefore,

$$G = \sum_{j=1}^m \sum_{i \in P_j} \exp(\beta_{0i}) \delta_{(\beta_{1i}, \eta_i)} = \sum_{j=1}^m \sum_{i \in P_j} \exp(\beta_{0i}^*) \delta_{(\beta_{1i}^*, \eta_i^*)} = G_*,$$

which proves the identifiable property.

**Proof of Equation** (27)**.** We begin the proof by introducing some key notations. Let $\mathcal{R}_{N_p'}(\Theta)$ be the class of regression functions corresponding to the mixing measures in $\mathcal{G}_{N_p'}(\Theta)$:

$$\mathcal{R}_{N_p'}(\Theta) := \{g_G(\boldsymbol{X}) : G \in \mathcal{G}_{N_p'}(\Theta)\}.$$

For any $\delta > 0$, we define the $L^2(\mu)$ neighborhood around the true function $g_{G_*}(Y|\boldsymbol{X})$ intersected with $\mathcal{R}_{N_p'}(\Theta)$ as

$$\mathcal{R}_{N_p'}(\Theta, \delta) := \{g \in \mathcal{R}_{N_p'}(\Theta) : \|g - g_{G_*}\|_{L^2(\mu)} \leq \delta\}.$$

To bound the complexity of this function class, van de Geer (2000) introduced the functional

$$\mathcal{J}_B(\delta, \mathcal{R}_{N_p'}(\Theta, \delta)) := \int_{\delta^2/2^{13}}^\delta H_B^{1/2}\left(t, \mathcal{R}_{N_p'}(\Theta, t), \|\cdot\|_{L^2(\mu)}\right) \, \mathrm{d}t \vee \delta, \tag{44}$$

where $H_B(\cdot)$ denotes the bracketing entropy of $\mathcal{R}_{N_p'}(\Theta, t)$ with respect to the $L^2(\mu)$ norm, and $a \vee b$ means $\max\{a, b\}$. By adapting the proof techniques of Theorems 7.4 and 9.2 from van de Geer (2000), we can establish the following lemma:

**Lemma A.4.** *Suppose $\Psi(\delta) \geq \mathcal{J}_B(\delta, \mathcal{R}_{N_p'}(\Theta, \delta))$ and that $\Psi(\delta)/\delta^2$ is non-increasing in $\delta$. Then there exists a universal constant $c$ and a sequence $(\delta_n)$ satisfying $\sqrt{n}\, \delta_n^2 \geq c\, \Psi(\delta_n)$ such that*

$$\mathbb{P}\left(\|g_{\widehat{G}_n} - g_{G_*}\|_{L^2(\mu)} > \delta\right) \leq c \exp\left(-\frac{n\delta^2}{c^2}\right),$$

*for all $\delta \geq \delta_n$.*

Next, we show that if the expert functions are Lipschitz continuous, then

$$H_B(\varepsilon, \mathcal{R}_{N_p'}(\Theta), \|\cdot\|_{L^2(\mu)}) \lesssim \log(1/\varepsilon), \tag{45}$$

for any $0 < \varepsilon \leq 1/2$. To see this, note that for any $g_G \in \mathcal{R}_{N_p'}(\Theta)$ the boundedness of $h$ ensures $h(\boldsymbol{X}, \eta) \leq M$ for all $\boldsymbol{X}$, for some constant $M$.

Let $\tau \leq \varepsilon$ and consider a $\zeta$-cover $\{\pi_1, \ldots, \pi_{\bar{N}}\}$ of $\mathcal{R}_{N_p'}(\Theta)$ under the $L^\infty$ norm, where $\bar{N} := N(\zeta, \mathcal{R}_{N_p'}(\Theta), \|\cdot\|_{L^\infty})$. Construct brackets $[L_i(\boldsymbol{X}), U_i(\boldsymbol{X})]$ for each $i \in [\bar{N}]$ by

$$L_i(\boldsymbol{X}) := \max\{\pi_i(\boldsymbol{X}) - \zeta, 0\}, \quad U_i(\boldsymbol{X}) := \max\{\pi_i(\boldsymbol{X}) + \zeta, M\}.$$

By construction, $\mathcal{R}_{N_p'}(\Theta) \subseteq \bigcup_{i=1}^{\bar{N}}[L_i, U_i]$ and the width satisfies $U_i - L_i \leq \min\{2\zeta, M\}$. Thus,

$$\|U_i - L_i\|^2 = \int (U_i - L_i)^2 \, d\mu(\boldsymbol{X}) \leq 4\zeta^2,$$

so that $\|U_i - L_i\| \leq 2\zeta$. Hence, by the definition of bracketing entropy,

$$H_B(2\zeta, \mathcal{R}_{N_p'}(\Theta), \|\cdot\|) \leq \log N = \log N(\zeta, \mathcal{R}_{N_p'}(\Theta), \|\cdot\|_{L^\infty}). \tag{46}$$

Therefore, we need to provide an upper bound for the covering number $\bar{N}$. In particular, we denote $\Delta := \{(\beta_1, \beta_0) \in \mathbb{R}^{Nd \times Nd} \times \mathbb{R}^{Nd} \times \mathbb{R} : (\beta_1, \beta_0, \eta) \in \Theta\}$ and $\Omega := \{\eta \in \mathbb{R}^q : (\beta_1, \beta_0, \eta) \in \Theta\}$. Since $\Theta$ is a compact set, $\Delta$ and $\Omega$ are also compact. Therefore, we can find $\zeta$-covers $\Delta_\zeta$ and $\Omega_\zeta$ for $\Delta$ and $\Omega$, respectively. We can check that

$$|\Delta_\zeta| \leq \mathcal{O}_P(\tau^{-(Nd+1)N_p'}), \quad |\Omega_\zeta| \lesssim \mathcal{O}_P(\tau^{-qN_p'}).$$

For each mixing measure $G = \sum_{i=1}^{N_p'} \exp(\beta_{0i})\delta_{(\beta_{1i}, \eta_i)} \in \mathcal{G}_{N_p'}(\Theta)$, we consider other two mixing measures:

$$\check{G} := \sum_{i=1}^{N_p'} \exp(\beta_{0i})\delta_{(\beta_{1i}, \overline{\eta}_i)}, \qquad \overline{G} := \sum_{i=1}^{N_p'} \exp(\overline{\beta}_{0i})\delta_{(\overline{\beta}_{1i}, \overline{\eta}_i)}.$$

Here, $\overline{\eta}_i \in \Omega_\zeta$ such that $\overline{\eta}_i$ is the closest to $\eta_i$ in that set, while $(\overline{\beta}_{1i}, \overline{\beta}_{0i}) \in \Delta_\zeta$ is the closest to $(\beta_{1i}, \beta_{0i})$ in that set. From the above formulations, we get that

$$
\begin{aligned}
\|g_G - g_{\check{G}}\|_{L^\infty} &= \sup_{\boldsymbol{X} \in \mathcal{X}} \sum_{j=1}^{N_p'} \frac{\exp(\beta_{1j}^\top \boldsymbol{X} + \beta_{0j}) \cdot |h(\boldsymbol{X}, \eta_j) - h(\boldsymbol{X}, \overline{\eta}_j)|}{\sum_{i'=1}^{N} \exp(\boldsymbol{X}^\top B_{i'}^0 \boldsymbol{X} + c_{i'}^0) + \sum_{j'=1}^{N_p'} \exp(\beta_{1j'}^\top \boldsymbol{X} + \beta_{0j'})} \\
&\leq \sum_{j=1}^{N_p'} \sup_{\boldsymbol{X} \in \mathcal{X}} \frac{\exp(\beta_{1j}^\top \boldsymbol{X} + \beta_{0j}) \cdot |h(\boldsymbol{X}, \eta_j) - h(\boldsymbol{X}, \overline{\eta}_j)|}{\sum_{i'=1}^{N} \exp(\boldsymbol{X}^\top B_{i'}^0 \boldsymbol{X} + c_{i'}^0) + \sum_{j'=1}^{N_p'} \exp(\beta_{1j'}^\top \boldsymbol{X} + \beta_{0j'})} \\
&\leq \sum_{j=1}^{N_p'} \sup_{\boldsymbol{X} \in \mathcal{X}} |h(\boldsymbol{X}, \eta_j) - h(\boldsymbol{X}, \overline{\eta}_j)| \\
&\leq \sum_{j=1}^{N_p'} \sup_{\boldsymbol{X} \in \mathcal{X}} [L_1(\boldsymbol{X}) \cdot \|\eta_j - \overline{\eta}_j\|] \lesssim N_p'\zeta \lesssim \zeta.
\end{aligned}
$$

Here, the second inequality occurs as the softmax weight is bounded by one, and the third inequality follows from the fact that the expert $h(\boldsymbol{X}, \cdot)$ is a Lipschitz function with some Lipschitz constant $L_1(\boldsymbol{X}) > 0$. Next, let us denote

$$D := \sum_{i'=1}^{N} \exp(\boldsymbol{X}^\top B_{i'}^0 \boldsymbol{X} + c_{i'}^0) + \sum_{j'=1}^{N_p'} \exp(\beta_{1j'}^\top \boldsymbol{X} + \beta_{0j'}),$$

$$\overline{D} := \sum_{i'=1}^{N} \exp(\boldsymbol{X}^\top B_{i'}^0 \boldsymbol{X} + c_{i'}^0) + \sum_{j'=1}^{N_p'} \exp(\overline{\beta}_{1j'}^\top \boldsymbol{X} + \overline{\beta}_{0j'}).$$

Then, we have

$$
\|g_{\check{G}} - g_{\overline{G}}\|_{L^\infty} = \sup_{\boldsymbol{X} \in \mathcal{X}} \left| \frac{1}{D} \Big( \sum_{i=1}^{N} \exp(\boldsymbol{X}^\top B_i^0 \boldsymbol{X} + c_i^0) h(\boldsymbol{X}, \eta_i^0) + \sum_{j=1}^{N_p'} \exp(\beta_{1j}^\top \boldsymbol{X} + \beta_{0j}) h(\boldsymbol{X}, \overline{\eta}_j) \Big) \right.
$$

$$
\left. - \frac{1}{\overline{D}} \Big( \sum_{i=1}^{N} \exp(\boldsymbol{X}^\top B_i^0 \boldsymbol{X} + c_i^0) h(\boldsymbol{X}, \eta_i^0) + \sum_{j=1}^{N_p'} \exp(\overline{\beta}_{1j}^\top \boldsymbol{X} + \overline{\beta}_{0j}) h(\boldsymbol{X}, \overline{\eta}_j) \Big) \right|
$$

$$
\leq \left| \frac{1}{D} - \frac{1}{\overline{D}} \right| \cdot \sum_{i=1}^{N} \sup_{\boldsymbol{X} \in \mathcal{X}} \left| \exp(\boldsymbol{X}^\top B_i^0 \boldsymbol{X} + c_i^0) h(\boldsymbol{X}, \eta_i^0) \right|
$$

$$
+ \sum_{j=1}^{N_p'} \sup_{\boldsymbol{X} \in \mathcal{X}} \left| \frac{\exp(\beta_{1j}^\top \boldsymbol{X} + \beta_{0j})}{D} - \frac{\exp(\overline{\beta}_{1j}^\top \boldsymbol{X} + \overline{\beta}_{0j})}{\overline{D}} \right| \cdot |h(\boldsymbol{X}, \overline{\eta}_j)|. \tag{47}
$$

Now, we will bound two terms in the above right hand side. Firstly, since both the input space $\mathcal{X}$ and the parameter space $\Theta$ are bounded, we have that

$$
\frac{1}{D} - \frac{1}{\overline{D}} \lesssim |D - \overline{D}| \leq \sum_{j'=1}^{N_p'} \left| \exp(\beta_{1j'}^\top \boldsymbol{X} + \beta_{0j'}) - \exp(\overline{\beta}_{1j'}^\top \boldsymbol{X} + \overline{\beta}_{0j'}) \right|
$$

$$
\lesssim \sum_{j'=1}^{N_p'} \left| (\beta_{1j} - \overline{\beta}_{1j})^\top \boldsymbol{X} + (\beta_{0j} - \overline{\beta}_{0j}) \right|
$$

$$
\leq \sum_{j'=1}^{N_p'} |(\beta_{1j} - \overline{\beta}_{1j})^\top \boldsymbol{X}| + |\beta_{0j} - \overline{\beta}_{0j}|
$$

$$
\lesssim \sum_{j=1}^{N_p'} \Big[ \|\beta_{1j} - \overline{\beta}_{1j}\| \cdot \|\boldsymbol{X}\| + |\beta_{0j} - \overline{\beta}_{0j}| \Big]
$$

$$
\leq N_p'(B + 1)\zeta \lesssim \zeta.
$$

As a result, we deduce that

$$
\left| \frac{1}{D} - \frac{1}{\overline{D}} \right| \cdot \sum_{i=1}^{N} \sup_{\boldsymbol{X} \in \mathcal{X}} \left| \exp(\boldsymbol{X}^\top B_i^0 \boldsymbol{X} + c_i^0) h(\boldsymbol{X}, \eta_i^0) \right| \lesssim \zeta. \tag{48}
$$

Regarding the second term, note that

$$
\frac{\exp(\beta_{1j}^\top \boldsymbol{X} + \beta_{0j})}{D} - \frac{\exp(\overline{\beta}_{1j}^\top \boldsymbol{X} + \overline{\beta}_{0j})}{\overline{D}}
$$

$$
= \exp(\beta_{1j}^\top \boldsymbol{X} + \beta_{0j}) \Big( \frac{1}{D} - \frac{1}{\overline{D}} \Big) + \frac{1}{\overline{D}} \Big[ \exp(\beta_{1j}^\top \boldsymbol{X} + \beta_{0j}) - \exp(\overline{\beta}_{1j}^\top \boldsymbol{X} + \overline{\beta}_{0j}) \Big].
$$

Since both the input space and the parameter space are bounded, we have

$$
\exp(\beta_{1j}^\top \boldsymbol{X} + \beta_{0j}) \Big( \frac{1}{D} - \frac{1}{\overline{D}} \Big) \lesssim \frac{1}{D} - \frac{1}{\overline{D}} \lesssim \zeta,
$$

$$
\frac{1}{\overline{D}} \Big[ \exp(\beta_{1j}^\top \boldsymbol{X} + \beta_{0j}) - \exp(\overline{\beta}_{1j}^\top \boldsymbol{X} + \overline{\beta}_{0j}) \lesssim (B + 1)\zeta \lesssim \zeta,
$$

which yields that

$$
\sum_{j=1}^{N_p'} \sup_{\boldsymbol{X} \in \mathcal{X}} \left| \frac{\exp(\beta_{1j}^\top \boldsymbol{X} + \beta_{0j})}{D} - \frac{\exp(\overline{\beta}_{1j}^\top \boldsymbol{X} + \overline{\beta}_{0j})}{\overline{D}} \right| \cdot |h(\boldsymbol{X}, \overline{\eta}_j)|
$$

$$
\lesssim \zeta \sum_{j=1}^{N_p'} \sup_{\boldsymbol{X} \in \mathcal{X}} |h(\boldsymbol{X}, \overline{\eta}_j)| \lesssim \zeta. \tag{49}
$$

From Equations (47), (48) and (49), we obtain that $\|g_{\check{G}} - g_{\overline{G}}\|_{L^\infty} \lesssim \zeta$. According to the triangle inequality, we have

$$\|g_G - g_{\overline{G}}\|_{L^\infty} \leq \|g_G - g_{\check{G}}\|_{L^\infty} + \|g_{\check{G}} - g_{\overline{G}}\|_{L^\infty} \lesssim \zeta.$$

By definition of the covering number, we deduce that

$$N(\zeta, \mathcal{R}_{N'_p}(\Theta), \|\cdot\|_{L^\infty}) \leq |\Delta_\zeta| \times |\Omega_\zeta| \leq \mathcal{O}(n^{-(Nd+1)N'_p}) \times \mathcal{O}(n^{-qN'_p}) \leq \mathcal{O}(n^{-(Nd+1+q)N'_p}). \tag{50}$$

Combined with Equations (46) and (50), we achieve the following result:

$$H_B\left(2\zeta, \mathcal{R}_{N'_p}(\Theta), \|\cdot\|_{L^2(\mu)}\right) \lesssim \log(1/\tau).$$

Let $\zeta = \varepsilon/2$, then we obtain that

$$H_B(\varepsilon, \mathcal{R}_{N'_p}(\Theta), \|\cdot\|_{L^2(\mu)}) \lesssim \log(1/\varepsilon).$$

As a result, it follows that

$$\mathcal{J}_B(\delta, \mathcal{R}_{N'_p}(\Theta, \delta)) = \int_{\delta^2/2^{13}}^{\delta} H_B^{1/2}(t, \mathcal{R}_{N'_p}(\Theta, t), \|\cdot\|_{L^2(\mu)})\, \mathrm{d}t \vee \delta \lesssim \int_{\delta^2/2^{13}}^{\delta} \log(1/t) dt \vee \delta. \tag{51}$$

Let $\Psi(\delta) = \delta \cdot [\log(1/\delta)]^{1/2}$, then $\Psi(\delta)/\delta^2$ is a non-increasing function of $\delta$. Furthermore, Equation (51) indicates that $\Psi(\delta) \geq \mathcal{J}_B(\delta, \mathcal{R}_{N'_p}(\Theta, \delta))$. In addition, let $\delta_n = \sqrt{\log(n)/n}$, then we get that $\sqrt{n}\delta_n^2 \geq c\Psi(\delta_n)$ for some universal constant $c$. Finally, by applying Lemma A.4, we achieve the desired conclusion of the theorem.

# B  TRAINING ALGORITHM OF SMOPE

---

**Algorithm 1** Training Process for Task $\mathcal{D}_t$

---

**Input:** Pre-trained ViT $f_\theta$, dataset $\mathcal{D}_t$, number of epochs $E$, hyperparameters $\epsilon$, $\alpha_{\text{router}}$, $\alpha_{\text{proto}}$, and accumulated expert activation frequencies $\{F_{j'}\}$
**Output:** Updated prompt parameters $\mathbf{P}$ and classifier parameters $\phi$

1: **if** $t > 1$ **then**
2:     Store the prompt parameters from the previous task: $\mathbf{P}_{\text{old}} \leftarrow \mathbf{P}$
3: **if** $t = 1$ **then**
4:     **for** $epoch = 1, \ldots, E/2$ **do**
5:         **for** $(\boldsymbol{x}_i^t, y_i^t) \in \mathcal{D}_t$ **do**
6:             Compute output $\hat{y} = g_\phi(f_\theta(\boldsymbol{x}_i^t; \mathbf{P}, K = -1))$
7:             Optimize $\mathbf{P}$ and $\phi$ with $\mathcal{L}_{\text{ce}}$            ▷ Dense training phase
8: **for** $epoch = 1, \ldots, E$ **do**
9:     **for** $(\boldsymbol{x}_i^t, y_i^t) \in \mathcal{D}_t$ **do**
10:         Compute output $\hat{y} = g_\phi(f_\theta(\boldsymbol{x}_i^t; \mathbf{P}, K))$
11:         Optimize $\mathbf{P}$ and $\phi$ using Equation (16)       ▷ Sparse training phase
12: **for** $c \in \mathcal{Y}^t$ **do**
13:     Estimate Gaussian distribution $\mathcal{G}_c$ using Equation (53)
14: **for** $epoch = 1, \ldots, E$ **do**
15:     Optimize $\phi$ with the $\mathcal{L}_{\text{tap}}$ from Equation (54)       ▷ Task-adaptive prediction
16: **for** $(\boldsymbol{x}_i^t, y_i^t) \in \mathcal{D}_t$ **do**
17:     Compute $\boldsymbol{z} = f_\theta(\boldsymbol{x}_i^t; \mathbf{P}, K)$
18:     Update accumulated frequencies of expert usage $\{F_{j'}\}$
    **return** $(\mathbf{P}, \phi)$

---

For the $l$-th MSA block, let $\boldsymbol{P}_{(l)}^K$ and $\boldsymbol{P}_{(l)}^V$ denote the prefix parameters where prompts are inserted. For clarity, the main text omits the layer index $(l)$ in the notation. The complete set of prompt

parameters is $\mathbf{P} = \{(\boldsymbol{P}_{(l)}^K, \ \boldsymbol{P}_{(l)}^V)\}_{l=1}^L$, where $L$ is the number of MSA blocks in the ViT. Given an input $\boldsymbol{x}$, its prompted representation is $\boldsymbol{z} = f_\theta(\boldsymbol{x}; \mathbf{P}, K)$, where $\theta$ denotes the frozen pre-trained ViT weights and $K$ is the number of selected prompt experts. Setting $K = -1$ activates a dense prompting mode that bypasses sparse expert selection. Using an MLP classifier $g_\phi$, the prediction is given by:

$$\hat{y} = g_\phi(\boldsymbol{z}) = g_\phi(f_\theta(\boldsymbol{x}; \ \mathbf{P}, K)). \tag{52}$$

During training, only the prompt parameters $\mathbf{P}$ and the classifier parameters $\phi$ are updated.

Following prior work on SMoE training (Wu et al., 2022; Lin et al., 2024; Cai et al., 2025), we first adopt dense training ($K = -1$) for several initial epochs on the first task. This provides a good initialization of the prompt parameters before enabling sparse expert selection.

For each task $t$, the model is optimized on $\mathcal{D}_t$ using the objective in Equation (16). We then refine the classifier parameters $\phi$ using the task-adaptive prediction (TAP) objective, following (Wang et al., 2023; Jiao et al., 2024; Le et al., 2024a), which mitigates classifier bias by modeling the Gaussian distributions of previously seen classes. Specifically, for each class $c \in \mathcal{Y}^i$ from tasks $i = 1, \ldots, t-1$, we maintain a Gaussian distribution $\mathcal{G}_c = \mathcal{N}(\boldsymbol{\mu}_c, \boldsymbol{\Sigma}_c)$, approximated from the prompted representations $\mathcal{D}_c^z = \{\boldsymbol{z} = f_\theta(\boldsymbol{x}; \mathbf{P}, K) \mid (\boldsymbol{x}, c) \in \mathcal{D}_i\}$. The distribution parameters are estimated as:

$$\boldsymbol{\mu}_c = \sum_{\boldsymbol{z}_c \in \mathcal{D}_c^z} \frac{\boldsymbol{z}_c}{|\mathcal{D}_c^z|}, \quad \boldsymbol{\Sigma}_c = \sum_{\boldsymbol{z}_c \in \mathcal{D}_c^z} \frac{(\boldsymbol{z}_c - \boldsymbol{\mu}_c)(\boldsymbol{z}_c - \boldsymbol{\mu}_c)^\top}{|\mathcal{D}_c^z|}. \tag{53}$$

The classifier $g_\phi$ is further optimized with the TAP loss:

$$\mathcal{L}_{\text{tap}} = \sum_{i=1}^{t} \sum_{c \in \mathcal{Y}^i} \sum_{\boldsymbol{z} \in \mathcal{Z}_{i,c}} -\log\left(\frac{\exp(g_\phi(\boldsymbol{z})[c])}{\sum_{j=1}^{t} \sum_{c' \in \mathcal{Y}^j} \exp(g_\phi(\boldsymbol{z})[c'])}\right) \tag{54}$$

where $\mathcal{Z}_{i,c}$ is constructed by sampling an equal number of pseudo-representations from $\mathcal{G}_c$ for each class $c \in \mathcal{Y}^i$. The full procedure is summarized in Algorithm 1.

## C  ADDITIONAL EXPERIMENTAL DETAILS

**Evaluation Metrics.** We report two standard metrics, Final Average Accuracy (FAA) and Cumulative Average Accuracy (CAA), as they inherently capture both plasticity and forgetting (Smith et al., 2023). FAA measures the overall performance by computing the average accuracy across all $T$ tasks after the completion of continual learning. CAA extends this by averaging the performance across all intermediate stages. Formally, let $S_{i,t}$ denote the accuracy on the $i$-th task after learning the $t$-th task, and define the average accuracy after $t$ tasks as $A_t = \frac{1}{t} \sum_{i=1}^{t} S_{i,t}$. Then, after all $T$ tasks are learned, FAA and CAA are given by FAA $= A_T$, and CAA $= \frac{1}{T} \sum_{t=1}^{T} A_t$.

**Data Augmentation.** Input images are resized to $224 \times 224$ and augmented following the protocol of Smith et al. (2023), including random horizontal flipping and standard normalization.

**Implementation Details.** We use a pre-trained ViT-B/16 model as the backbone. Training is conducted using the AdamW optimizer with hyperparameters $\beta_1 = 0.9$ and $\beta_2 = 0.999$. Following the setup in VQ-Prompt (Jiao et al., 2024), 20% of the training data is reserved for validation, and a grid search is used to tune hyperparameters. We evaluate L2P++, Deep L2P++, DualPrompt, CODA-Prompt, and VQ-Prompt using the official implementation provided by VQ-Prompt. For OVOR, we similarly employ the original codebase. Recent work by Feng et al. (2025) has identified an implementation issue in the official HiDe-Prompt codebase, specifically in its prompt retrieval mechanism. This issue introduces information leakage, resulting in overestimated performance, particularly when inference is performed with a batch size of 1, while the training batch size remains consistent with the original setup. To address this, we adopt the corrected prompt retrieval code provided by Feng et al. (2025) for both HiDe-Prompt and NoRGa, while preserving all other components from the original implementations.

**Hyperparameters.** For SMoPE, we fix the prompt length to $N_p = 25$ and select $K = 5$ prompt experts in all experiments. Prompts are inserted into the first six MSA blocks (see Table 5). The

Table 5: Detailed prompt configurations used for our main experiments. "Location" indicates the specific MSA block where the prompt is inserted. Here, $N$ denotes the number of prompts, and $N_p$ denotes the prompt length.

| Method | Location | Dataset | Hyperparameters |
|---|---|---|---|
| L2P++ | [1] | All | $N = 30, N_p = 20$ |
| Deep L2P++ | [1 2 3 4 5] | All | $N = 30, N_p = 20$ |
| DualPrompt | [1 2] | All | G: $N = 1, N_p = 5$ |
| | [3 4 5] | All | E: $N = 10, N_p = 20$ |
| CODA-Prompt | [1 2 3 4 5] | All | $N = 100, N_p = 8$ |
| OVOR | [1 2 3] | All | G: $N = 1, N_p = 5$ |
| | [4 5] | All | E: $N = 1, N_p = 20$ |
| HiDe-Prompt | [1 2 3 4 5] | ImageNet-R | $N = 10, N_p = 40$ |
| | | CIFAR-100 | $N = 10, N_p = 10$ |
| | | CUB-200 | $N = 10, N_p = 40$ |
| NoRGa | [1 2 3 4 5] | ImageNet-R | $N = 10, N_p = 40$ |
| | | CIFAR-100 | $N = 10, N_p = 10$ |
| | | CUB-200 | $N = 10, N_p = 40$ |
| VQ-Prompt | [1 2 3 4 5] | All | $N = 10, N_p = 8$ |
| SMoPE | [1 2 3 4 5 6] | All | $N = 1, N_p = 25$ |

Table 6: Performance comparison on ImageNet-R with varying task numbers. ↑ indicates higher is better. FAA and CAA are averaged over 5 runs. **Bold** denotes the best results, excluding joint training.

| Method | 5-task | | 10-task | | 20-task | | 50-task | |
|---|---|---|---|---|---|---|---|---|
| | FAA (↑) | CAA (↑) | FAA (↑) | CAA (↑) | FAA (↑) | CAA (↑) | FAA (↑) | CAA (↑) |
| Joint-Train | 82.06 | | 82.06 | | 82.06 | | 82.06 | |
| L2P++ | $70.83 \pm 0.58$ | $78.34 \pm 0.47$ | $69.29 \pm 0.73$ | $78.30 \pm 0.69$ | $65.89 \pm 1.30$ | $77.15 \pm 0.65$ | $55.65 \pm 0.89$ | $64.88 \pm 0.86$ |
| Deep L2P++ | $73.93 \pm 0.37$ | $80.14 \pm 0.54$ | $71.66 \pm 0.64$ | $79.63 \pm 0.90$ | $68.42 \pm 1.20$ | $78.68 \pm 1.03$ | $58.48 \pm 0.54$ | $67.04 \pm 0.80$ |
| DualPrompt | $73.05 \pm 0.50$ | $79.47 \pm 0.40$ | $71.32 \pm 0.62$ | $78.94 \pm 0.72$ | $67.89 \pm 1.39$ | $77.42 \pm 0.80$ | $58.31 \pm 1.18$ | $65.51 \pm 0.95$ |
| CODA-Prompt | $76.51 \pm 0.38$ | $82.04 \pm 0.54$ | $75.45 \pm 0.56$ | $81.59 \pm 0.82$ | $72.37 \pm 1.19$ | $79.88 \pm 1.06$ | $66.96 \pm 0.71$ | $75.19 \pm 0.72$ |
| OVOR | $75.81 \pm 0.16$ | $79.31 \pm 0.42$ | $75.25 \pm 0.21$ | $79.78 \pm 0.65$ | $72.45 \pm 0.42$ | $77.60 \pm 1.15$ | $65.84 \pm 0.59$ | $74.84 \pm 0.69$ |
| HiDe-Prompt | $75.00 \pm 0.05$ | $79.68 \pm 0.43$ | $74.25 \pm 0.19$ | $79.64 \pm 0.53$ | $73.71 \pm 0.37$ | $79.14 \pm 0.72$ | $70.82 \pm 0.54$ | $76.81 \pm 0.83$ |
| NoRGa | $75.17 \pm 0.19$ | $79.64 \pm 0.62$ | $74.39 \pm 0.08$ | $79.68 \pm 0.41$ | $73.89 \pm 0.33$ | $79.22 \pm 0.84$ | $71.02 \pm 0.63$ | $76.99 \pm 0.58$ |
| VQ-Prompt | $79.23 \pm 0.29$ | $82.96 \pm 0.50$ | $78.71 \pm 0.22$ | $83.24 \pm 0.68$ | **78.10** $\pm 0.22$ | $82.70 \pm 1.16$ | $75.31 \pm 0.61$ | $81.52 \pm 0.81$ |
| **SMoPE** | **80.09** $\pm 0.42$ | **84.47** $\pm 0.95$ | **79.32** $\pm 0.42$ | **84.39** $\pm 0.77$ | $77.81 \pm 0.36$ | **83.38** $\pm 0.98$ | **75.54** $\pm 0.49$ | **81.94** $\pm 0.93$ |

hyperparameter $\epsilon$ is tuned over the set $\{0.0, 0.1, \ldots, 0.9, 1.0\}$. The weighting coefficients $\alpha_{\text{router}}$ and $\alpha_{\text{proto}}$ are tuned over the set $\{5e^{-6}, 1e^{-5}, 5e^{-5}, 1e^{-4}, 5e^{-4}, 1e^{-3}\}$. For all baseline methods, we use the configurations and hyperparameters reported in their respective original papers to ensure a fair comparison.

# D    ADDITIONAL EXPERIMENTAL RESULTS

## D.1    PERFORMANCE ACROSS VARYING TASK LENGTHS

To assess the scalability and generalizability of SMoPE, we follow prior work (Huang et al., 2024; Jiao et al., 2024) by partitioning ImageNet-R into 5, 10, and 20 sequential tasks, ensuring a fair comparison with existing methods. Additionally, we introduce a 50-task split to more rigorously evaluate SMoPE's capacity to handle longer task sequences. The results are shown in Table 6. Across all configurations, SMoPE consistently outperforms task-specific methods such as HiDe-Prompt and NoRGa, despite using a single shared prompt. Its performance remains comparable to VQ-Prompt, further validating its robustness under varying task lengths. These findings suggest that SMoPE can generalize effectively as the number of tasks increases. However, relying on a single prompt for an indefinite number of tasks may eventually lead to knowledge interference as the task sequence grows. To address this limitation, future work could explore strategies for scaling the number of prompt experts to better support longer task sequences.

## D.2    PERFORMANCE ACROSS VARYING PROMPT LENGTHS AND NUMBER OF PROMPT EXPERTS

We investigated the impact of prompt length $N_p$ and the number of selected experts $K$ on model performance, with results presented in Figure 3. Across a wide range of configurations, our method

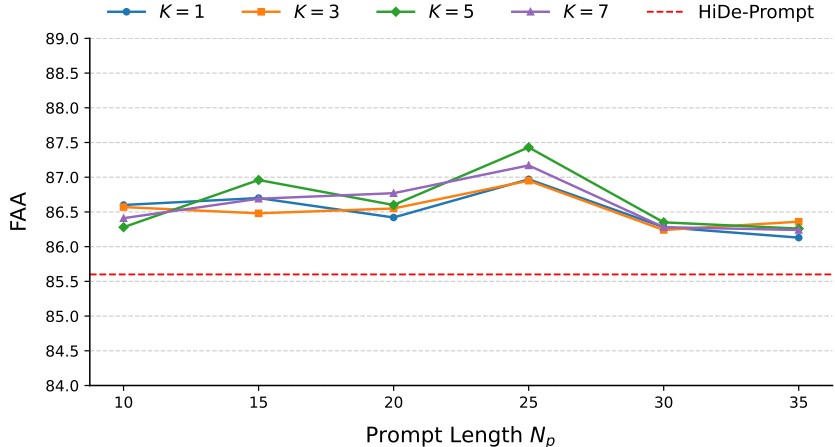

Figure 3: **Impact of Prompt Length $N_p$ and Number of Selected Experts $K$ on Performance.** Performance across different combinations of $N_p$ and $K$ values on CUB-200 with a 10-task split.

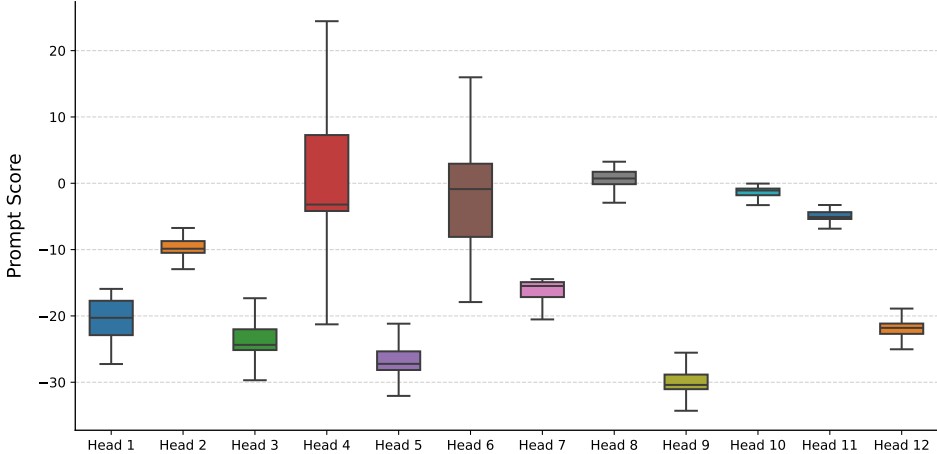

Figure 4: **Distribution of Prompt Expert Scores.** Box plot illustrating the distribution of prompt expert scores $\tilde{s}_{j'}$ across attention heads in the first MSA block on the CUB-200 dataset.

consistently surpasses task-specific prompting approaches, particularly HiDe-Prompt, demonstrating the robustness of SMoPE. Performance tends to degrade when $N_p$ is small, as indicated by lower FAA scores across various values of $K$. Increasing $N_p$ generally improves performance up to a certain threshold. However, excessively large values of $N_p$ lead to a decline in performance, likely due to the increased difficulty of effectively training all $N_p$ prompt experts. A larger number of experts also means that each receives less training data, which hinders learning and complicates the selection of relevant experts, ultimately reducing FAA scores. Regarding the number of selected experts, we find that setting $K = 5$ yields strong and stable performance across different prompt lengths. Based on these observations, we use $N_p = 25$ and $K = 5$ in all our experiments.

### D.3 DETAILED ANALYSIS OF ADAPTIVE NOISE

**Adaptive Noise Formulation.** Our adaptive noise formulation is derived directly from min–max normalization. From Equation (12), the selected expert set can be expressed as:

$$K_{\boldsymbol{X}} = \underset{S}{arg\,max} \sum_{j' \in S} \frac{\tilde{s}_{j'}(\boldsymbol{X}) - \epsilon_{j'}}{\max_j \tilde{s}_j(\boldsymbol{X}) - \min_j \tilde{s}_j(\boldsymbol{X})} = \underset{S}{arg\,max} \sum_{j' \in S} \frac{\tilde{s}_{j'}(\boldsymbol{X}) - \epsilon_{j'} - \min_j \tilde{s}_j(\boldsymbol{X})}{\max_j \tilde{s}_j(\boldsymbol{X}) - \min_j \tilde{s}_j(\boldsymbol{X})}$$

$$= \underset{S}{arg\,max} \sum_{j' \in S} \left( \frac{\tilde{s}_{j'}(\boldsymbol{X}) - \min_j \tilde{s}_j(\boldsymbol{X})}{\max_j \tilde{s}_j(\boldsymbol{X}) - \min_j \tilde{s}_j(\boldsymbol{X})} - \frac{\epsilon_{j'}}{\max_j \tilde{s}_j(\boldsymbol{X}) - \min_j \tilde{s}_j(\boldsymbol{X})} \right).$$

Table 7: Performance comparison on ImageNet-R and CUB-200 using 10-task split with different noise formulations. ↑ indicates that higher values are better. **Bold** highlights the best results.

| Method | ImageNet-R | | CUB-200 | |
|---|---|---|---|---|
| | FAA (↑) | CAA (↑) | FAA (↑) | CAA (↑) |
| Fixed Noise | $79.03 \pm 0.31$ | $\mathbf{84.42} \pm 0.68$ | $86.86 \pm 0.47$ | $90.37 \pm 0.60$ |
| Uniform Noise | $79.04 \pm 0.63$ | $84.29 \pm 0.70$ | $87.14 \pm 0.45$ | $90.40 \pm 0.65$ |
| Adaptive Noise | $\mathbf{79.32} \pm 0.42$ | $84.39 \pm 0.77$ | $\mathbf{87.43} \pm 0.39$ | $\mathbf{91.11} \pm 0.55$ |

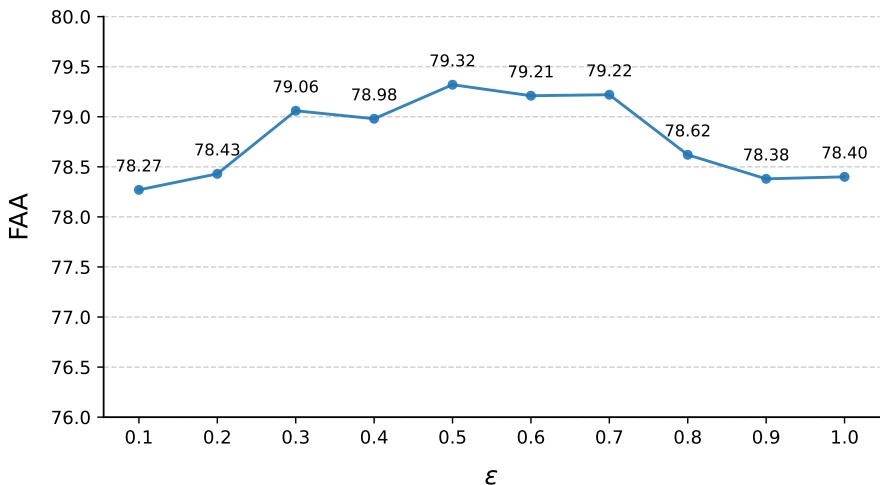

Figure 5: **Impact of $\epsilon$ on Performance.** Performance across different $\epsilon$ values on the ImageNet-R dataset using a 10-task split.

Thus, our adaptive noise can be interpreted as performing min–max normalization of the expert scores, followed by subtracting a scaled $\epsilon$ from the scores of important experts. As a result, the noise is adaptively scaled to the dynamic range of the scores, eliminating the need for manual tuning of noise magnitudes across different attention heads.

Previous work has typically encouraged expert exploration by injecting random noise into expert scores, either with a fixed magnitude (Nguyen et al., 2024) or by sampling from predefined distributions (*e.g.,* Uniform) (Shazeer et al., 2017; Fedus et al., 2022). However, in pre-trained models, the range of expert scores can vary considerably across attention heads (see Figure 4), making it difficult to select an appropriate fixed noise level. In contrast, our method requires tuning only a single noise parameter, $\epsilon$, which is constrained to the interval $[0, 1]$. This approach eliminates the need for head-specific noise scaling and significantly reduces the complexity of hyperparameter tuning in large-scale models.

To assess the effectiveness of our adaptive noise approach, we compare it with the following baseline variants:

- *Fixed Noise*: A constant noise value $\epsilon$ is subtracted from the scores of frequently activated prompt experts across all attention heads.
- *Uniform Noise*: Noise is sampled from a uniform distribution $\mathcal{U}(-\epsilon, \epsilon)$ and added to all prompt expert scores across all attention heads.

The experimental results, summarized in Table 7, demonstrate that our adaptive noise formulation surpasses these baselines, highlighting its effectiveness. While our noise formulation is specifically tailored for continual learning to help preserve knowledge in important experts, it may also be applicable to other settings, which we leave for future exploration.

**Impact of $\epsilon$ on Performance.** To evaluate the impact of $\epsilon$ on model performance, we report FAA results on ImageNet-R across a range of $\epsilon$ values, as shown in Figure 5. When $\epsilon = 0.0$, performance

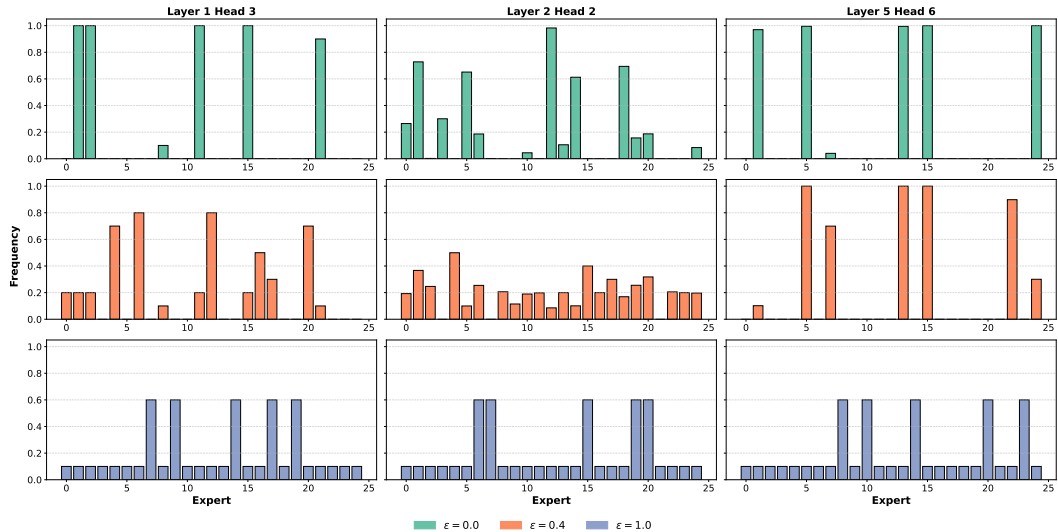

Figure 6: **Impact of $\epsilon$ on Activation Frequencies.** Results on CUB-200 with a prompt length of $N_p = 25$ and $K = 5$. We present the results for several representative attention heads and visualize the frequency with which prompt experts are activated after training on all tasks for different values of $\epsilon$.

is suboptimal due to imbalanced expert utilization. As $\epsilon$ increases, performance improves, suggesting enhanced exploration and more balanced expert engagement. However, excessively large values of $\epsilon$ can reduce expert specialization, as a broader set of experts is activated more frequently. This also hinders the reuse of frequently activated trained experts, limiting knowledge transfer. Overall, a moderate $\epsilon$ value achieves a favorable trade-off between exploration and stability, resulting in optimal performance.

**Impact of $\epsilon$ on Activation Frequencies of Prompt Experts.** To investigate the influence of the regularization parameter $\epsilon$ on the activation frequencies of prompt experts, we select several representative attention heads and visualize their activation patterns across varying values of $\epsilon$ using the CUB-200 dataset. The results are shown in Figure 6.

At $\epsilon = 0.0$, we observe that certain attention heads activate only a limited subset of prompt experts. However, as the input distribution to these attention heads shifts with the introduction of new tasks, different sets of experts may be activated. This variation is primarily driven by the relevance of each expert to the input, as determined by the score function. Interestingly, even in the absence of adaptive noise, some attention heads (e.g., Layer 2, Head 2, as depicted in the figure) demonstrate balanced expert utilization, with no significant imbalances in activation frequency. At $\epsilon = 1.0$, the regularization penalty reaches its maximum, substantially reducing the scores of frequently activated experts while encouraging the activation of underutilized ones. As shown in the figure, this penalty leads to a more balanced distribution of expert utilization across all attention heads. We find that $\epsilon = 0.4$ is the optimal value and yields the best performance. At this setting, the distribution of expert activation is relatively balanced, though some attention heads still exhibit imbalanced expert utilization. This highlights a key trade-off in continual learning: activating too many experts can dilute the specialization of each expert, as they are exposed to less data during training. Furthermore, an excessive number of activated experts complicates the task of selecting the most relevant experts for a given input. This challenge mirrors issues observed in methods using task-specific prompts, where an increasing number of tasks makes it progressively harder to identify the correct task identity and its corresponding prompt.

## D.4 PERFORMANCE ACROSS VARYING LOSS WEIGHTS

To enhance the selection process of prompt experts, we introduce two loss functions: $\mathcal{L}_{\text{router}}$ and $\mathcal{L}_{\text{proto}}$. These are designed to encourage expert specialization while preserving previously acquired knowledge. We investigate the impact of their corresponding loss weights, $\alpha_{\text{router}}$ and $\alpha_{\text{proto}}$, as defined in Equation (16). The results, shown in Figure 7, demonstrate that the model achieves optimal

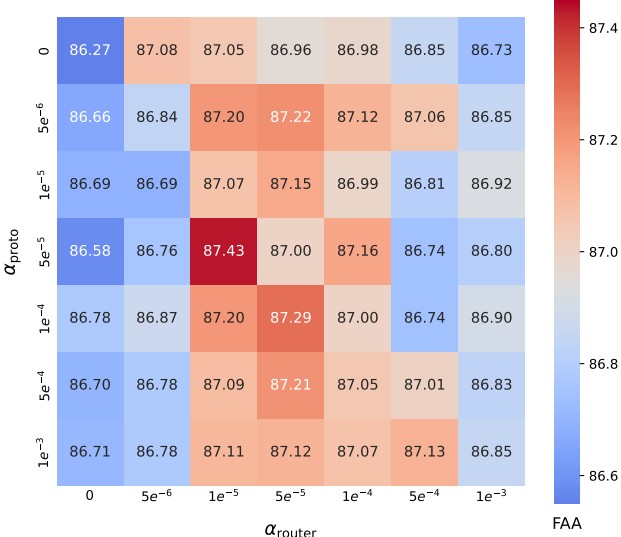

Figure 7: **Impact of $\alpha_{\text{router}}$ and $\alpha_{\text{proto}}$ on Performance.** Performance across different combinations of $\alpha_{\text{router}}$ and $\alpha_{\text{proto}}$ values on the CUB-200 dataset using a 10-task split.

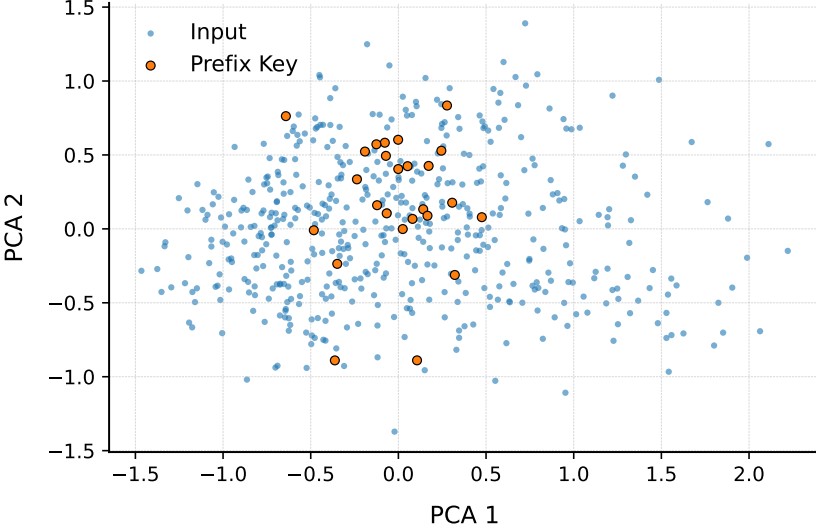

Figure 8: **Visualization of Prefix Keys.** PCA is applied to a representative attention head to visualize the distribution of the corresponding input embeddings and prefix key vectors.

performance when these weights are sufficiently large, suggesting that both components contribute positively to overall performance. Furthermore, performance remains strong across a relatively wide range of values, highlighting the robustness and effectiveness of incorporating $\mathcal{L}_{\text{router}}$ and $\mathcal{L}_{\text{proto}}$ into the training objective.

### D.5 DETAILED ANALYSIS OF PROTOTYPE LOSS

As discussed in Section 3.4, the activation of a prompt expert $f_{N+j'}$ is determined by its score function $\tilde{s}_{j'}$, which in turn depends on the associated prefix key $p_{j'}^{K}$. Thus, the set of prefix keys implicitly defines the regions of the input space where each expert is most responsive. Building on this insight, we propose leveraging prefix keys from previous tasks as prototypes, serving as implicit memory representations of past input distributions. To support this intuition, we select a representative

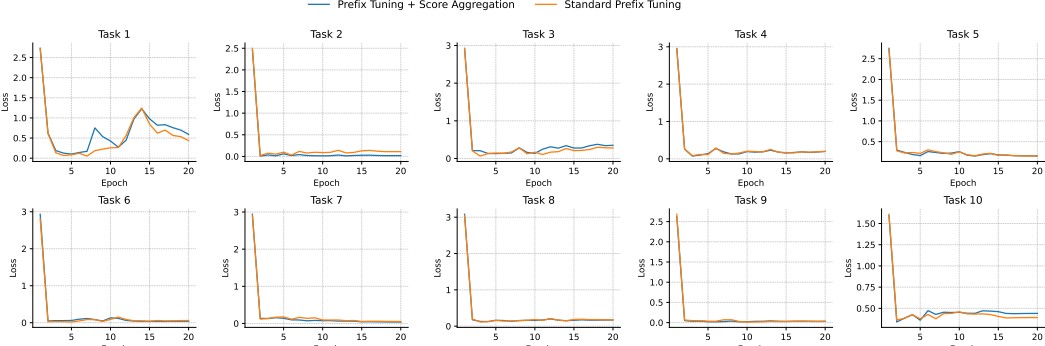

Figure 9: **Sample Efficiency Evaluation.** Validation loss on ImageNet-R (10-task split) measured throughout the training process for each task.

attention head and visualize its corresponding input embeddings alongside the associated prefix keys using Principal Component Analysis (PCA), as shown in Figure 8. The visualization reveals that prefix keys occupy a region of the embedding space densely populated by input samples. This proximity suggests that prefix keys encode salient features of the input distribution, thereby functioning as effective prototypes.

This perspective aligns with findings in recent literature. For example, Chen et al. (2022) showed that routers can learn cluster-center features, facilitating the decomposition of complex tasks into simpler sub-problems addressable by individual experts. Furthermore, the expert selection process can be interpreted as a classification problem, where the prefix keys act as a linear classification head. In this context, recent studies have proposed that the weight vector associated with each class in a classifier can be interpreted as a class prototype (Snell et al., 2017; Qi et al., 2018; Zhao et al., 2024), further supporting our hypothesis.

### D.6 SAMPLE EFFICIENCY EVALUATION

As demonstrated in Appendix A, applying the prompt-attention score aggregation technique yields MoE models within attention heads that maintain the same estimation rate and sample efficiency as those achieved by the standard prefix tuning formulation. To empirically support this theoretical result, we follow Le et al. (2024a) and evaluate sample efficiency by tracking the validation loss on ImageNet-R throughout the training process for each new task. The results, presented in Figure 9, show that both models exhibit comparable convergence rates. This suggests that incorporating prompt-attention score aggregation does not compromise the model's ability to efficiently adapt to new tasks.

### D.7 ANALYSIS OF PROMPT EXPERT ACTIVATION FREQUENCIES

To provide direct evidence for our central claim that sparse activation mitigates interference, this section analyzes how expert activation patterns evolve as the model encounters a sequence of tasks. As illustrated in Figure 10, which visualizes activation frequencies on the CUB-200 dataset, the model initially activates a sparse and concentrated subset of prompt experts. As training progresses across subsequent tasks, the set of activated experts becomes increasingly larger and more diverse.

This dynamic and adaptive activation pattern is fundamental to how our method mitigates interference. In contrast to dense, single-prompt baselines (*e.g.,* OVOR), where a single set of parameters must be continually updated for each new task—risking the overwriting of previously acquired knowledge—our sparse approach distributes the learning load across a broader pool of specialized experts. New tasks can activate and train new or underutilized experts, leaving the parameters of experts essential for earlier tasks largely unaffected. This selective activation is crucial for preserving acquired knowledge while maintaining plasticity for new learning.

Furthermore, this diversification does not preclude effective knowledge reuse. The activation mechanism in the SMoPE framework is input-driven rather than task-ID-driven, as discussed in Section 3.2. Consequently, an expert can be reactivated for subsequent tasks if its specialized function is relevant to the current input. This mechanism facilitates positive knowledge transfer while preventing the destructive interference.

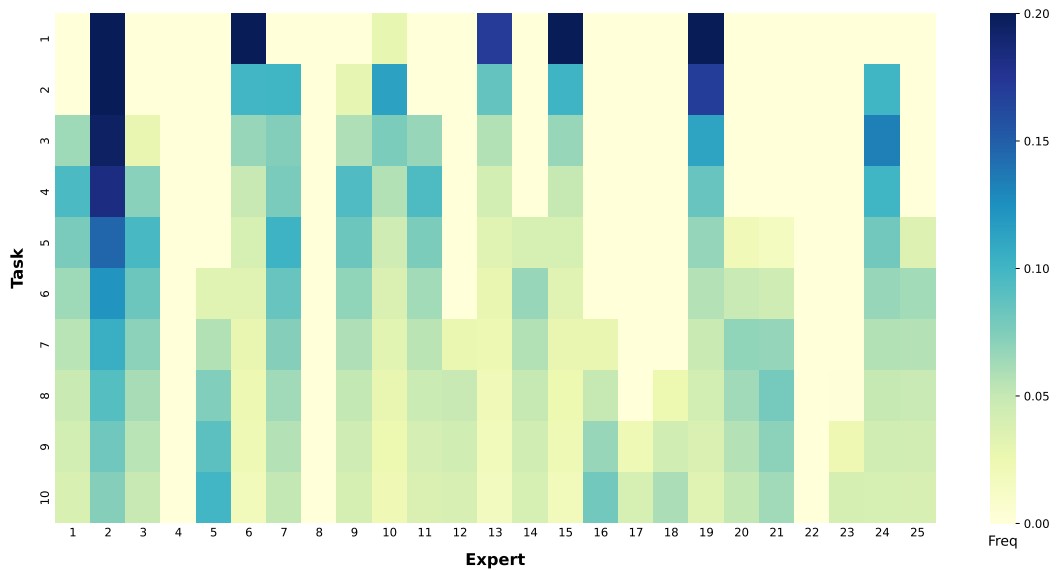

Figure 10: **Evolution of Prompt Expert Activation Frequencies Across Sequential Tasks.** Results on CUB-200 with a prompt length of $N_p = 25$ and $K = 5$. We present the results for a representative attention head and visualize the frequency with which prompt experts are activated across sequential tasks.

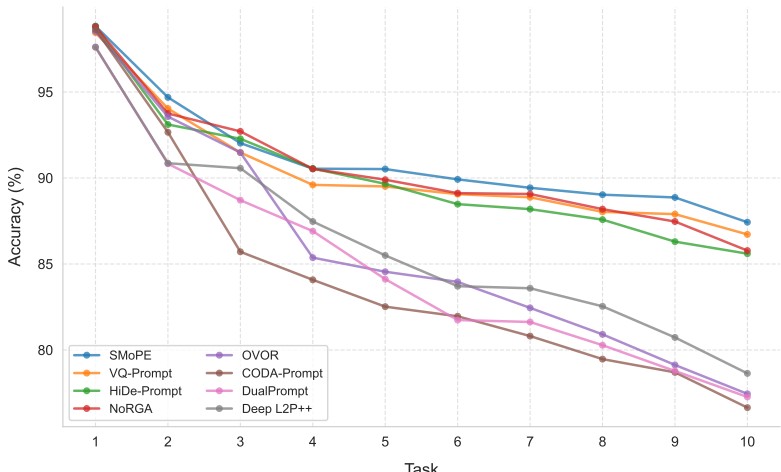

Figure 11: **Accuracy Curves Across Incremental Steps.** Performance of different methods on CUB-200 under the 10-task incremental learning setting.

### D.8    ACCURACY CURVES ACROSS INCREMENTAL STEPS

To provide a more comprehensive understanding of the method's behavior, we present accuracy curves illustrating how FAA performance evolves across sequential tasks on CUB-200 under the 10-task setting. The results are shown in Figure 11. As observed, OVOR performance deteriorates rapidly as training progresses. This decline can be attributed to knowledge interference arising from updating all prompt parameters simultaneously. In contrast, SMoPE maintains consistently strong performance across all tasks. Notably, its accuracy surpasses that of methods employing task-specific prompts, such as HiDe-Prompt. These results demonstrate the effectiveness and robustness of the SMoPE architecture, even when using shared prompt parameters.

### E    USE OF LARGE LANGUAGE MODELS

Large language models were employed solely for editorial purposes, including grammar correction and spelling refinement. They were not used for content generation, data analysis, or the design of experiments.

