# OpenReview forum: "One-Prompt Strikes Back: Sparse Mixture of Experts for Prompt-based Continual Learning"
_ICLR.cc/2026/Conference — ICLR 2026 Poster_

### Official Review · Reviewer_L3Zd · 2025-10-26

**Soundness:** 2
**Presentation:** 3
**Contribution:** 3
**Rating:** 4
**Confidence:** 4

**Summary:**

This paper proposes SMoPE, a framework for continual learning that combines the efficiency of shared prompts with the adaptability of task-specific prompts.SMoPE structures a single shared prompt into multiple “prompt experts” within a sparse mixture-of-experts (MoE) architecture, dynamically activating only the most relevant subset to reduce knowledge interference. It introduces a prompt-attention score aggregation mechanism for efficient expert selection, an adaptive noise mechanism to balance expert utilization, and a prototype-based loss that uses prefix keys as implicit task memory. Experiments on standard benchmarks such as ImageNet-R, CIFAR-100, and CUB-200 show that SMoPE outperforms task-specific prompt methods while cutting computational costs by up to 50%. Overall, SMoPE demonstrates that structured sparsity in shared prompts can effectively mitigate forgetting and achieve state-of-the-art continual learning performance with low computational costs.

**Strengths:**

1. This paper proposes SMoPE, an approach that integrates a sparse mixture of experts architecture into the prefix tuning framework, featuring a prompt-attention score aggregation mechanism for efficient and dynamic expert selection.
2. This paper introduces an adaptive noise mechanism to encourage balanced expert utilization without overwriting prior knowledge, and a prototype-based loss that leverages prefix keys as implicit memory.
3. This paper demonstrates that SMoPE achieves state-of-the-art results on multiple CL benchmarks, while using significantly fewer parameters and reducing computation compared to existing prompt-based methods.

**Weaknesses:**

1. The proposed method integrates multiple components, including a sparse mixture of experts architecture built on prefix tuning, an adaptive noise mechanism, prefix keys as prototypes, and task-adaptive prediction. However, the paper does not include an ablation study to analyze the individual contribution of each component. This omission makes it difficult to assess which parts of the system are most critical to its overall performance.
2. The rationale behind the unified proxy score that aggregates individual scores in Equation (7) is not clearly explained. Without a theoretical or empirical justification for this formulation, it is unclear whether the aggregation is mathematically or conceptually sound.
3. The experimental evaluation is not sufficiently comprehensive. The results are primarily reported for 10-split settings, leaving out potentially informative scenarios such as 5-split, 20-split, and 50-split evaluations. Including these would strengthen the empirical validation and demonstrate the method’s robustness across different data partitioning conditions.

**Questions:**

1. Could the authors provide additional ablation studies to clarify the contribution of each component (sparse moe, adaptive noise mechanism, prefix keys as prototypes, and task-adaptive prediction) to the overall performance?

2. Would it be possible to include more experimental results under different data-split settings (such as 5-split, 20-split, and 50-split) to demonstrate the robustness of the proposed approach? Now only 5-split, 10-split, 20-split for ImageNetR are included in the Appendix.

3. Could the authors elaborate on the motivation and theoretical justification for the unified proxy score in Equation (7)? In particular, how was the aggregation of individual scores determined to be appropriate?

---

> ### Author Response · Authors · 2025-11-14
> **Response to Reviewer L3Zd (Part 1)**
>
> Thank you for your constructive feedback and insightful comments. Below, we provide a point-by-point response and summarize the corresponding revisions in the final version.
>
>
> **Q1: Missing Ablation Study**
>
> We appreciate your comment and would like to clarify that the ablation study is already presented in Section 4.2 and Table 4 of the main text, where we provide a comprehensive analysis of the proposed components. This analysis systematically evaluates the individual contribution of each module. **The quantitative results clearly demonstrate the effect of each element, as also noted by Reviewer QtZY**. Our findings confirm that the integration of all proposed components is essential. **As Reviewer Uup3 observed, this combination effectively balances expert utilization and preserves past knowledge, thereby mitigating catastrophic forgetting.** Additionally, detailed analyses of each component, going beyond simple ablation, are provided in Appendix D.
>
>
> **Q2: Additional Data-Split Settings**
>
> We thank the reviewer for this constructive suggestion. To clarify, we previously included experiments in Appendix D.1 to assess the scalability and generalizability of our method on the ImageNet-R dataset with 5, 10, and 20 splits. **These settings were selected to maintain consistency with prior studies [1, 2], ensuring fair and direct comparability.**
>
> To further address the reviewer’s concern and provide a more comprehensive evaluation of our method’s robustness, we have conducted additional experiments on the ImageNet-R dataset under a more challenging 50-split configuration. The corresponding results have been incorporated into Table 6 of the main paper. A summary of the results under the 50-split setting is presented below:
>
> | **Method**      | **FAA** | **CAA** | **Params (M)** |
> |-------------|:---:|:---:|:----------:|
> | L2P++       |  $55.65 \pm 0.89$   | $64.88 \pm 0.86$    |   1.10         |
> | Deep L2P++  | $58.48 \pm 0.54$    | $67.04 \pm 0.80$    |   4.78         |
> | DualPrompt  | $58.31 \pm 1.18$    |  $65.51 \pm 0.95$   |    4.82        |
> | CODA-Prompt |  $66.96 \pm 0.71$   |  $75.19 \pm 0.72$   |    3.99        |
> | OVOR        |  $65.84 \pm 0.59$   |  $74.84 \pm 0.69$   |     0.26       |
> | HiDe-Prompt |  $70.82 \pm 0.54$   |   $76.81 \pm 0.83$  |     10.35       |
> | NoRGa       |  $71.02 \pm 0.63$   |  $76.99 \pm 0.58$   |        10.35    |
> | VQ-Prompt   | $75.31 \pm 0.61$    |  $81.52 \pm 0.81$   |     0.50       |
> | SMoPE       |  $\textbf{75.54} \pm 0.49$   | $\textbf{81.94} \pm 0.93$    |     $\textbf{0.38}$      |
>
>
> The results demonstrate that SMoPE consistently achieves the best performance in both FAA and CAA metrics while **using significantly fewer parameters** compared to other task-specific prompt methods such as HiDe-Prompt and NoRGa. Notably, it also achieves performance comparable to VQ-Prompt, the recent state-of-the-art method, but with **greater operational efficiency**. SMoPE requires only a single forward pass for sparse expert activation, in contrast to VQ-Prompt's two forward passes to the entire transformer model: one to select prompts via a query-key mechanism and a second to make the final prediction. This strong performance, combined with superior efficiency, highlights the robustness of our approach.
>
>
> [1] OVOR: OnePrompt with Virtual Outlier Regularization for Rehearsal-Free Class-Incremental Learning. ICLR 2024
>
> [2] Vector Quantization Prompting for Continual Learning. NeurIPS 2024.

---

> ### Author Response · Authors · 2025-11-14
> **Response to Reviewer L3Zd (Part 2)**
>
> **Q3: Motivation and Theoretical Justification for the Unified Proxy Score**
>
> We thank the reviewer for this insightful question.
>
> **Motivation**: The primary motivation for introducing the unified proxy score arises from the architectural requirements of integrating the sparse MoE framework with prefix-tuning. In a standard prefix-tuning implementation, as detailed in Section 3.1, each prompt expert $f_{N+j'}(X)$ is associated with multiple score functions $s_{1, N+j'},\dots,s_{N, N+j'}$. **This creates a many-to-one mapping that is inherently incompatible with the gating mechanism of a sparse MoE architecture, which requires a single, comparable criterion to rank and select the top-$K$ experts for a given input.** To bridge this gap, we proposed the unified proxy score $\tilde{s}_{j'}$ in Equation (7). This provides a single score value for each expert, enabling the necessary comparison and routing required by the sparse MoE model.
>
> **Theoretical Justification and Benefits:** Regarding the theoretical justification, a primary concern is whether this simplification from multiple scores to a single proxy score might adversely affect the model's performance. We address this directly in our theoretical analysis. The proofs in Appendix A formally demonstrate that our unified proxy score does not degrade the model's convergence rate. This theoretical finding is further supported by the empirical results presented in Appendix D.6.
>
> In addition to its theoretical soundness, the unified proxy score **substantially improves computational efficiency**. As analyzed in Section 3.2, the original prefix-tuning approach requires calculating all score functions for a single prompt expert, resulting in a computational complexity of $\mathcal{O}(Nd_k)$. In contrast, our unified score reduces this cost to $\mathcal{O}(d_k)$, achieving an $N$-fold speedup. Furthermore, our proxy score functions are directly computed from the input features ${X}$ of the corresponding attention layer. This design allows for the selection of prompt parameters within a single forward pass. By contrast, prior methods such as L2P, DualPrompt, HiDe-Prompt, and VQ-Prompt require two full forward passes of the input through the entire model: one to compute the query features for prompt selection and a second to process the input with the selected prompts. Therefore, **our proposed unified proxy score is not only a necessary component for enabling the integration of the sparse MoE architecture, but it is also a theoretically sound and computationally efficient solution that enhances the overall performance and practicality of the model.**

---

> > ### Comment · Reviewer_L3Zd · 2025-11-16
> >
> > The authors have largely addressed my main concerns in the rebuttal, particularly by highlighting the ablation study results, clarifying the motivation and theoretical justification for the unified proxy score, and providing additional experimental results. Accordingly, I have raised my score from 4 to 6.

---

> > > ### Author Response · Authors · 2025-11-16
> > > **Thank You**
> > >
> > > Dear Reviewer L3Zd,
> > >
> > > We're glad our rebuttal addresses most of your concerns and appreciate that you increase your rating to 6.
> > >
> > > We will incorporate your suggestions into the revision of our paper as discussed. Please feel free to let us know if you have any further concerns.
> > >
> > > Best,
> > >
> > > The Authors

---

### Official Review · Reviewer_3X9D · 2025-10-29

**Soundness:** 3
**Presentation:** 3
**Contribution:** 3
**Rating:** 6
**Confidence:** 4

**Summary:**

This paper tackles the trade-off between efficiency and performance in prompt-based continual learning. Current methods either use task-specific prompts, which suffer from scaling issues and inference overhead, or a single shared prompt, which leads to performance degradation due to interference. The authors propose SMOPE, a method that structures a single shared prompt as a sparse Mixture of Experts. It uses a novel prompt-attention score aggregation mechanism to efficiently select a small subset of "prompt experts" for each input, mitigating interference. Additionally, an adaptive noise mechanism is introduced to balance expert utilization while preserving prior knowledge, and a prototype-based loss leverages prefix keys as implicit memory to enhance specialization. The work effectively combines ideas from MoE and prefix tuning into the single-prompt CL setting, showing strong results.

**Strengths:**

-   The paper addresses a clear and important problem: reconciling the efficiency benefits of a single shared prompt with the performance advantages of task-specific prompts in continual learning.
-   The core idea of interpreting the single prefix prompt pool as a Mixture of Experts and applying sparse activation is novel and interesting, building cleverly upon recent interpretations linking prefix tuning and MoE structures.
-   The experiments are thorough and the presentation is easy to follow, demonstrating strong performance against relevant baselines across multiple datasets and pre-training paradigms.

**Weaknesses:**

-   The prototype loss using prefix keys is an intriguing idea, but its relative contribution compared to the benefits gained from sparse activation and adaptive noise seems modest according to the ablation study. It's unclear how crucial this component is for mitigating forgetting versus simply enabling better expert specialization.
-   The method introduces several components and associated hyperparameters, including the number of selected experts K, the noise level epsilon, and the loss weights alpha_router and alpha_proto. While some sensitivity analyses are provided, the interplay between these hyperparameters and the overall robustness to their selection could be explored further.
-   The central claim is that sparse activation mitigates interference. While plausible and supported by the overall performance gains, the paper lacks direct analysis or visualization showing how expert activation patterns evolve across tasks and whether overlap is indeed reduced compared to a dense single-prompt baseline like OVOR.

**Questions:**

-   Based on the ablation study in Table 4, the prototype loss seems to provide the final increment of performance. How critical is this component for preventing forgetting of past tasks, as opposed to simply improving expert routing for the current task? Could you provide analysis comparing forgetting metrics with and without this loss?
-   Could you provide more details on the selection process for the number of experts K and the loss weighting hyperparameters? How sensitive is the method's performance to variations in K, alpha_router, and alpha_proto, especially concerning their relative balance?
-   Would it be possible to visualize expert activation frequencies or provide statistics showing how expert usage changes across the sequence of tasks? This could offer more direct evidence for the claims about reduced interference and knowledge reuse.

---

> ### Author Response · Authors · 2025-11-14
> **Response to Reviewer 3X9D**
>
> Thank you for your constructive feedback and insightful comments, as well as for giving our paper **a positive score of 6**. Below, we provide a point-by-point response and summarize the corresponding revisions in the final version.
>
> **Q1. Prototype Loss Contribution**
>
> Thank you for your valuable feedback. The prototype loss $\mathcal{L}_\text{proto}$ plays a central role in mitigating catastrophic forgetting. Its design is based on the principle that the set of prefix keys implicitly defines the regions of the input space in which each prompt expert is most responsive. This is further supported by our analysis in Appendix D.5 (Figure 8), which shows that prefix keys occupy regions of the embedding space densely populated by input samples.
>
> Building on this insight, we treat the prefix keys as representative prototypes for previously learned tasks. The prototype loss encourages each expert to maintain its specialization by ensuring continued responsiveness to inputs associated with its learned expertise. This mechanism helps the model remember which inputs should activate specific experts, thereby reducing forgetting.
>
> To empirically validate the efficacy of $\mathcal{L}_\text{proto}$, we conducted an ablation study comparing our full model (SMoPE) against a variant without this loss. The results are summarized in the table below.
>
> |Method|FAA|CAA|Forgetting|
> |-|-|-|-|
> |SMoPE | $\textbf{87.43} \pm 0.39$|$\textbf{91.11} \pm 0.55$ | $3.83 \pm 0.33$      |
> | W/O $\mathcal{L}_\text{proto}$| $87.05 \pm 0.40$ |  $90.47 \pm 0.40$ | $4.28 \pm 0.42$   |
>
> As shown in the table, incorporating the prototype loss yields a notable reduction in forgetting, confirming its effectiveness in preserving prior knowledge.
>
> **Q2: Hyperparameter Selection and Sensitivity**
>
> Thank you for your question. The selection of key hyperparameters, including the number of selected prompt experts $K$ and the loss weighting coefficients $\alpha_\text{router}$ and $\alpha_\text{proto}$ was guided by empirical analyses detailed in the appendices.
>
> For the number of experts, we uniformly set $K = 5$ across all experiments for simplicity. This decision is supported by the sensitivity analysis presented in Appendix D.2, which evaluates the model’s performance across different values of $K$. The weighting coefficients $\alpha_\text{router}$ and $\alpha_\text{proto}$ were determined through a grid search over the set $[5e^{-6},\ 1e^{-5},\ 5e^{-5},\ 1e^{-4},\ 5e^{-4},\ 1e^{-3}]$, as described in Appendix C.
>
> We provide dedicated sensitivity analyses for $K$ in Appendix D.2 and for the loss weights in Appendix D.4. Conducting a joint sensitivity analysis over all hyperparameters is computationally prohibitive and difficult to visualize due to the high-dimensional search space (which would require $6 \times 6 \times 4 = 144$ experiments). Therefore, consistent with standard practice in deep-learning research, we assess the impact of each hyperparameter independently. This approach enables a clear and interpretable understanding of the contribution of each component to overall model performance.
>
> **Q3: Expert Activation Pattern**
>
> We thank the reviewer for this insightful suggestion. To directly address the concern regarding the mechanism by which sparse activation mitigates interference, we have incorporated a new analysis and visualization in Appendix D.7. This addition aims to provide concrete evidence of how expert activation patterns evolve across tasks and to substantiate our central claim.
>
> The visualization (reproduced as Figure 10 in the appendix) illustrates the evolution of expert-activation frequencies over a sequence of incremental tasks. A key observation is that during the initial task, the model activates only a sparse and concentrated subset of prompt experts. As training progresses across subsequent tasks, the set of activated experts becomes increasingly larger and more diverse.
>
> This dynamic and adaptive activation pattern offers direct evidence of how our method mitigates interference. In contrast to dense, single-prompt baselines (e.g., OVOR), where the same parameters must be repeatedly updated for each new task, thereby risking the overwriting of previously learned knowledge, our sparse approach distributes the learning load across a broader pool of specialized experts. New tasks can activate and train new or underutilized experts, leaving the parameters of experts essential for earlier tasks largely unaffected. This selective activation is fundamental to preserving acquired knowledge while maintaining plasticity for new learning.
>
> Importantly, diversification does not preclude knowledge reuse. Instead, activation in the SMoPE framework is input-driven rather than task-ID-driven, as discussed in Section 3.2. Consequently, an expert may be reactivated for later tasks if its specialized function remains relevant to the current input. This mechanism supports positive knowledge transfer while avoiding the destructive interference.

---

### Official Review · Reviewer_QtZY · 2025-10-30

**Soundness:** 3
**Presentation:** 2
**Contribution:** 3
**Rating:** 6
**Confidence:** 5

**Summary:**

This paper introduces SMoPE, a framework for continual learning with prompt that employs a sparse mixture of prompt experts architecture within the prefix-tuning paradigm. By structuring a single shared prompt as multiple prompt experts and using sparse MoE routing, the method aims to mitigate catastrophic forgetting and knowledge interference. Experiments on continual learning benchmarks, including ImageNet-R, CIFAR-100, and CUB-200, demonstrate that SMoPE achieves superior performance to existing methods.

**Strengths:**

1. The authors carefully derive how prefix tuning maps onto a multi-gate MoE, resulting in the novel idea of sparse activation of prompt experts.
2. The empirical results show that SMoPE can outperform both single-prompt and task-specific prompt baselines. The ablation study offers in-depth insight into which components matter, with clear quantitative impacts.

**Weaknesses:**

1. The literature review, though broad, primarily references earlier or closely related prompt-learning studies but omits several recent and highly relevant advances [1][2]. Incorporating these works into the related work section and including them in the comparative evaluation would strengthen the contextual foundation of the paper.
2. While the proposed method is valuable, individual contributions such as the adaptive noise mechanism, prototype-based loss[3], and prompt score aggregation[2] extend or increment existing ideas. The presented extensions are incremental in nature, and although their integration is beneficial, the distinctiveness of each mechanism could be more forcefully argued.

3. The results in Table 6 indicate that SMoPE underperforms compared to existing methods as the number of tasks increases. The authors also acknowledge that interference may persist when task numbers grow. Providing additional analyses, such as accuracy curves across incremental steps, would offer a more comprehensive understanding of the method’s behavior and limitations.

[1] Roy A, Moulick R, Verma V K, et al. Convolutional prompting meets language models for continual learning[C]//Proceedings of the IEEE/CVF conference on computer vision and pattern recognition. 2024: 23616-23626.

[2]Gao Z, Cen J, Chang X. Consistent prompting for rehearsal-free continual learning[C]//Proceedings of the IEEE/CVF Conference on Computer Vision and Pattern Recognition. 2024: 28463-28473.

[3]Zhu F, Zhang X Y, Wang C, et al. Prototype augmentation and self-supervision for incremental learning[C]//Proceedings of the IEEE/CVF conference on computer vision and pattern recognition. 2021: 5871-5880.

**Questions:**

Please refer to Weaknesses

---

> ### Author Response · Authors · 2025-11-14
> **Response to Reviewer QtZY (Part 1)**
>
> Thank you for your constructive feedback and insightful comments, as well as for giving our paper **a positive score of 6**. Below, we provide a point-by-point response and summarize the corresponding revisions in the final version.
>
> **Q1: Comparison with Related Works**
>
> Thank you for highlighting these important works. While several recent studies [1, 2] have made notable strides in advancing prompt-based continual learning, they still face challenges regarding efficiency. Specifically, although [1] introduces a mechanism to mitigate incorrect prompt selection at test time, it continues to follow the conventional paradigm in which the number of prompts grows linearly with the number of tasks. In contrast, the strategy proposed in [2] addresses prompt growth using Large Language Models (LLMs), but this approach incurs substantial computational overhead due to reliance on an external model.
>
> Our method, SMoPE, differs by providing **a more efficient solution**. By employing a single shared prompt across all tasks, SMoPE avoids the linear growth of prompt-related parameters. Moreover, unlike prior task-specific prompting methods, SMoPE does not require precomputed query representations from a full forward pass for prompt selection, thereby reducing both computational and memory costs.
>
> To further substantiate the advantages of SMoPE, we have expanded our discussion of [1] and [2] in Section 2 and added direct comparative evaluations in Section 4 of the revised manuscript (highlighted in blue). Below, we summarize the results on the ImageNet-R dataset under the 10-task setting. Additional details are available in Table 1.
>
> | Method        | Params (M) | FAA (↑)      | CAA (↑)      |
> |----------------|------------|--------------|--------------|
> | Deep L2P++     | 4.78       | 71.66 ± 0.64 | 79.63 ± 0.90 |
> | DualPrompt     | 1.10       | 71.32 ± 0.62 | 78.94 ± 0.72 |
> | CODA-Prompt    | 3.99       | 75.45 ± 0.56 | 81.59 ± 0.82 |
> | OVOR           | 0.26       | 75.25 ± 0.21 | 79.78 ± 0.65 |
> | HiDe-Prompt    | 4.21       | 74.25 ± 0.19 | 79.64 ± 0.53 |
> | NoRGA          | 4.21       | 74.39 ± 0.08 | 79.68 ± 0.41 |
> | ConvPrompt     | 3.35       | 77.30 ± 0.45 | 81.46 ± 0.67 |
> | CPrompt        | 0.92       | 77.14 ± 0.11 | 82.92 ± 0.70 |
> | VQ-Prompt      | 0.50       | 78.71 ± 0.22 | 83.24 ± 0.68 |
> | **SMoPE**      | **0.38**   | **79.32 ± 0.42** | **84.39 ± 0.77** |
>
>
> As demonstrated in our results, SMoPE outperforms these methods on both FAA and CAA, highlighting the **effectiveness and efficiency** of our proposed approach.

---

> ### Author Response · Authors · 2025-11-14
> **Response to Reviewer QtZY (Part 2)**
>
> **Q2: The Distinctiveness of Each Mechanism**
>
> Thank you for this constructive comment and for recognizing the value of our proposed method. We respectfully maintain that the mechanisms introduced in our work are fundamentally distinct from prior approaches in terms of their **formulation, motivation, and objectives**. Below, we elaborate on the novelty and contributions of each component.
>
> - **Adaptive Noise Mechanism:** We believe that our adaptive noise mechanism represents a substantive conceptual departure from prior work (Reviewer L3Zd). Existing methods introduce noise primarily to promote balanced expert utilization in sparse MoE layers. In contrast, our mechanism introduces noise at the **attention-head level**, a structural component already present in the model, rather than at a separate MoE layer. Moreover, the formulation of our adaptive noise is fundamentally different: **it scales the noise magnitude dynamically based on the range of the attention scores**, thereby eliminating the need for manual tuning across heads. Appendix D.3 provides a direct comparison with common noise formulations in the MoE literature.
>
> More importantly, our motivation differs significantly from prior work. Whereas previous methods use noise to balance expert load, our approach is **explicitly designed for continual learning**. Noise is applied selectively to frequently activated experts to preserve their knowledge from earlier tasks, thereby mitigating catastrophic forgetting. This objective clearly differentiates our method from previous noise-based strategies.
>
> - **Prototype-Based Loss**: Our prototype-based loss, which leverages prefix keys, also introduces a distinct perspective (Reviewer 3X9D; Reviewer L3Zd). **Previous work [3] typically constructs prototypes from the final-layer features of the input**, requiring a full forward pass and additional computation to form these prototypes. In contrast, **we directly use the prefix keys as prototypes**, an efficient design that introduces no additional overhead.
>
> **This choice is motivated by our novel observation** that the prefix keys implicitly define the regions of the input space where each prompt expert is most responsive. As shown in Appendix D.5 (Figure 8), prefix keys occupy regions of the embedding space densely populated by input samples. By treating them as prototypes for previously learned tasks, our loss encourages experts to maintain responsiveness to inputs associated with their learned expertise, thereby reducing forgetting.
>
> - **Prompt Score Aggregation:** The prompt score aggregation mechanism serves a fundamentally **different purpose** from prior approaches such as [2]. Earlier work mitigates prompt selection errors by augmenting training with random prompts from previous tasks. Our mechanism instead **addresses an architectural incompatibility between prefix-tuning and sparse MoE frameworks.**
>
> In standard prefix-tuning (Section 3.1), each prompt expert $f_{N+j'}(X)$ is associated with multiple score functions $s_{1, N+j'},\dots,s_{N, N+j'}$, creating a many-to-one mapping that conflicts with MoE gating, which requires a single comparable score per expert. To resolve this, we introduce the unified proxy score $\tilde{s}_{j'}$ in Equation (7), which provides a single criterion for ranking and selecting the top-$K$ experts.
>
> We provide a formal proof in Appendix A that this formulation preserves convergence rates, with empirical validation in Appendix D.6. Furthermore, this mechanism improves computational efficiency by reducing the score computation cost from $\mathcal{O}(Nd_k)$ to $\mathcal{O}(d_k)$, achieving an N-fold reduction in complexity (see Section 3.2). Furthermore, our proxy score functions are directly computed from the input features ${X}$ of the corresponding attention layer. This design allows for the selection of prompt parameters within a single forward pass. By contrast, prior methods such as L2P, DualPrompt, HiDe-Prompt, and VQ-Prompt require two full forward passes of the input through the entire model: one to compute the query features for prompt selection and a second to process the input with the selected prompts. Thus, **the unified proxy score provides a theoretically grounded and computationally efficient solution essential for integrating sparse MoE and prefix-tuning architectures.**
>
> **In summary, we belive that the formulation, motivation, and integration of our proposed mechanisms introduce distinct advances.** Each component addresses a specific challenge in continual learning with sparse MoE models, collectively enhancing the robustness and efficiency of the overall framework.

---

> > ### Author Response · Authors · 2025-11-14
> > **Response to Reviewer QtZY (Part 3)**
> >
> > **Q3: Evaluating Performance with an Increased Number of Tasks**
> >
> > We thank the reviewer for this valuable comment. In our response to Reviewer L3Zd's suggestion, we conducted additional experiments to assess the scalability of our model under a larger number of tasks. The results for the 50-task setting are included in Table 6 of the revised manuscript and summarized below:
> >
> > | **Method**      | **FAA** | **CAA** | **Params (M)** |
> > |-------------|:---:|:---:|:----------:|
> > | L2P++       |  $55.65 \pm 0.89$   | $64.88 \pm 0.86$    |   1.10         |
> > | Deep L2P++  | $58.48 \pm 0.54$    | $67.04 \pm 0.80$    |   4.78         |
> > | DualPrompt  | $58.31 \pm 1.18$    |  $65.51 \pm 0.95$   |    4.82        |
> > | CODA-Prompt |  $66.96 \pm 0.71$   |  $75.19 \pm 0.72$   |    3.99        |
> > | OVOR        |  $65.84 \pm 0.59$   |  $74.84 \pm 0.69$   |     0.26       |
> > | HiDe-Prompt |  $70.82 \pm 0.54$   |   $76.81 \pm 0.83$  |     10.35       |
> > | NoRGa       |  $71.02 \pm 0.63$   |  $76.99 \pm 0.58$   |        10.35    |
> > | VQ-Prompt   | $75.31 \pm 0.61$    |  $81.52 \pm 0.81$   |     0.50       |
> > | SMoPE       |  $\textbf{75.54} \pm 0.49$   | $\textbf{81.94} \pm 0.93$    |     $\textbf{0.38}$      |
> >
> >
> > These results demonstrate that **SMoPE maintains strong performance across both FAA and CAA metrics as the number of tasks increases to 50,** indicating its robustness under more challenging continual learning conditions. Notably, SMoPE achieves this level of performance while utilizing **substantially fewer parameters** than other task-specific prompt-based approaches such as HiDe-Prompt and NoRGa.
> >
> > While SMoPE slightly underperforms VQ-Prompt on the 20-task setting, it surpasses VQ-Prompt under the more demanding 50-task scenario. Our aim is not to emphasize superior absolute performance but to demonstrate that SMoPE delivers **competitive results with state-of-the-art methods** while offering **distinct efficiency advantages**, as highlighted in our abstract.
> >
> > A key advantage of SMoPE over VQ-Prompt is computational efficiency. As shown in Table 4 of the main paper, **SMoPE requires fewer parameters and reduces computation costs by up to 50% during both training and inference.** These findings highlight the dual strengths of SMoPE: its ability to effectively mitigate catastrophic forgetting while maintaining superior computational efficiency.
> >
> >
> > **Q4: Accuracy Curves Across Incremental Steps**
> >
> > Thank you for this valuable suggestion. We have included the accuracy curve and corresponding analysis in Appendix D.8 (highlighted in blue). The results show that OVOR’s performance declines rapidly as training progresses, likely due to knowledge interference caused by updating all prompt parameters simultaneously. In contrast, **SMoPE maintains consistently strong performance across all tasks**, exhibiting stable accuracy throughout incremental training. Notably, its accuracy also surpasses that of task-specific prompt methods such as HiDe-Prompt. These findings underscore the effectiveness and robustness of the SMoPE architecture, even with shared prompt parameters.

---

### Official Review · Reviewer_Uup3 · 2025-10-31

**Soundness:** 3
**Presentation:** 3
**Contribution:** 2
**Rating:** 8
**Confidence:** 4

**Summary:**

This paper proposes SMoPE, a method that integrates a sparse Mixture of Experts (MoE) architecture into prefix tuning for prompt-based continual learning. The core idea structuring a single shared prompt as multiple “prompt experts” and activating only a sparse subset per input is intuitively appealing and aims to reconcile the trade-off between parameter efficiency and performance degradation due to interference. While the paper reports strong empirical results across several benchmarks, I have some concerns regarding the novelty, methodological rigor, and experimental fairness that limit the contribution’s significance.

**Strengths:**

1.  SMoPE achieves SOTA results on multiple continual learning benchmarks while using only a single shared prompt.
2. By activating only K out of Np prompt experts per input, SMoPE reduces both trainable parameters and FLOPs by up to 50% compared to task-specific baselines.
3. The combination of sparse expert routing, adaptive noise, and prototype-based loss successfully balances expert utilization and preserves past knowledge, reducing catastrophic forgetting.

**Weaknesses:**

1. The connection between prefix tuning and MoE is leveraged as the main motivation. However, SMoPE’s key components, prompt-attention score aggregation, adaptive noise for load balancing, and a prototype loss using prefix keysbare largely adaptations of existing techniques (e.g., sparse routing in MoE, load-balancing noise, and prototype-based memory). The score aggregation mechanism simplifies computation but lacks a compelling theoretical justification beyond sample complexity equivalence (which is shown only under strong identifiability assumptions). The method feels more like an engineering combination than a conceptual advance.
2. The paper claims superiority over task-specific prompt methods like HiDe-Prompt and NoRGa. However, it explicitly adopts a corrected implementation from Feng et al. (2025). While this correction may be valid, its use without independent verification or ablation (e.g., showing performance with and without the fix) undermines the fairness of the comparison. If the corrected baselines underperform due to implementation conservatism, SMoPE’s gains may be overstated.
3. SMoPE relies on a fixed-size shared prompt (Np = 25) regardless of the number of tasks (up to 20 in experiments). While results are strong in this regime, the method offers no mechanism to expand capacity as the task sequence grows indefinitely, a fundamental requirement for practical continual learning.
4. The assumption of "strong identifiability" in the supplementary materials is overly idealized and difficult to strictly meet in practical neural network experts.

**Questions:**

Pls see above

---

> ### Author Response · Authors · 2025-11-14
> **Response to Reviewer Uup3 (Part 1)**
>
> Thank you for your constructive feedback and insightful comments, as well as for giving our paper **a positive score of 8**. Below, we provide a point-by-point response and summarize the corresponding revisions in the final version.
>
>
> **Q1: Clarifying the Conceptual Novelty of SMoPE’s Mechanisms**
>
>
> We thank the reviewer for this thoughtful observation. While our work indeed inspired by existing techniques, we respectfully emphasize that our contribution lies in the conceptual unification and reformulation of these components to address the specific challenges of continual learning. The connection between prefix tuning and sparse MoE only serves as a foundational lens through which we design mechanisms explicitly tailored to mitigate catastrophic forgetting. Below, we clarify the theoretical and methodological novelty of each component.
>
>
> - **Prompt Score Aggregation:** To our knowledge, this is the first work to introduce a prompt score aggregation mechanism. This component resolves a fundamental architectural incompatibility between prefix-tuning and sparse MoE frameworks.
>
> In standard prefix-tuning (Section 3.1), each prompt expert $f_{N+j'}(X)$ is associated with multiple score functions $s_{1, N+j'},\dots,s_{N, N+j'}$, resulting in a many-to-one mapping. This is incompatible with the sparse MoE gating mechanism, which requires a single, comparable score for expert selection.
>
> To address this, we introduce a unified proxy score $\tilde{s}_{j'}$ in Equation (7), providing a single criterion for ranking and selecting the top-$K$ experts. We formally prove in Appendix A that this formulation preserves the model’s convergence rate, and our empirical results in Appendix D.6 support this claim. This mechanism also improves computational efficiency, reducing the score computation cost from $\mathcal{O}(Nd_k)$ to $\mathcal{O}(d_k)$, achieving an N-fold reduction in complexity (see Section 3.2).
>
> Furthermore, our proxy score functions are directly computed from the input features ${X}$ of the corresponding attention layer. This design enables prompt selection within a single forward pass. In contrast, prior methods such as L2P, DualPrompt, HiDe-Prompt, and VQ-Prompt require two full forward passes: one to compute query features for prompt selection and another to process the input with the selected prompts. Thus, **the unified proxy score offers both theoretical grounding and computational benefits, making it essential for integrating prefix-tuning with sparse MoE architectures.**
>
>
> - **Adaptive Noise Mechanism:** Our adaptive noise mechanism differs conceptually and structurally from prior load-balancing noise techniques in the MoE literature. Unlike existing methods that inject noise at the MoE layer, our approach applies noise at the attention-head level, a component intrinsic to the model architecture.
>
> Crucially, **the noise magnitude is dynamically scaled according to the range of attention scores**, removing the need for manual per-head hyperparameter tuning. A detailed comparison with conventional formulations is provided in Appendix D.3.
>
> Most importantly, our motivation is distinct from that of prior work. While previous methods use noise primarily to balance expert load, our approach is **explicitly designed for continual learning**. Noise is selectively applied to frequently activated experts to protect their learned knowledge of previous tasks, thereby mitigating catastrophic forgetting. This objective and formulation distinguish our method from existing noise-based strategies.
>
> - **Prototype-Based Loss**: Our use of prefix keys for the prototype-based loss also introduces a novel perspective (Reviewer 3X9D; Reviewer L3Zd). Unlike prior approaches that construct prototypes from final-layer features, we use prefix keys directly as prototypes, incurring no additional computational cost.
>
> This design is motivated by our novel observation that prefix keys implicitly define input-space regions to which each prompt expert is most responsive. As shown in Appendix D.5 (Figure 8), these keys cluster around samples associated with their respective tasks. By treating these keys as prototypes for past tasks, the loss encourages each expert to remain sensitive to its established domain, preserving input–expert associations and mitigating forgetting.
>
> While individual components draw inspiration from established methods, their specific formulations, motivations, and synergistic integration within our framework represent a **conceptual advance** rather than an engineering assembly. Each mechanism addresses a distinct challenge at the intersection of continual learning, prefix-tuning, and sparse MoE architectures. Collectively, they form a principled and computationally efficient framework that enhances both model stability and adaptability in continual learning settings.

---

> ### Author Response · Authors · 2025-11-14
> **Response to Reviewer Uup3 (Part 2)**
>
> **Q2: Comparison with HiDe-Prompt**
>
> Thank you for your comment. A critical aspect of our evaluation methodology is ensuring a fair and rigorous comparison with existing state-of-the-art methods. Our decision to adopt the corrected implementation is motivated by a significant issue in the official HiDe-Prompt code, which, as demonstrated by Feng et al. (2025), leads to an inflation of its performance. This issue arises from the prompt retrieval mechanism, which inadvertently allows task ID information to leak during evaluation.
>
> To empirically illustrate this flaw, we present a comparative analysis of the official HiDe-Prompt implementation under different test-time batch size configurations. As shown in the table below, there is **a stark performance decrease when the test-time batch size is set to 1 compared with a larger batch size of 64**. Because **the model being evaluated is identical** in both scenarios, this discrepancy highlights the information-leakage problem: with a larger batch size, the model has access to batch-level information that would not be available in a practical continual learning setting.
>
> | Method                                | ImageNet-R | CIFAR-100 | CUB-200 |
> |---------------------------------------|:----------:|:---------:|:-------:|
> | HiDe-Prompt (Official, test_bs=64)) |    78.93   |   91.11   |  87.95  |
> | HiDe-Prompt (Official, test_bs=1)  |    71.77   |   85.94   |  84.33  |
> | HiDe-Prompt (Corrected)                     |    74.25   |   88.27   |  85.60  |
> | SMoPE                                 |    79.32   |   89.23   |  87.43  |
>
> Given this evidence, a direct comparison with the flawed official implementation of HiDe-Prompt would not yield a fair or accurate assessment of its capabilities. Therefore, to ensure a valid and scientifically sound comparison, we adopted the corrected prompt retrieval code provided by Feng et al. (2025). Importantly, all other components of the HiDe-Prompt model were preserved exactly as in the original implementation.
>
> By benchmarking SMoPE against this more robust and fairly evaluated baseline, we have greater confidence that the observed performance improvements stem from our proposed architectural and methodological advancements, rather than from inconsistencies in the evaluation of prior work.
>
>
> **Q3: Mechanism to Expand Capacity**
>
>
> We thank the reviewer for this insightful question. As discussed in Section 5, we acknowledge that our current implementation does not include a dynamic capacity-expansion mechanism, and we have identified this as an important direction for future work.
>
> Nevertheless, we believe that the SMoPE architecture can provide a strong and efficient foundation for developing such a mechanism. Given the architectural similarities between SMoPE and sparse MoE, an established methodology for scaling model capacity, we believe that SMoPE holds comparable potential. A straightforward extension, for example, would be to dynamically introduce new prompt experts when the model encounters novel tasks. The central research challenge, however, lies not merely in adding experts but in determining *when* and *how* to do so efficiently. Naively allocating new experts per task, as seen in prior task-specific prompt approaches such as HiDe-Prompt and NoRGa, often leads to inefficient parameter utilization. Our results, which show that SMoPE outperforms these methods despite a fixed parameter budget, underscore this inefficiency and highlight the superior parameter efficiency of our approach.
>
> In summary, while we defer the design of a principled and dynamically expandable mechanism to future work, we maintain that SMoPE can offer a robust and scalable architectural foundation upon which such mechanisms can be effectively built.
>
> **Q4: The assumption of "strong identifiability"**
>
> Thank you for your comment. While we acknowledge that this is a formal condition, we show in Appendix A (L838–841) that it is satisfied by several expert networks commonly used in practice.
>
> For instance, let us consider a widely used two-layer feed-forward network $h(x,(W_1,b,W_2)):=W_2\sigma(W_1x+b)$, where $x,b\in\mathbb{R}^{d}$, $W_1,W_2\in\mathbb{R}^{d\times d}$, and the element-wise activation function $\sigma$ can be ReLU, GELU, sigmoid or a polynomial function $z\mapsto z^p$ for $p\geq 2$. It can be validated that the expert function $h$ is strongly identifiable by taking the derivatives with respect to its parameters and using the definition of linear independence.
>
> Moreover, our use of this assumption aligns with prior theoretical work [1, 2]. It has been employed in recent analyses investigating the theoretical foundations of prefix tuning and its connections to MoE, thereby establishing a clear precedent for its use in this research area.
>
> [1] Revisiting Prefix-tuning: Statistical Benefits of Reparameterization among Prompts. ICLR 2025
>
> [2] Mixture of Experts Meets Prompt-Based Continual Learning. NeurIPS 2024

---

### Author Response · Authors · 2025-11-14
**Response to Reviewers' Comments**

Dear Area Chairs and Reviewers,

We would like to express our sincere gratitude to the reviewers for their time, effort, and constructive feedback in evaluating our submission. Their insightful comments have greatly contributed to enhancing the clarity and completeness of our work. We are particularly encouraged by the positive assessments provided, which highlight the following key aspects of our contribution:

- **Contribution**:
    - **Clear Problem Formulation**: The paper is recognized for addressing a well-defined and important problem: reconciling the efficiency benefits of a shared prompt with the performance advantages of task-specific prompts in continual learning (Reviewer 3X9D). The proposed methodology was also acknowledged as valuable (Reviewer QtZY).
    - **Efficacy in Mitigating Catastrophic Forgetting**: Our proposed method, SMoPE, was commended for achieving state-of-the-art performance on multiple continual learning benchmarks, notably using a single shared prompt (Reviewers Uup3, QtZY, L3Zd).
    - **Computational Efficiency**: A key advantage of SMoPE is its superior efficiency. It demonstrates strong performance with a significant reduction in both parameter count and computational overhead compared to existing prompt-based methods. Specifically, SMoPE reduces trainable parameters and FLOPs by up to 50% relative to task-specific baselines (Reviewers Uup3, L3Zd).
    - **Innovative Adaptive Noise Mechanism**: The introduction of an adaptive noise mechanism was recognized as a novel contribution that facilitates balanced expert utilization without overwriting prior knowledge (Reviewer L3Zd).
    - **Novel Use of Prototype-Based Loss**: The concept of using prefix keys as implicit memory through a prototype-based loss was identified as an intriguing and valuable contribution (Reviewers 3X9D, L3Zd).
    - **Novelty in Sparse Expert Activation**: The core idea of interpreting prefix-tuning within the Mixture of Experts framework and employing sparse activation was seen as both novel and compelling (Reviewers QtZY, 3X9D). This includes a noteworthy prompt-attention score aggregation mechanism for dynamic and efficient expert selection (Reviewer L3Zd).

- **Soundness**:
    - **Methodological Soundness**: The integration of sparse expert routing, adaptive noise, and a prototype-based loss was found to be an effective approach for balancing expert utilization and mitigating catastrophic forgetting (Reviewer Uup3).
    - **Rigorous Experimentation**: The experimental validation was commended for its thoroughness, demonstrating robust performance against relevant baselines across diverse datasets and pre-training paradigms (Reviewer 3X9D).
    - **Insightful Ablation Studies**: The ablation study was acknowledged for providing valuable insights into the contributions of individual components, supported by clear quantitative analysis (Reviewer QtZY).

- **Presentation**: The manuscript was noted for being well-written and easy to follow (Reviewer 3X9D).

We deeply appreciate the reviewers' positive feedback and thoughtful suggestions. We will address each reviewer's specific concerns in our detailed responses below and describe the corresponding revisions made based on their valuable recommendations.

Best regards,

Authors

---

### Author Response · Authors · 2025-12-02
**Discussion Overview for New Area Chair**

Dear Area Chair,

We hope this message finds you well, and we sincerely thank you for your efforts under these challenging circumstances. We understand that revisiting the reverted discussion threads requires additional work, and we greatly appreciate your commitment to maintaining the integrity and fairness of the evaluation process.

To support your assessment, we provide below a concise summary of the key points from the reviews and discussions of our paper prior to the leakage issue. Specifically, we highlight the main elements of our rebuttal before November 27, 2025, to assist your evaluation and final decision:

* *Overview of per-reviewer discussions*

* *Summary of contributions acknowledged by reviewers*

* *Summary of individual reviewer comments and our responses*

Before the data-leakage incident, our paper had received consistently positive evaluations, including **four recommendations for acceptance**. Reviewer L3Zd *raised their score from 4 to 6* after our rebuttal fully addressed their concerns, while the remaining reviewers maintained their initial recommendations for acceptance. Although three reviewers had not yet uploaded their discussion replies due to the shortened discussion window, our detailed responses and additional experimental results were designed to directly and thoroughly address their stated concerns.


****
**Overview of Per-Reviewer Discussion**


| **Reviewer** | **Initial Score** | **Final Score** | **Response**                                                                                                                  |
|--------------|-------------------|-----------------|-------------------------------------------------------------------------------------------------------------------------------|
| **Uup3**     | 8                 | 8               | Did not participate in the discussion but maintained a positive assessment until the discussion period ended.                 |
| **QtZY**     | 6                 | 6               | Did not participate in the discussion but maintained a positive assessment until the discussion period ended.                 |
| **3X9D**     | 6                 | 6               | Did not participate in the discussion but maintained a positive assessment until the discussion period ended.                 |
| **L3Zd**     | 4                 | 6               | Engaged with the rebuttal; all concerns (including ablation studies, additional experiments, and the unified proxy score) were addressed, and the score was raised from 4 to 6. |


****

**Summary of Contributions**

We summarize below the key contributions of our work, as acknowledged by the reviewers:

- **Clear problem formulation**: The paper addresses a well-defined and important problem: reconciling the efficiency of shared prompts with the performance benefits of task-specific prompts in continual learning (Reviewer 3X9D). The overall methodology was also regarded as valuable (Reviewer QtZY).
- **Efficacy in mitigating catastrophic forgetting**: SMoPE was commended for achieving state-of-the-art performance on multiple continual learning benchmarks while using a single shared prompt (Reviewers Uup3, QtZY, L3Zd).
 - **Computational efficiency**: A key advantage of SMoPE is its efficiency. It achieves strong performance with a substantial reduction in both parameter count and computational overhead compared to existing prompt-based methods. In particular, SMoPE reduces trainable parameters and FLOPs by up to 50% relative to task-specific baselines (Reviewers Uup3, L3Zd).
- **Innovative adaptive noise mechanism**: The proposed adaptive noise mechanism was recognized as a novel contribution that encourages balanced expert utilization without overwriting prior knowledge (Reviewer L3Zd).
- **Prototype-based loss with prefix keys as implicit memory**: The use of prefix keys as implicit memory through a prototype-based loss was identified as an intriguing and valuable idea that enhances expert specialization (Reviewers 3X9D, L3Zd).
- **Novel sparse expert activation in a MoE view of prefix-tuning**: The core idea of interpreting prefix-tuning within an MoE framework and employing sparse expert activation was seen as both novel and compelling (Reviewers QtZY, 3X9D). The prompt-attention score aggregation mechanism for dynamic and efficient expert selection was further highlighted as a notable design choice (Reviewer L3Zd).

---

> ### Author Response · Authors · 2025-12-02
>
> **Summary of Individual Reviews**
>
> We summarize below each reviewer’s main concerns and our corresponding responses. We are grateful for the constructive feedback, which has helped improve the paper. All changes made in response to these comments are reflected in the revised manuscript and highlighted in blue.
>
> **1. Reviewer L3Zd: Score increased from 4 to 6**
>
> **Reviewer concerns**: The reviewer requested: (i) ablation study, (ii) additional experiments under a 50-split setting, and (iii) justification for the unified proxy score.
>
> **Our response**:
> * We clarified that the ablation study is already presented in Section 4 of the main paper, where we analyze the contribution of each component. Reviewer QtZY also noted that these ablations clearly demonstrate the impact of each module.
> * We additionally conducted experiments under a more challenging 50-task split to evaluate robustness. The results show that SMoPE outperforms task-specific prompt baselines and achieves performance comparable to recent state-of-the-art methods, while providing substantially better efficiency.
> * For the unified proxy score, we explained that its role is to aggregate multiple expert-specific scores into a single unified score per expert. This design supports sparse expert activation within the sparse MoE architecture and improves computational efficiency. We further provided theoretical justification in Appendix A and empirical validation in Appendix D.6.
>
> As a result, Reviewer L3Zd raised their score from 4 to 6.
>
> **2. Reviewer Uup3: Strongly positive feedback (Score 8)**
>
> **Reviewer concerns**: The reviewer asked about (i) the conceptual novelty of each component in SMoPE, (ii) the comparison with HiDe-Prompt, (iii) mechanisms for capacity expansion, and (iv) the assumption of strong identifiability.
>
> **Our response**:
>
> * We clarified that each component of SMoPE is purposefully designed to address core challenges in prompt-based continual learning. We elaborated on both the theoretical and methodological novelty of these components earlier in the discussion.
> * We corrected inaccuracies in the original HiDe-Prompt implementation that had led to inflated results. For a fair comparison, we adopted the correct prompt-retrieval code and re-ran experiments, confirming that SMoPE’s gains stem from our architectural and methodological contributions rather than inconsistencies in prior evaluation.
> * Regarding capacity expansion, we acknowledged that the current version of SMoPE does not yet include an explicit expansion mechanism, but we emphasized that the framework offers a solid basis on which such extensions can be developed in future work.
> * We clarified, with a concrete example, that the strong identifiability assumption is satisfied by several expert network structures commonly used in practice, as already discussed in Appendix A.
>
> **3. Reviewer QtZY: Positive feedback (Score 6)**
>
> **Reviewer concerns**: The reviewer suggested: (i) adding two recent prompt-based continual learning methods (ii) including an additional analysis to better illustrate SMoPE’s behavior, and (iii) clarifying the distinctiveness of each component.
>
> **Our response**:
>
> * We incorporated the suggested works into Section 2 of the revised manuscript, thereby providing a more comprehensive literature review. We also added experimental comparisons demonstrating SMoPE’s performance improvements over these methods.
> * We clarified that the mechanisms introduced in SMoPE are distinct in their formulation, motivation, and objectives. The novelty and role of each component were elaborated to clearly differentiate them from existing approaches.
> * To further illustrate SMoPE’s behavior, we added incremental accuracy curves, which highlight the robustness and effectiveness of the architecture even when using shared prompt parameters.
>
> **4. Reviewer 3X9D: Positive feedback (Score 6)**
>
> **Reviewer concerns**: he reviewer requested: (i) more evidence of the contribution of the prototype-based loss, (ii) more details on hyperparameter selection and sensitivity, and (iii) visualizations of expert frequencies to reveal how expert usage evolves across tasks.
>
> **Our response**:
> * We analyzed the role of the prototype-based loss and conducted additional experiments showing that it encourages each expert to specialize on inputs within its learned domain, thereby reducing forgetting.
> * We provided detailed information on hyperparameter selection, including the number of experts and the weighting coefficients of the loss terms, in Appendices C and D of the revised manuscript.
> * We added visualizations of expert frequencies in Appendix D.8, showing that SMoPE mitigates interference relative to a dense single-prompt baseline. Our mechanism activates a more diverse and larger set of experts as tasks progress, effectively reducing overwriting of previously learned knowledge.

---

### Meta-Review · Area_Chair_FMCt · 2026-01-07

**Summary:**

The reviewers’ main concerns center on the perceived incremental nature of the contribution, the clarity and justification of the formulation (especially Eq. 7), and whether the experimental evidence sufficiently supports the claims.

In the rebuttal, the authors clarify the motivation and objectives of the method, provide a more detailed justification of Eq. 7, and add additional experimental results that directly address these concerns. These revisions improve the clarity of the paper and strengthen the empirical support.

While the method builds on existing components, the reviewers generally agree that their integration and empirical validation constitute a meaningful contribution. Since the major concerns are adequately addressed and the remaining issues primarily relate to positioning rather than correctness or significance, I recommend acceptance.

**Reviewer Concerns:**

In the initial reviews, three reviewers give positive scores, while Reviewer L3Zd gives a negative score.

The positive reviewers raise largely similar concerns: some components of the proposed method build on existing techniques, and the manuscript initially lacks sufficiently deep analysis of these components, which weakens the perceived strength of the contribution. At the same time, these reviewers also acknowledge that the experimental results indicate tangible benefits from integrating these components. In the rebuttal, the authors provide clarifications from the perspectives of formulation, motivation, and objectives, which are clear and helpful. I therefore expect that these reviewers maintain or slightly increase their scores.

Reviewer L3Zd’s main concerns are that the presentation is not sufficiently comprehensive and that the motivation and analysis surrounding Eq. 7 are unclear. In the response, the authors offer a more detailed justification of Eq. 7 and add further experimental evidence, including additional results that also address concerns raised by other reviewers. These additions substantially strengthen the paper, and the experimental support is now sufficient. Moreover, Reviewer L3Zd explicitly indicates a willingness to raise their score in light of these changes. I therefore believe that this reviewer likely increases their score if they are able to participate fully in the discussion.

**Reviewer Scores:**

See Reviewer Concerns

---

### Decision · Program_Chairs · 2026-01-26

Accept (Poster)